# Patient-specific Boolean models of signalling networks guide personalised treatments

Arnau Montagud[1,2,3,4]*, Jonas Béal[1,2,3], Luis Tobalina[5‡], Pauline Traynard[1,2,3], Vigneshwari Subramanian[5§], Bence Szalai[5,6], Róbert Alföldi[7], László Puskás[7], Alfonso Valencia[4,8], Emmanuel Barillot[1,2,3], Julio Saez-Rodriguez[5,9†], Laurence Calzone[1,2,3]*†

[1]Institut Curie, PSL Research University, Paris, France; [2]INSERM, U900, Paris, France; [3]MINES ParisTech, PSL Research University, CBIO-Centre for Computational Biology, Paris, France; [4]Barcelona Supercomputing Center (BSC), Plaça Eusebi Güell, 1-3, Barcelona, Spain; [5]Faculty of Medicine, Joint Research Centre for Computational Biomedicine (JRC-COMBINE), RWTH Aachen University, Aachen, Germany; [6]Semmelweis University, Faculty of Medicine, Department of Physiology, Budapest, Hungary; [7]Astridbio Technologies Ltd, Szeged, Hungary; [8]ICREA, Pg. Lluís Companys 23, Barcelona, Spain; [9]Faculty of Medicine and Heidelberg University Hospital, Institute of Computational Biomedicine, Heidelberg University, Heidelberg, Germany

*For correspondence:
arnau.montagud@bsc.es (AM);
laurence.calzone@curie.fr (LC)

†These authors contributed equally to this work

Present address: ‡Bioinformatics and Data Science, Research and Early Development, Oncology R&D, AstraZeneca, Cambridge, United Kingdom; §Data Science & Artificial Intelligence, Imaging & Data Analytics, Clinical Pharmacology & Safety Sciences, R&D, AstraZeneca, Gothenburg, Sweden

**Abstract** Prostate cancer is the second most occurring cancer in men worldwide. To better understand the mechanisms of tumorigenesis and possible treatment responses, we developed a mathematical model of prostate cancer which considers the major signalling pathways known to be deregulated. We personalised this Boolean model to molecular data to reflect the heterogeneity and specific response to perturbations of cancer patients. A total of 488 prostate samples were used to build patient-specific models and compared to available clinical data. Additionally, eight prostate cell line-specific models were built to validate our approach with dose-response data of several drugs. The effects of single and combined drugs were tested in these models under different growth conditions. We identified 15 actionable points of interventions in one cell line-specific model whose inactivation hinders tumorigenesis. To validate these results, we tested nine small molecule inhibitors of five of those putative targets and found a dose-dependent effect on four of them, notably those targeting HSP90 and PI3K. These results highlight the predictive power of our personalised Boolean models and illustrate how they can be used for precision oncology.

## Editor's evaluation

This paper presents a mathematical model for prioritizing drugs for prostate cancer patients based on signal network database. The manuscript is of broad interest to the field of oncology and precision medicine. The methodology developed is sophisticated and relevant to real patient prostate cancer data. The predictions from the model are validated in an experimental setting and provide suggestions for the personalisation of prostate cancer treatment. The study can serve as a roadmap for future development of predictive, personalized models.

## Introduction

Like most cancers, prostate cancer arises from mutations in single somatic cells that induce deregulations in processes such as proliferation, invasion of adjacent tissues and metastasis. Not all prostate patients respond to the treatments in the same way, depending on the stage and type of their tumour (*Chen and Zhou, 2016*) and differences in their genetic and epigenetic profiles (*Toth et al., 2019*; *Yang et al., 2018*). The high heterogeneity of these profiles can be explained by a large number of interacting proteins and the complex cross-talks between the cell signalling pathways that can be altered in cancer cells. Because of this complexity, understanding the process of tumorigenesis and tumour growth would benefit from a systemic and dynamical description of the disease. At the molecular level, this can be tackled by a simplified mechanistic cell-wide model of protein interactions of the underlying pathways, dependent on external environmental signals.

Although continuous mathematical modelling has been widely used to study cellular biochemistry dynamics (e.g. ordinary differential equations) (*Goldbeter, 2002*; *Kholodenko et al., 1995*; *Le Novère, 2015*; *Sible and Tyson, 2007*; *Tyson et al., 2019*), this formalism does not scale up well to large signalling networks, due to the difficulty of estimating kinetic parameter values (*Babtie and Stumpf, 2017*). In contrast, the logical (or logic) modelling formalism represents a simpler means of abstraction where the causal relationships between proteins (or genes) are encoded with logic statements, and dynamical behaviours are represented by transitions between discrete states of the system (*Kauffman, 1969*; *Thomas, 1973*). In particular, Boolean models, the simplest implementation of logical models, describe each protein as a binary variable (ON/OFF). This framework is flexible, requires in principle no quantitative information, can be hence applied to large networks combining multiple pathways, and can also provide a qualitative understanding of molecular systems lacking detailed mechanistic information.

In the last years, logical and, in particular, Boolean modelling has been successfully used to describe the dynamics of human cellular signal transduction and gene regulations (*Calzone et al., 2010*; *Cho et al., 2016*; *Flobak et al., 2015*; *Grieco et al., 2013*; *Helikar et al., 2008*; *Traynard et al., 2016*) and their deregulation in cancer (*Fumiã and Martins, 2013*; *Hu et al., 2015*). Numerous applications of logical modelling have shown that this framework is able to delineate the main dynamical properties of complex biological regulatory networks (*Abou-Jaoudé et al., 2011*; *Fauré et al., 2006*).

However, the Boolean approach is purely qualitative and does not consider the real time of cellular events (half time of proteins, triggering of apoptosis, etc.). To cope with this issue, we developed the MaBoSS software to compute continuous Markov Chain simulations on the model state transition graph (STG), in which a model state is defined as a vector of nodes that are either active or inactive. In practice, MaBoSS associates transition rates for activation and inhibition of each node of the network, enabling it to account for different time scales of the processes described by the model. Given some initial conditions, MaBoSS applies a Monte-Carlo kinetic algorithm (or Gillespie algorithm) to the STG to produce time trajectories (*Stoll et al., 2017*; *Stoll et al., 2012*) such that the time evolution of the model state probabilities can be estimated. Stochastic simulations can easily explore the model dynamics with different initial conditions by varying the probability of having a node active at the beginning of the simulations and by modifying the model such that it accounts for genetic and environmental perturbations (e.g. presence or absence of growth factors or death receptors). For each case, the effect on the probabilities of selected read-outs can be measured (*Cohen et al., 2015*; *Montagud et al., 2019*).

When summarising the biological knowledge into a network and translating it into logical terms, the obtained model is generic and cannot explain the differences and heterogeneity between patients' responses to treatments. Models can be trained with dedicated perturbation experiments (*Dorier et al., 2016*; *Saez-Rodriguez et al., 2009*), but such data can only be obtained with non-standard procedures such as microfluidics from patients' material (*Eduati et al., 2020*). To address this limitation, we developed a methodology to use different omics data that are more commonly available to personalise generic models to individual cancer patients or cell lines and verified that the obtained models correlated with clinical results such as patient survival information (*Béal et al., 2019*). In the present work, we apply this approach to prostate cancer to suggest targeted therapy to patients based on their omics profile (*Figure 1*). We first built 488 patient- and eight cell line prostate-specific models using data from The Cancer Genome Atlas (TCGA) and the Genomics of Drug Sensitivity in Cancer (GDSC) projects, respectively. Simulating these models with the MaBoSS framework, we identified

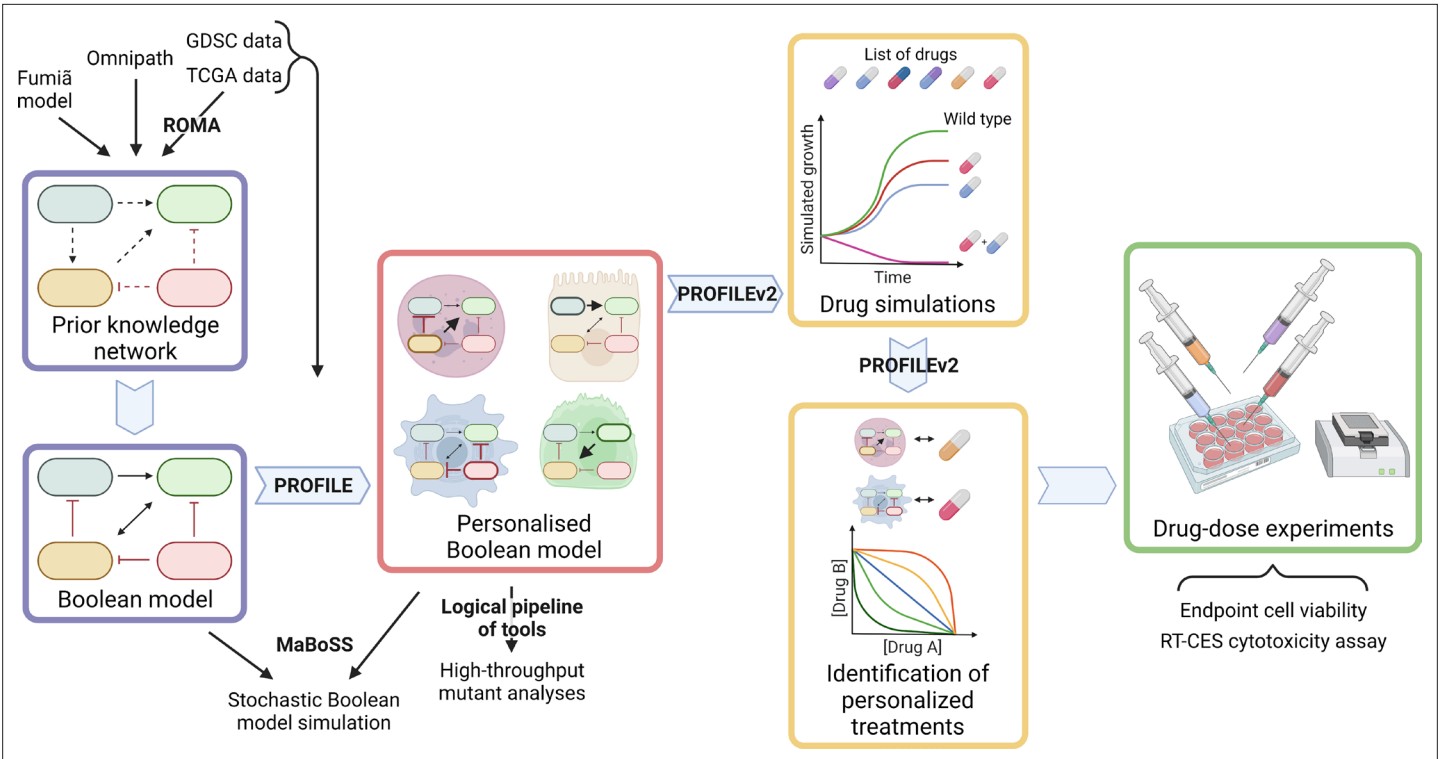

**Figure 1.** Workflow to build patient-specific Boolean models and to uncover personalised drug treatments from present work. We gathered data from *Fumiã and Martins, 2013* Boolean model, Omnipath (*Türei et al., 2021*) and pathways identified with ROMA (*Martignetti et al., 2016*) on the TCGA data to build a prostate-specific prior knowledge network. This network was manually converted into a prostate Boolean model that could be stochastically simulated using MaBoSS (*Stoll et al., 2017*) and tailored to different TCGA and GDSC datasets using our PROFILE tool to have personalised Boolean models. Then, we studied all the possible single and double mutants on these tailored models using our logical pipeline of tools (*Montagud et al., 2019*). Using these personalised models and our PROFILE_v2 tool presented in this work, we obtained tailored drug simulations and drug treatments for 488 TCGA patients and eight prostate cell lines. Lastly, we performed drug-dose experiments on a shortlist of candidate drugs that were particularly interesting in the LNCaP prostate cell line. Created with BioRender.com.

points of intervention that diminish the probability of reaching pro-tumorigenic phenotypes. Lastly, we developed a new methodology to simulate drug effects on these data-tailored Boolean models and present a list of viable drugs and treatments that could be used on these patient- and cell line-specific models for optimal results. Experimental validations were performed on the LNCaP prostate cell line with two predicted targets, confirming the predictions of the model.

## Results
### Prostate Boolean model construction

A network of signalling pathways and genes relevant for prostate cancer progression was assembled to recapitulate the potential deregulations that lead to high-grade tumours. Dynamical properties were added onto this network to perform simulations, uncover therapeutic targets and explore drug combinations. The model was built upon a generic cancer Boolean model by *Fumiã and Martins, 2013*, which integrates major signalling pathways and their substantial cross-talks. The pathways include the regulation of cell death and proliferation in many tumours.

This initial generic network was extended to include prostate cancer-specific genes (e.g. SPOP, AR, etc.), pathways identified using ROMA (*Martignetti et al., 2016*), OmniPath (*Türei et al., 2021*), and up-to-date literature. ROMA is applied on omics data, either transcriptomics or proteomics. In each pathway, the genes that contribute the most to the overdispersion are selected. ROMA was applied to the TCGA transcriptomics data using gene sets from cancer pathway databases (Appendix 1, Section 1.1.3, *Appendix 1—figure 1*). These results were used as guidelines to extend the network to fully cover the alterations found in prostate cancer patients. OmniPath was used to complete our

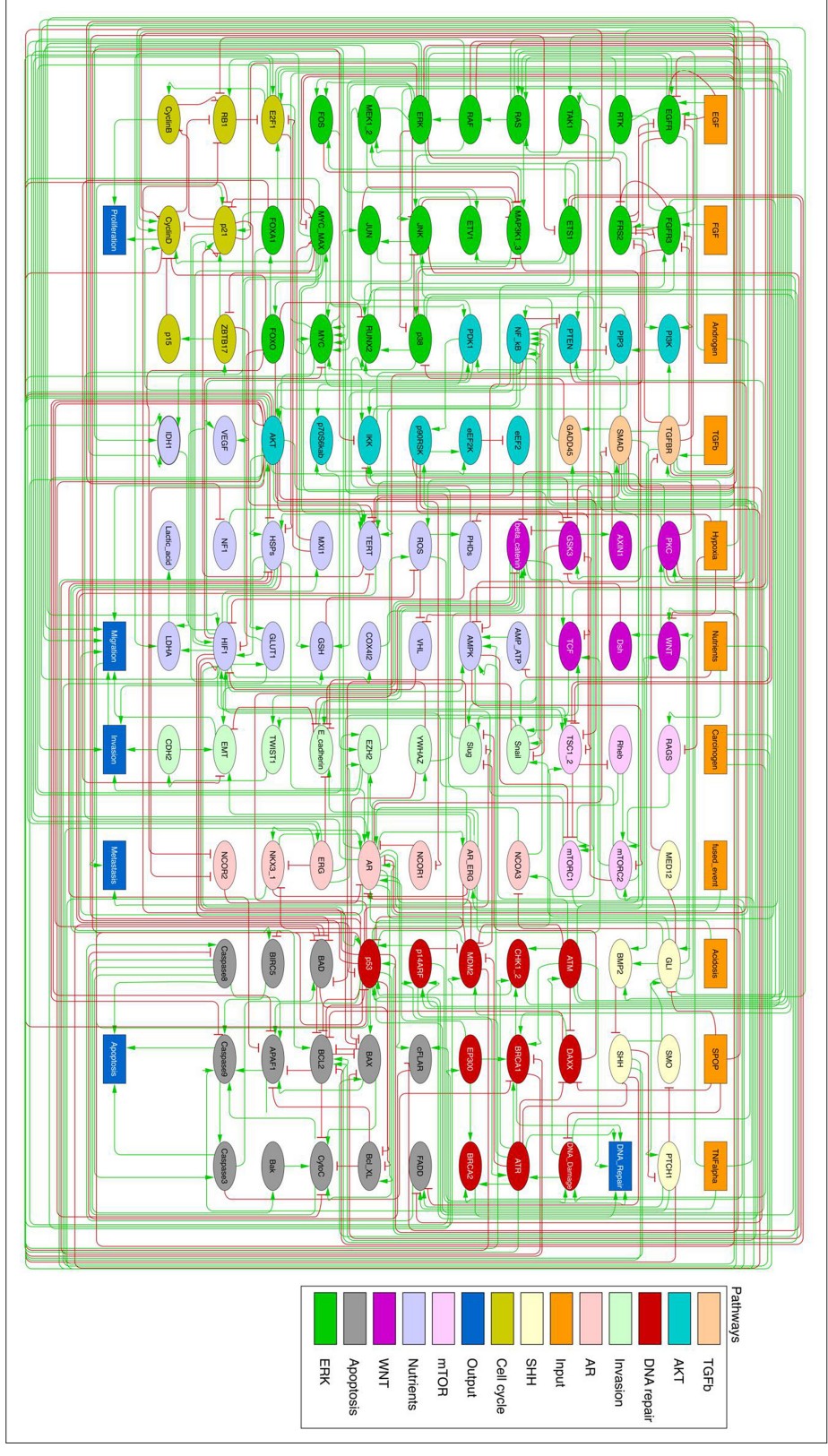

**Figure 2.** Prostate Boolean model used in present work. Nodes (ellipses) represent biological entities, and arcs are positive (green) or negative (red) influences of one entity on another one. Orange rectangles correspond to inputs (from left to right: Epithelial Growth Factor (EGF), Fibroblast Growth Factor (FGF), Transforming Growth Factor beta (TGFbeta), Nutrients, Hypoxia, Acidosis, Androgen, fused_event, Tumour Necrosis Factor alpha (TNFalpha),

*Figure 2 continued on next page*

*Figure 2 continued*
SPOP, Carcinogen) and dark blue rectangles to outputs that represent biological phenotypes (from left to right: Proliferation, Migration, Invasion, Metastasis, Apoptosis, DNA_repair), the read-outs of the model. This network is available to be inspected as a Cytoscape file in the **Supplementary file 1**.

network finding connections between the proteins of interest known to play a role in the prostate and the ones identified with ROMA, and the list of genes already present in the model (Appendix 1, Sections 1.1.3 and 1.1.4, *Appendix 1—figures 2 and 3*). The final network includes pathways such as androgen receptor, MAPK, Wnt, NFkB, PI3K/AKT, MAPK, mTOR, SHH, the cell cycle, the epithelial-mesenchymal transition (EMT), apoptosis and DNA damage pathways.

This network was then converted into a Boolean model where variables can take two values: 0 (inactivate or absent) or 1 (activate or present). Our model aims at predicting prostate phenotypic behaviours for healthy and cancer cells in different conditions. Nine inputs that represent some of these physiological conditions of interest were considered: *Epithelial Growth Factor (EGF)*, *Fibroblast Growth Factor (FGF)*, *Transforming Growth Factor beta (TGFbeta)*, *Nutrients*, *Hypoxia*, *Acidosis*, *Androgen*, *Tumour Necrosis Factor alpha (TNF alpha)*, and *Carcinogen*. These input nodes have no regulation. Their value is fixed according to the simulated experiment to represent the status of the microenvironmental characteristics (e.g. the presence or absence of growth factors, oxygen, etc.). A more complex multiscale approach would be required to consider the dynamical interaction with other cell types and the environment.

We defined six variables as output nodes that allow the integration of multiple phenotypic signals and simplify the analysis of the model. Two of these phenotypes represent the possible growth status of the cell: *Proliferation* and *Apoptosis*. *Apoptosis* is activated by Caspase 8 or Caspase 9, while *Proliferation* is activated by cyclins D and B (read-outs of the G1 and M phases, respectively). The *Proliferation* output is described in published models as specific stationary protein activation patterns, namely the following sequence of activation of cyclins: Cyclin D, then Cyclin E, then Cyclin A, and finally Cyclin B (*Traynard et al., 2016*). Here, we considered a proper sequence when Cyclin D activates first, allowing the release of the transcriptional factor E2F1 from the inhibitory complex it was forming with the RB (retinoblastoma protein), and then triggering a series of events leading to the activation of Cyclin B, responsible for the cell's entry into mitosis (Appendix 1, Section 2.2, *Appendix 1—figure 5*). We also define several phenotypic outputs that are readouts of cancer hallmarks: *Invasion*, *Migration*, (bone) *Metastasis* and *DNA repair*. The final model accounts for

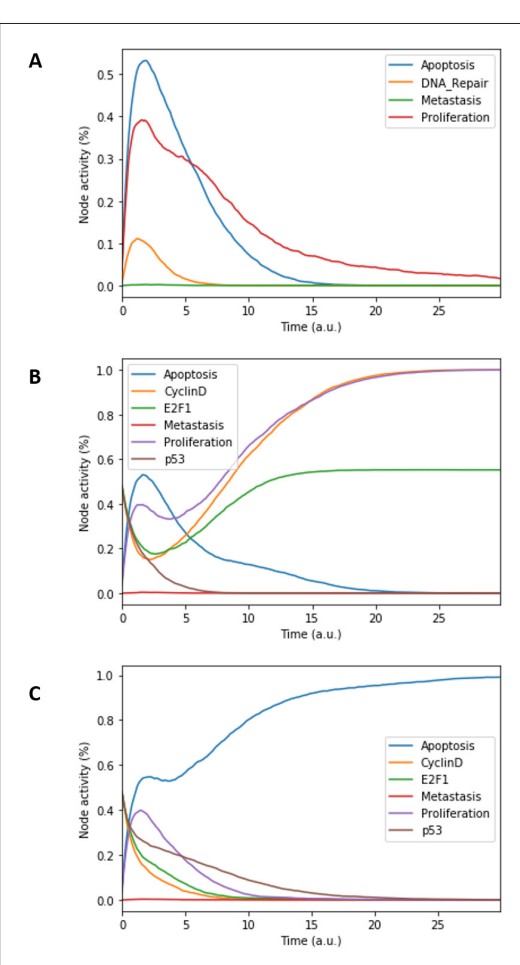

**Figure 3.** Prostate Boolean model MaBoSS simulations. (**A**) The model was simulated with all initial inputs set to 0 and all other variables random. All phenotypes are 0 at the end of the simulations, which should be understood as a quiescent state, where neither proliferation nor apoptosis is active. (**B**) The model was simulated with growth factors (*EGF* and *FGF*), *Nutrients* and *Androgen* ON. (**C**) The model was simulated with *Carcinogen, Androgen, TNFalpha, Acidosis*, and *Hypoxia* ON.

133 nodes and 449 edges (*Figure 2*, *Supplementary file 1*, and in GINsim format at the address: http://ginsim.org/model/signalling-prostate-cancer).

## Prostate Boolean model simulation

The model can be considered as a model of healthy prostate cells when no mutants (or fused genes) are present. We refer to this model as the wild type model. These healthy cells mostly exhibit quiescence (neither proliferation nor apoptosis) in the absence of any input (*Figure 3A*). When *Nutrients* and growth factors (*EGF* or *FGF*) are present, *Proliferation* is activated (*Figure 3B*). *Androgen* is necessary for AR activation and helps in the activation of *Proliferation*, even though it is not necessary when *Nutrients* or growth factors are present. Cell death factors (such as Caspase 8 or 9) trigger *Apoptosis* in the absence of *SPOP*, while *Hypoxia* and *Carcinogen* facilitate apoptosis but are not necessary if cell death factors are present (*Figure 3C*).

In our model, the progression towards metastasis is described as a stepwise process. *Invasion* is first activated by known pro-invasive proteins: either β-catenin (*Francis et al., 2013*) or a combination of *CDH2* (*De Wever et al., 2004*), *SMAD* (*Daroqui et al., 2012*), or *EZH2* (*Ren et al., 2012*). *Migration* is then activated by *Invasion* and *EMT* and with either *AKT* or *AR* (*Castoria et al., 2011*). Lastly, (bone) *Metastasis* is activated by *Migration* and one of three nodes: *RUNX2* (*Altieri et al., 2009*), *ERG* (*Adamo and Ladomery, 2016*) or ERG fused with TMPRSS2 (*St John et al., 2012*), FLI1, ETV1 or ETV4 (*Cancer Genome Atlas Research Network, 2015*).

This prostate Boolean model was simulated stochastically using MaBoSS (*Stoll et al., 2017*; *Stoll et al., 2012*) and validated by recapitulating known phenotypes of prostate cells under physiological conditions (*Figure 3* and Appendix 1, Sections 2.2 and 2.3, *Appendix 1—figures 5–7*). In particular, we tested that combinations of inputs lead to non-aberrant phenotypes such as growth factors leading to apoptosis in wild type conditions; we also verified that the cell cycle events occur in proper order: as CyclinD gets activated, RB1 is phosphorylated and turned OFF, allowing E2F1 to mediate the synthesis of CyclinB (see *Supplementary file 2* for the jupyter notebook and the simulation of diverse cellular conditions).

## Personalisation of the prostate Boolean model

### Personalised TCGA prostate cancer patient Boolean models

We tailored the generic prostate Boolean model to a set of 488 TCGA prostate cancer patients (Appendix 1, Section 4, *Appendix 1—figure 9*) using our personalisation method (PROFILE) (*Béal et al., 2019*), constructing 488 individual Boolean models, one for each patient. Personalised models were built using three types of data: discrete data such as mutations and copy number alterations (CNA) and continuous data such as RNAseq data. For discrete data, the nodes corresponding to the mutations or the CNA were forced to 0 or 1 according to the effect of alterations, based on a priori knowledge (i.e. if the mutation was reported to be activating or inhibiting the gene's activity). For continuous data, the personalisation method modifies the value for the transition rates of model variables and their initial conditions to influence the probability of some transitions. This corresponds, in a biologically meaningful way, to translating genetic mutations as lasting modifications making the gene independent of regulation, and to translating RNA expression levels as modulation of a signal but not changing the regulation rules (see Materials and methods and in Appendix 1, Section 4.1, *Appendix 1—figures 10–14*).

We assess the general behaviour of the individual patient-specific models by comparing the model outputs (i.e. probabilities to reach certain phenotypes) with clinical data. Here, the clinical data consist of a Gleason grade score associated with each patient, which in turn corresponds to the gravity of the tumour based on its appearance and the stage of invasion (*Chen and Zhou, 2016*; *Gleason, 1992*; *Gleason, 1977*). We gathered the output probabilities for all patient-specific models and confronted them to their Gleason scores. The phenotype *DNA_repair*, which can be interpreted as a sensor of DNA damage and genome integrity which could lead to DNA repair, seems to separate low and high Gleason scores (*Figure 4A* and Appendix 1, Section 4.1, *Appendix 1—figures 15–18*), confirming that DNA damage pathways are activated in patients (*Marshall et al., 2019*) but may not lead to the triggering of apoptosis in this model (Appendix 1, Section 4.1, *Appendix 1—figure 11*). Also, the centroids of Gleason grades tend to move following *Proliferation*, *Migration* and *Invasion* variables. We then looked at the profiles of the phenotype scores across patients and their Gleason grade and

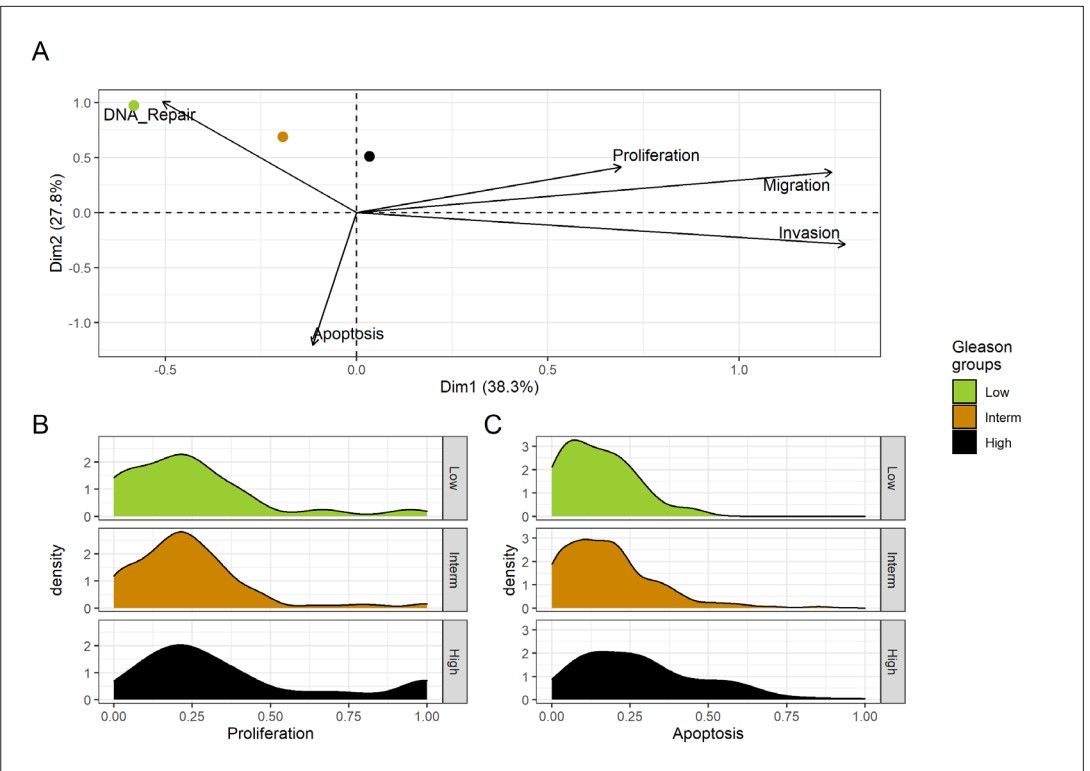

**Figure 4.** Associations between simulations and Gleason grades (GG). (**A**) Centroids of the Principal Component Analysis of the samples according to their Gleason grades (GG). The personalisation recipe used was mutations and copy number alterations (CNA) as discrete data and RNAseq as continuous data. Density plots of *Proliferation* (**B**) and *Apoptosis* (**C**) scores according to GG; each vignette corresponds to a specific sub-cohort with a given GG. Kruskal-Wallis rank sum test across GG is significant for Proliferation (p-value = 0.00207) and Apoptosis (p-value = 2.83E-6).

The online version of this article includes the following source data for figure 4:

**Source code 1.** R code needed to obtain *Figure 4*.

**Source data 1.** Processed dataset needed to obtain the phenotype distributions of *Figure 4B, C*, with *Figure 4— source code 1*.

**Source data 2.** Processed dataset needed to obtain the PCA of *Figure 4A*, with *Figure 4—source code 1*.

found that the density of high *Proliferation* score (close to 1, *Figure 4B*) tends to increase as the Gleason score increases (from low to intermediate to high) and these distributions are significantly different (Kruskal-Wallis rank sum test, p-value = 0.00207; Appendix 1, Section 4.1). The *Apoptosis* phenotype probabilities, however, do not have a clear trend across grades (*Figure 4C*), even though the distributions are significantly different (Kruskal-Wallis rank sum test, p-value = 2.83E-6; Appendix 1, Section 4.1).

## Personalised drug predictions of TCGA Boolean models

Using the 488 TCGA patient-specific models, we looked in each patient for genes that, when inhibited, hamper *Proliferation* or promote *Apoptosis* in the model. We focused on these inhibitions as most drugs interfere with the protein activity related to these genes, even though our methodology allows us to study increased protein activity related to over-expression of genes as well (*Béal et al., 2019*; *Montagud et al., 2019*). Interestingly, we found several genes that were found as suitable points of intervention in most of the patients (MYC_MAX complex and SPOP were identified in more than 80% of the cases) (Appendix 1, Section 4.2, *Appendix 1—figures 19 and 20*), but others were specific to only some of the patients (MXI1 was identified in only 4 patients, 1% of the total, GLI in only 7% and WNT in 8% of patients). All the TCGA-specific personalised models can be found in *Supplementary file 3*, and the TCGA mutants and their phenotype scores can be found in *Supplementary file 4*.

**Table 1.** List of selected nodes, their corresponding genes and drugs that were included in the drug analysis of the models tailored for TCGA patients and LNCaP cell line.

| Node | Gene | Compound / Inhibitor name | Clinical stage | Source |
|---|---|---|---|---|
| | | PI-103 | Preclinical | Drug Bank |
| | | Enzastaurin | Phase 3 | Drug Bank |
| AKT | AKT1, AKT2, AKT3 | Archexin, Pictilisib | Phase 2 | Drug Bank |
| | | Abiraterone, Enzalutamide, Formestane, Testosterone propionate | Approved | Drug Bank |
| AR | AR | 5alpha-androstan-3beta-ol | Preclinical | Drug Bank |
| Caspase8 | CASP8 | Bardoxolone | Preclinical | Drug Bank |
| cFLAR | CFLAR | - | - | - |
| | | Afatinib, Osimertinib, Neratinib, Erlotinib, Gefitinib | Approved | Drug Bank |
| EGFR | EGFR | Varlitinib | Phase 3 | Drug Bank |
| | | Olmutinib, Pelitinib | Phase 2 | Drug Bank |
| | | Isoprenaline | Approved | Drug Bank |
| | | Perifosine | Phase 3 | Drug Bank |
| | MAPK1 | Turpentine, SB220025, Olomoucine, Phosphonothreonine | Preclinical | Drug Bank |
| | | Arsenic trioxide | Approved | Drug Bank |
| | | Ulixertinib, Seliciclib | Phase 2 | Drug Bank |
| | MAPK3, MAPK1 | Purvalanol | Preclinical | Drug Bank |
| | | Sulindac, Cholecystokinin | Approved | Drug Bank |
| ERK | MAPK3 | 5-iodotubercidin | Preclinical | Drug Bank |
| GLUT1 | SLC2A1 | Resveratrol | Phase 4 | Drug Bank |
| HIF-1 | HIF1A | CAY-10585 | Preclinical | Drug Bank |
| | | Cladribine | Approved | Drug Bank |
| HSPs | HSP90AA1, HSP90AB1, HSP90B1, HSPA1A, HSPA1B, HSPB1 | 17-DMAG | Phase 2 | Drug Bank |
| | | NMS-E973 | Preclinical | Drug Bank |
| | | Trametinib, Selumetinib | Approved | Drug Bank |
| | | Perifosine | Phase 3 | Drug Bank |
| MEK1_2 | MAP2K1, MAP2K2 | PD184352 (CI-1040) | Phase 2 | Drug Bank |
| MYC_MAX | complex of MYC and MAX | 10058-F4 (for MAX) | Preclinical | Drug Bank |
| p14ARF | CDKN2A | - | - | - |
| PI3K | PIK3CA, PIK3CB, PIK3CG, PIK3CD, PIK3R1, PIK3R2, PIK3R3, PIK3R4, PIK3R5, PIK3R6, PIK3C2A, PIK3C2B, PIK3C2G, PIK3C3 | PI-103 | Preclinical | Drug Bank |
| | | Pictilisib | Phase 2 | Drug Bank |
| | NOX1, NOX3, NOX4 | Fostamatinib | Approved | Drug Bank |
| | | Dextromethorphan | Approved | Drug Bank |
| ROS | NOX2 | Tetrahydroisoquinolines (CHEMBL3733336, CHEMBL3347550, CHEMBL3347551) | Preclinical | ChEMBL |
| SPOP | SPOP | - | - | - |
| | | Grn163l | Phase 2 | Drug Bank |
| TERT | TERT | BIBR 1532 | Preclinical | ChEMBL |

Furthermore, we explored the possibility of finding combinations of treatments that could reduce the *Proliferation* phenotype or increase the *Apoptosis* one. To lower the computational power need, we narrowed down the list of potential candidates to a set of selected genes that are targets of already-developed drugs relevant in cancer progression (*Table 1*) and analysed the simulations of the models with all the single and combined perturbations.

We used the models to grade the effect that the combined treatments have in each one of the 488 TCGA patient-specific models' phenotypes. This list of combinations of treatments can be used to compare the effects of drugs on each TCGA patient and allows us to propose some of them for individual patients and to suggest drugs suitable to groups of patients (*Supplementary file 4*). Indeed, the inactivation of some of the targeted genes had a greater effect in some patients than in others, suggesting the possibility for the design of personalised drug treatments. For instance, for the TCGA-EJ-5527 patient, the use of MYC_MAX complex inhibitor reduced *Proliferation* to 66%. For this patient, combining MYC_MAX with other inhibitors, such as AR or AKT, did not further reduce the *Proliferation* score (67% in these cases). Other patients have MYC_MAX as an interesting drug target, but the inhibition of this complex did not have such a dramatic effect on their *Proliferation* scores as in the case of TCGA-EJ-5527. Likewise, for the TCGA-H9-A6BX patient, the use of SPOP inhibitor increased *Apoptosis* by 87%, while the use of a combination of cFLAR and SPOP inhibitors further increased *Apoptosis* by 89%. For the rest of this section, we focus on the analysis of clinical groups rather than individuals.

Studying the decrease of *Proliferation*, we found that AKT is the top hit in Gleason Grades 1, 2, 3, and 4, seconded by EGFR and SPOP in Grade 1, by SPOP and PIP3 in Grade 2, by PIP3 and AR in Grade 3, and by CyclinD and MYC_MAX in Grade 4. MYC_MAX is the top hit in Grade 5, seconded by AR (Appendix 1, Section 4.2, *Appendix 1—figure 19*). In regard to the increase of *Apoptosis*, SPOP is the top hit in all grades, seconded by SSH in Grades 1, 2, and 3 and by AKT in Grade 4 (Appendix 1, Section 4.2, *Appendix 1—figure 20*). It is interesting to note here that many of these genes are targeted by drugs (*Table 1*). Notably, AR is the target of the drug Enzalutamide, which is indicated for men with an advanced stage of the disease (*Scott, 2018*), or that MYC is the target of BET bromo-domain inhibitors and are generally effective in castration-resistant prostate cancer cases (*Coleman et al., 2019*).

The work on patient data provided possible insights and suggested patient- and grade-specific potential targets. To validate our approach experimentally, we personalised the prostate model to different prostate cell lines, where we performed drug assays to confirm the predictions of the model.

## Personalised drug predictions of LNCaP Boolean model

We applied the methodology for personalisation of the prostate model to eight prostate cell lines available in GDSC (*Iorio et al., 2016*): 22RV1, BPH-1, DU-145, NCI-H660, PC-3, PWR-1E, and VCaP (results in Appendix 1, Section 5 and are publicly available in *Supplementary file 5*). We decided to focus the validation on one cell line, LNCaP.

LNCaP, first isolated from a human metastatic prostate adenocarcinoma found in a lymph node (*Horoszewicz et al., 1983*), is one of the most widely used cell lines for prostate cancer studies. Androgen-sensitive LNCaP cells are representative of patients sensitive to treatments as opposed to resistant cell lines such as DU-145. Additionally, LNCaP cells have been used to obtain numerous subsequent derivatives with different characteristics (*Cunningham and You, 2015*).

The LNCaP personalisation was performed based on mutations as discrete data and RNA-Seq as continuous data. The resulting LNCaP-specific Boolean model was then used to identify all possible combinations of mutations (interpreted as effects of therapies) and to study the synergy of these perturbations. For that purpose, we automatically performed single and double mutant analyses on the LNCaP-specific model (knock-out and overexpression) (*Montagud et al., 2019*) and focused on the model phenotype probabilities as read-outs of the simulations. The analysis of the complete set of simulations for the 32,258 mutants can be found in the Appendix 1, Section 6.1 and in *Supplementary file 6*, where the LNCaP cell line-specific mutants and their phenotype scores are reported for all mutants. Among all combinations, we identified the top 20 knock-out mutations that depleted *Proliferation* or increased *Apoptosis* the most. As some of them overlapped, we ended up with 29 nodes: *AKT, AR, ATR, AXIN1, Bak, BIRC5, CDH2, cFLAR, CyclinB, CyclinD, E2F1, eEF2K, eEF2, eEF2K, EGFR, ERK, HSPs, MED12, mTORC1, mTORC2, MYC, MYC_MAX, PHDs, PI3K, PIP3, SPOP, TAK1, TWIST1,*

*and VHL.* We used the scores of these nodes to further trim down the list to have 10 final nodes (*AKT, AR, cFLAR, EGFR, ERK, HSPs, MYC_MAX, SPOP,* and *PI3K*) and added seven other nodes whose genes are considered relevant in cancer biology, such as *AR_ERG* fusion, *Caspase8, HIF1, GLUT1, MEK1_2, p14ARF, ROS,* and *TERT* (*Table 1*). We did not consider the overexpression mutants as they have a very difficult translation to drug uses and clinical practices.

To further analyse the mutant effects, we simulated the LNCaP model with increasing node inhibition values to mimic the effect of drugs' dosages using a methodology we specifically developed for this purpose (PROFILE_v2 and available at https://github.com/ArnauMontagud/PROFILE_v2; *Montagud, 2022a*). Six simulations were done for each inhibited node, with 100% of node inhibition (proper knock-out), 80%, 60%, 40%, 20% and 0% (no inhibition) (see Materials and methods). A nutrient-rich media with EGF was used for these simulations that correspond to experimental conditions that are tested here. We show results on three additional sets of initial conditions in the Appendix 1, Section 6, *Appendix 1—figure 27*: a nutrient-rich media with androgen, with androgen and EGF, and with none, . We applied this gradual inhibition, using increasing drugs' concentrations, to a reduced list of drug-targeted genes relevant for cancer progression (*Table 1*). We confirmed that the inhibition of different nodes affected differently the probabilities of the outputs (Appendix 1, Section 7.3.1, *Appendix 1—figures 34 and 35*). Notably, the *Apoptosis* score was slightly promoted when knocking out *SPOP* under all growth conditions (Appendix 1, Section 7.3.1, *Appendix 1—figure 35*). Likewise, *Proliferation* depletion was accomplished when *HSPs* or *MYC_MAX* were inhibited under all conditions and, less notably, when *ERK, EGFR, SPOP,* or *PI3K* were inhibited (Appendix 1, Section 7.3.1, *Appendix 1—figure 35*).

Additionally, these gradual inhibition analyses can be combined to study the interaction of two simultaneously inhibiting nodes (Appendix 1, Section 7.3.2, *Appendix 1—figures 36 and 37*). For instance, the combined gradual inhibition of *ERK* and *MYC_MAX* nodes affects the *Proliferation* score in a balanced manner (*Figure 5A*) even though *MYC_MAX* seems to affect this phenotype more, notably at low activity levels. By extracting subnetworks of interaction around *ERK* and *MYC_MAX* and comparing them, we found that the pathways they belong to have complementary downstream targets participating in cell proliferation through targets in MAPK and cell cycle pathways. This complementarity could explain the synergistic effects observed (*Figure 5A and C*).

Lastly, drug synergies can be studied using Bliss Independence using the results from single and combined simulations with gradual inhibitions. This score compares the combined effect of two drugs with the effect of each one of them, with a synergy when the value of this score is lower than 1. We found that the combined inhibition of *ERK* and *MYC_MAX* nodes on the *Proliferation* score was synergistic (*Figure 5C*). Another synergistic pair is the combined gradual inhibition of *HSPs* and *PI3K* nodes that also affects the *Proliferation* score in a joint manner (*Figure 5B*), with some Bliss Independence synergy found (*Figure 5D*). A complete study on the Bliss Independence synergy of all the drugs considered in the present work on *Proliferation* and *Apoptosis* phenotypes can be found in Appendix 1, Section 7.3.2, *Appendix 1—figures 38 and 39*.

## Experimental validation of predicted targets

### Drugs associated with the proposed targets

To identify drugs that could act as potential inhibitors of the genes identified with the Boolean model, we explored the drug-target associations in DrugBank (*Wishart et al., 2018*) and ChEMBL (*Gaulton et al., 2017*). We found drugs that targeted almost all genes corresponding to the nodes of interest in *Table 1*, except for cFLAR, p14ARF, and SPOP. However, we could not identify experimental cases where drugs targeting both members of the proposed combinations were available (Appendix 1, Section 7.1 and in *Supplementary file 6*). One possible explanation is that the combinations predicted by the model suggest, in some cases, to overexpress the potential target and most of the drugs available act as inhibitors of their targets.

Using the cell line-specific models, we tested if the LNCaP cell line was more sensitive than the rest of the prostate cell lines to the LNCaP-specific drugs identified in *Table 1*. We compared GDSC's Z-score of these drugs in LNCaP with their Z-scores in all GDSC cell lines (*Figure 6* and Appendix 1, Section 7.2, *Appendix 1—figure 33*). We observed that LNCaP is more sensitive to drugs targeting AKT or TERT than the rest of the studied prostate cell lines. Furthermore, we saw that the drugs that targeted the genes included in the model allowed the identification of cell line specificities (Appendix

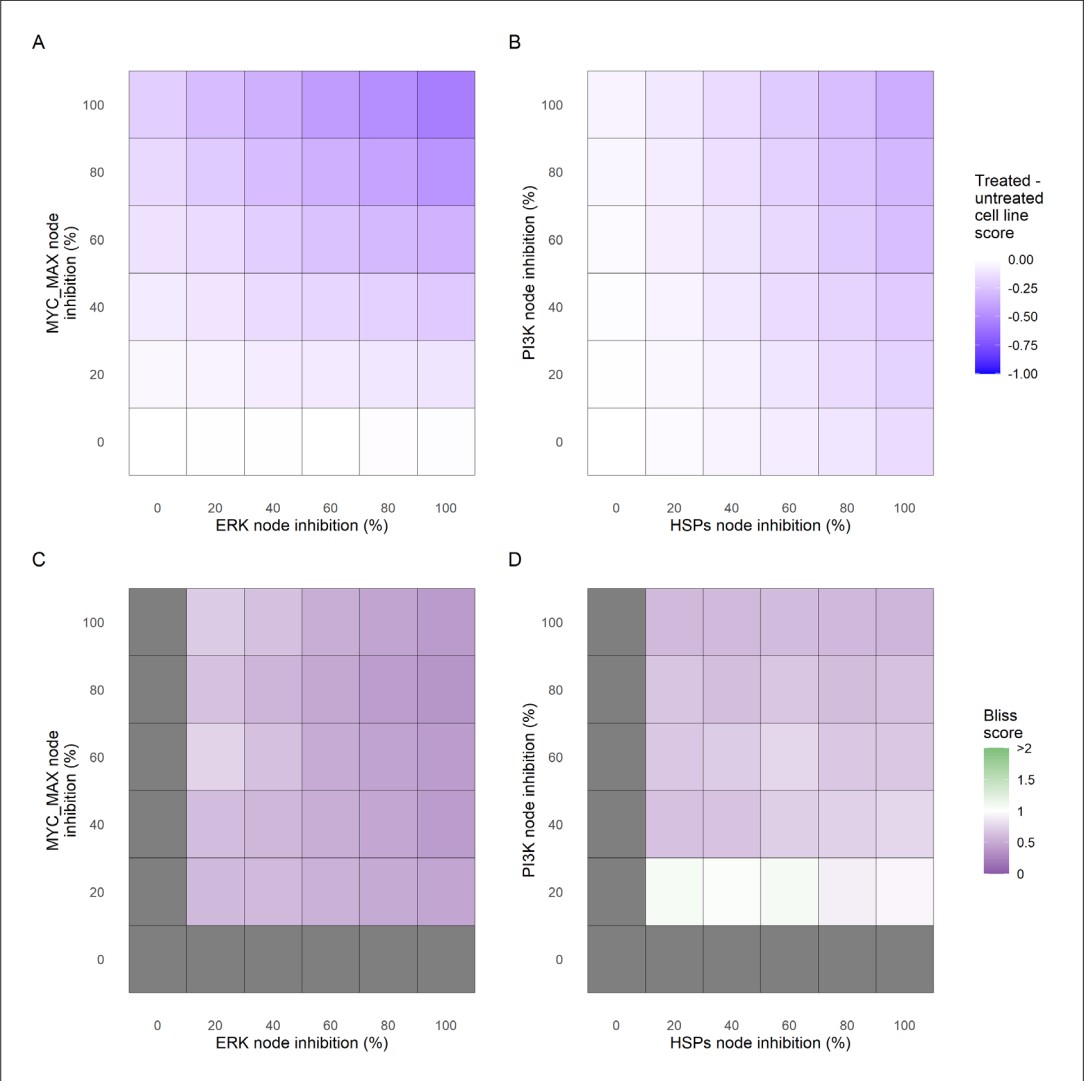

**Figure 5.** Phenotype score variations and synergy upon combined ERK and MYC_MAX (**A and C**) and HSPs and PI3K (**B and D**) inhibition under *EGF* growth condition. Proliferation score variation (**A**) and Bliss Independence synergy score (**C**) with increased node activation of nodes ERK and MYC_MAX. Proliferation score variation (**B**) and Bliss Independence synergy score (**D**) with increased node activation of nodes HSPs and PI3K. Bliss Independence synergy score <1 is characteristic of drug synergy, grey colour means one of the drugs is absent and thus no synergy score is available.

The online version of this article includes the following source data for figure 5:

**Source code 1.** R code needed to perform the drug dosage experiments and obtain *Figure 5* from the main text and *Appendix 1—figures 27, 34–39*.

**Source data 1.** Processed datasets needed to obtain the phenotype score variations and synergy values of *Figure 5* with *Figure 5—source code 1*.

1, Section 7.1). For instance, target enrichment analysis showed that LNCaP cell lines are especially sensitive to drugs targeting PI3K/AKT/mTOR, hormone-related (AR targeting) and Chromatin (bromo-domain inhibitors, regulating Myc) pathways (adjusted p-values from target enrichment: 0.001, 0.001, and 0.032, respectively, Appendix 1, Section 7.1, *Appendix 1—table 2*), which corresponds to the model predictions (*Table 1*). Also, the LNCaP cell line is more sensitive to drugs targeting model-identified nodes than to drugs targeting other proteins (Appendix 1, Section 7.1, *Appendix 1—figure 32*, Mann-Whitney U p-value 0.00041), and this effect is specific for LNCaP cell line (Mann-Whitney U p-values ranging from 0.0033 to 0.38 for other prostate cancer cell lines).

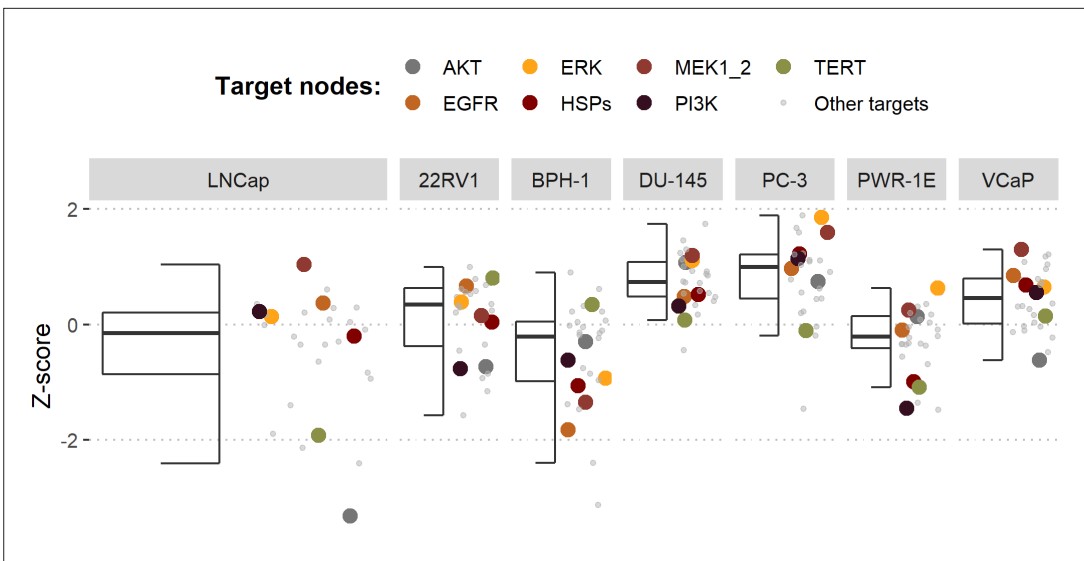

**Figure 6.** Model-targeting drugs' sensitivities across prostate cell lines. GDSC z-score was obtained for all the drugs targeting genes included in the model for all the prostate cell lines in GDSC. Negative values mean that the cell line is more sensitive to the drug. Drugs included in *Table 1* were highlighted. 'Other targets' are drugs targeting model-related genes that are not part of *Table 1*.

The online version of this article includes the following source data for figure 6:

**Source code 1.** R code needed to obtain *Figure 6*.

**Source data 1.** Processed dataset needed to obtain *Figure 6* with *Figure 6—source code 1*.

**Source data 2.** Processed dataset needed to obtain *Figure 6* with *Figure 6—source code 1*.

Overall, the drugs proposed through this analysis suggest the possibility to repurpose drugs that are used in treating other forms of cancer for prostate cancer and open the avenue for further experimental validations based on these suggestions.

## Experimental validation of drugs in LNCaP

To validate the model predictions of the candidate drugs, we selected four drugs that target HSPs and PI3K and tested them in LNCaP cell line experiments by using endpoint cell viability measurement assays and real-time cell survival assays using the xCELLigence system (see Materials and methods). The drug selection was a compromise between the drugs identified by our analyses (*Table 1*) and their effect in diminishing LNCaP's proliferation (see the previous section). In both assays, drugs that target HSP90AA1 and PI3K/AKT pathway genes retrieved from the model analyses were found to be effective against cell proliferation.

The Hsp90 chaperone is expressed abundantly and plays a crucial role in the correct folding of a wide variety of proteins such as protein kinases and steroid hormone receptors (*Schopf et al., 2017*). Hsp90 can act as a protector of less stable proteins produced by DNA mutations in cancer cells (*Barrott and Haystead, 2013*; *Hessenkemper and Baniahmad, 2013*). Currently, Hsp90 inhibitors are in clinical trials for multiple indications in cancer (*Chen et al., 2020*; *Iwai et al., 2012*; *Le et al., 2017*). The PI3K/AKT signalling pathway controls many different cellular processes such as cell growth, motility, proliferation, and apoptosis and is frequently altered in different cancer cells (*Carceles-Cordon et al., 2020*; *Shorning et al., 2020*). Many PI3K/AKT inhibitors are in different stages of clinical development, and some of them are approved for clinical use (*Table 1*).

Notably, Hsp90 (NMS-E973,17-DMAG) and PI3K/AKT pathway (PI-103, Pictilisib) inhibitors showed a dose-dependent activity in the endpoint cell viability assay determined by the fluorescent resazurin after a 48 hr incubation (*Figure 7*). This dose-dependent activity is more notable in Hsp90 drugs (NMS-E973,17-DMAG) than in PI3K/AKT pathway (Pictilisib) ones and very modest for PI-103.

We studied the real-time response of LNCaP cell viability upon drug addition and saw that the LNCaP cell line is sensitive to Hsp90 and PI3K/AKT pathway inhibitors (*Figures 8 and 9*, respectively).

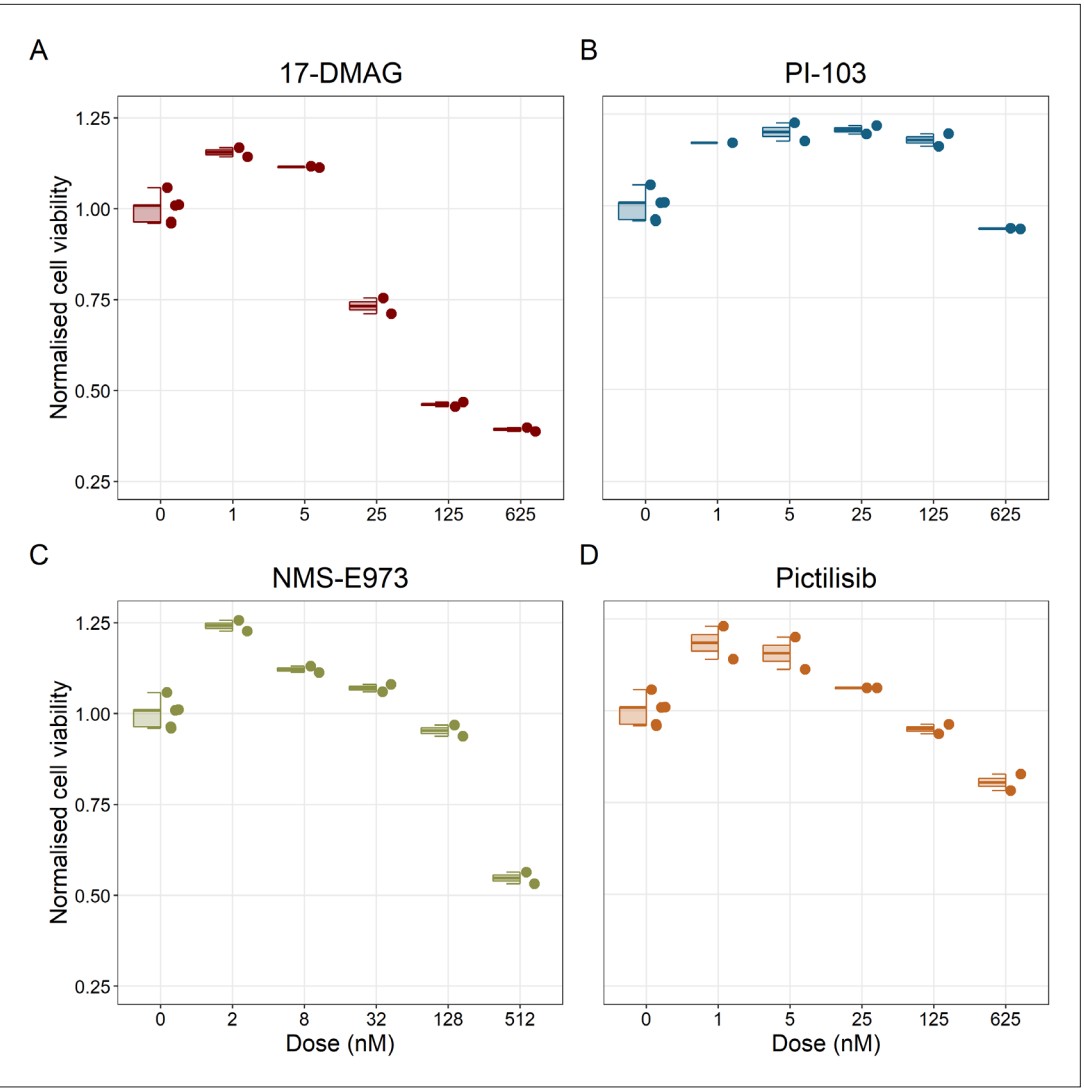

**Figure 7.** Cell viability assay determined by the fluorescent resazurin after a 48 hours incubation showed a dose-dependent response to different inhibitors. (**A**) Cell viability assay of LNCaP cell line response to 17-DMAG HSP90 inhibitor. (**B**) Cell viability assay of LNCaP cell line response to PI-103 PI3K/AKT pathway inhibitor. (**C**) Cell viability assay of LNCaP cell line response to NMS-E973 HSP90 inhibitor. (**D**) Cell viability assay of LNCaP cell line response to Pictilisib PI3K/AKT pathway inhibitor. Concentrations of drugs were selected to capture their drug-dose response curves. The concentrations for the NMS-E973 are different from the rest as this drug is more potent than the rest (see Materials and methods).

The online version of this article includes the following source data for figure 7:

**Source code 1.** R code needed to obtain *Figure 7*.

**Source data 1.** Processed dataset needed to obtain *Figure 7* with *Figure 7—source code 1*.

**Source data 2.** Processed dataset needed to obtain with *Figure 7—source code 1*.

Both Hsp90 inhibitors tested, 17-DMAG and NMS-E973, reduced the cell viability 12 hr after drug supplementation (*Figure 8A* for 17-DMAG and *Figure 8B* for NMS-E973), with 17-DMAG having a stronger effect and in a more clear concentration-dependent manner than NMS-E973 (Appendix 1, Section 8, *Appendix 1—figure 40*, panels B-D for 17-DMAG and panels F-H for NMS-E973).

Likewise, both PI3K/AKT pathway inhibitors tested, Pictilisib and PI-103, reduced the cell viability immediately after drug supplementation (*Figure 9A* for Pictilisib and *Figure 9B* for PI-103), in a concentration-dependent manner (Appendix 1, Section 8, *Appendix 1—figure 41B-D*, for Pictilisib

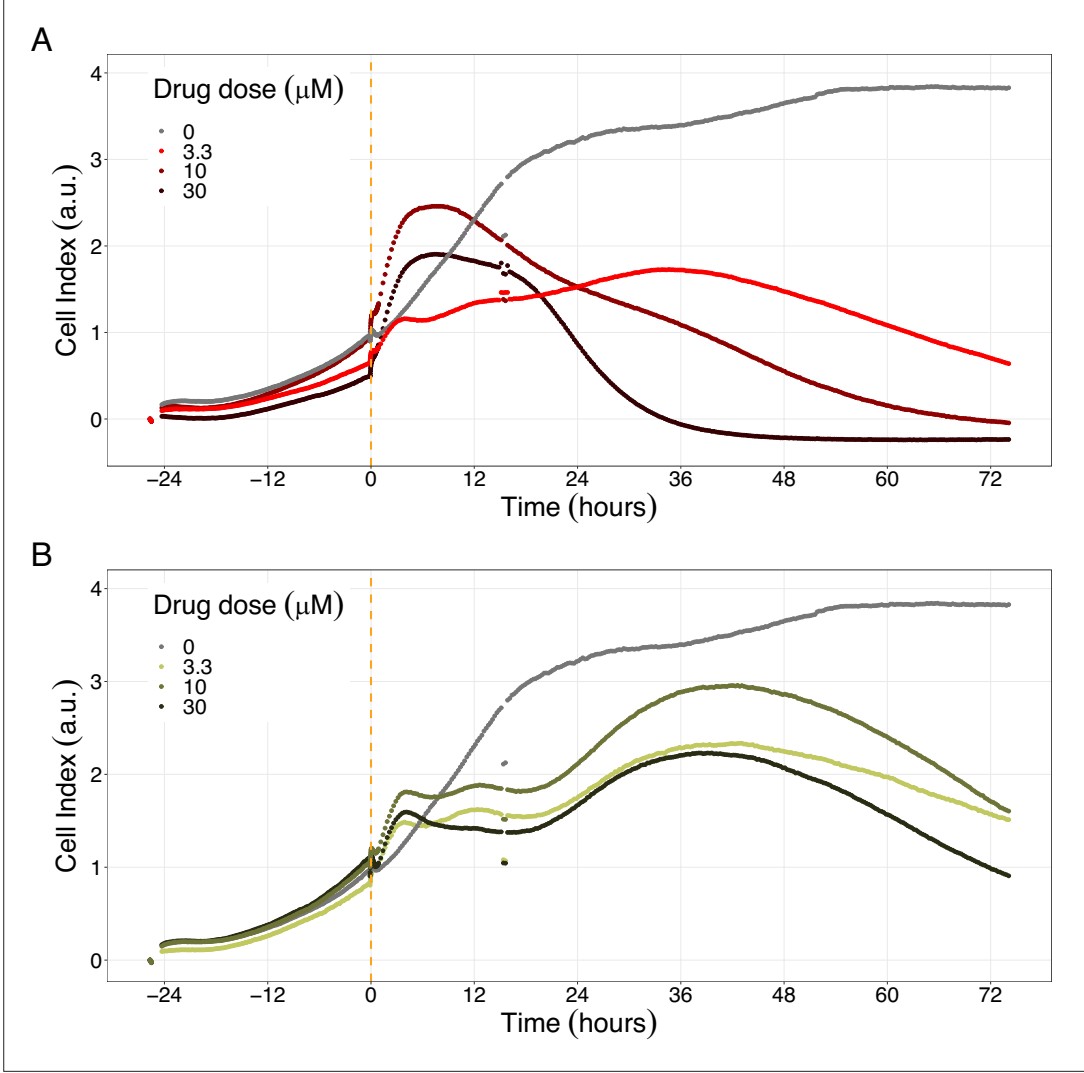

**Figure 8.** Hsp90 inhibitors resulted in dose-dependent changes in the LNCaP cell line growth. (**A**) Real-time cell electronic sensing (RT-CES) cytotoxicity assay of Hsp90 inhibitor, 17-DMAG, that uses the Cell Index as a measurement of the cell growth rate (see the Materials and methods section). The yellow dotted line represents the 17-DMAG addition. (**B**) RT-CES cytotoxicity assay of Hsp90 inhibitor, NMS-E973. The yellow dotted line represents the NMS-E973 addition.

The online version of this article includes the following source data for figure 8:

**Source data 1.** Processed dataset to obtain *Figures 8 and 9* with *Figure 8—source code 1*.

**Source code 1.** R code needed to obtain *Figures 8 and 9* with *Figure 8—source data 1*.

and panels F-H for PI-103). In addition, Hsp90 inhibitors had a more prolonged effect on the cells' proliferation than PI3K/AKT pathway inhibitors.

## Discussion

Clinical assessment of cancers is moving toward more precise, personalised treatments, as the times of one-size-fits-all treatments are no longer appropriate, and patient-tailored models could boost the success rate of these treatments in clinical practice. In this study, we set out to develop a methodology to investigate drug treatments using personalised Boolean models. Our approach consists of building a model that represents the patient-specific disease status and retrieving a list of proposed interventions that affect this disease status, notably by reducing its pro-cancerous behaviours. In this work, we have showcased this methodology by applying it to TCGA prostate cancer patients and to

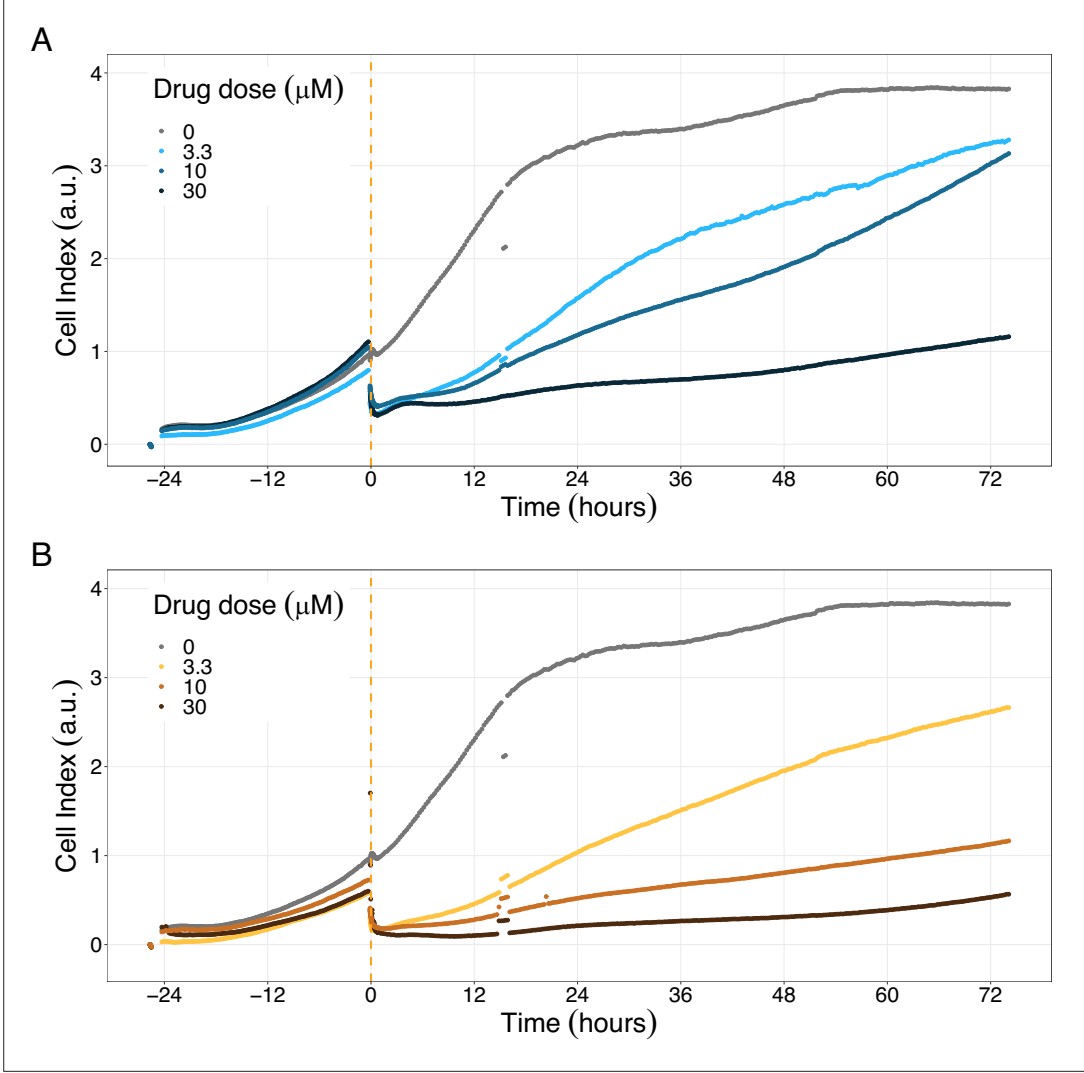

**Figure 9.** PI3K/AKT pathway inhibition with different PI3K/AKT inhibitors shows the dose-dependent response in LNCaP cell line growth. (**A**) Real-time cell electronic sensing (RT-CES) cytotoxicity assay of PI3K/AKT pathway inhibitor, PI-103, that uses the Cell Index as a measurement of the cell growth rate (see the Materials and methods section). The yellow dotted line represents the PI-103 addition. (**B**) RT-CES cytotoxicity assay of PI3K/AKT pathway inhibitor, Pictilisib. The yellow dotted line represents the Pictilisib addition.

GDSC prostate cancer cell lines, finding patient- and cell line-specific targets and validating selected cell line-specific predicted targets (*Figure 1*).

First, a prostate cancer Boolean model that encompasses relevant signalling pathways in cancer was constructed based on already published models, experimental data analyses and pathway databases (*Figure 2*). The influence network and the assignment of logical rules for each node of this network were obtained from known interactions described in the literature (*Figure 3*). This model describes the regulation of invasion, migration, cell cycle, apoptosis, androgen, and growth factors signalling in prostate cancer (Appendix 1, Section 1).

Second, from this generic Boolean model, we constructed personalised models using the different datasets, that is 488 patients from TCGA and eight cell lines from GDSC. We obtained Gleason score-specific behaviours for TCGA's patients when studying their *Proliferation* and *Apoptosis* scores, observing that high *Proliferation* scores are higher in high Gleason grades (*Figure 4*). Thus, the use of these personalised models can help rationalise the relationship of Gleason grading with some of these phenotypes.

Likewise, GDSC data was used with the prostate model to obtain cell line-specific prostate models (*Figure 6*). These models show differential behaviours, notably in terms of *Invasion* and *Proliferation* phenotypes (Appendix 1, Section 5, *Appendix 1—figure 21*). One of these cell line-specific models, LNCaP, was chosen, and the effects of all its genetic perturbations were thoroughly studied. We studied 32,258 mutants, including single and double mutants, knock-out and over-expressed, and their phenotypes (Appendix 1, Section 6.1, *Appendix 1—figures 28 and 29*). Thirty-two knock-out perturbations that depleted *Proliferation* and/or increased *Apoptosis* were identified, and 16 of them were selected for further analyses (*Table 1*). The LNCaP-specific model was simulated using different initial conditions that capture different growth media's specificities, such as RPMI media with and without androgen or epidermal growth factor (Appendix 1, Section 6, *Appendix 1—figure 27*).

Third, these personalised models were used to simulate the inhibition of druggable genes and proteins, uncovering new treatment's combination and their synergies. We developed a methodology to simulate drug inhibitions in Boolean models, termed PROFILE_v2, as an extension of previous works (*Béal et al., 2019*). The LNCaP-specific model was used to obtain simulations with nodes and pairs of nodes corresponding to the genes of interest inhibited with varying strengths. This study allowed us to compile a list of potential targets (*Table 1*) and to identify potential synergies among genes in the model (*Figure 5*). Some of the drugs that targeted these genes, such as AKT and TERT, were identified in GDSC as having more sensitivity in LNCaP than in the rest of the prostate cancer cell lines (*Figure 6*). In addition, drugs that targeted genes included in the model allowed the identification of cell line specificities (Appendix 1, Section 5).

Fourth, we validated the effect of Hsp90 and PI3K/AKT pathway inhibitors on the LNCaP cell line experimentally, finding a concentration-dependent inhibition of the cell line viability as predicted, confirming the role of the drugs targeting these proteins in reducing LNCaP's proliferation (*Figures 7 and 8*). Notably, these targets have been studied in other works on prostate cancer (*Chen et al., 2020*; *Le et al., 2017*).

The study presented here enables the study of drug combinations and their synergies. One reason for searching for combinations of drugs is that these have been described for allowing the use of lower doses of each of the two drugs reducing their toxicity (*Bayat Mokhtari et al., 2017*), evading compensatory mechanisms and combating drug resistances (*Al-Lazikani et al., 2012*; *Krzyszczyk et al., 2018*).

Even if this approach is attractive and promising, it has some limitations. The scope of present work is to test this methodology on a prostate model and infer patient-specific prostate cancer treatments. The method need to be adapted if it were to be expanded to study other cancers by using other models and target lists. The analyses performed with the mathematical model do not aim to predict drug dosages per se but to help in the identification of potential candidates. The patient-specific changes in *Proliferation* and *Apoptosis* scores upon mutation are maximal theoretical yields that are used to rank the different potential treatments and should not be used as a direct target for experimental results or clinical trials. Our methodology suggests treatments for individual patients, but the obtained results vary greatly from patient to patient, which is not an uncommon issue of personalised medicine (*Ciccarese et al., 2017*; *Molinari et al., 2018*). This variability is an economic challenge for labs and companies to pursue true patient-specific treatments and also poses challenges in clinical trial designs aimed at validating the model based on the selection of treatments (*Cunanan et al., 2017*). Nowadays, and because of these constraints, it might be more commercially interesting to target group-specific treatments, which can be more easily related to clinical stages of the disease.

Mathematical modelling of patient profiles helps to classify them in groups with differential characteristics, providing, in essence, a grade-specific treatment. We, therefore, based our analysis on clinical grouping defined by the Gleason grades, but some works have emphasised the difficulty to properly assess them (*Chen and Zhou, 2016*) and, as a result, may not be the perfect predictor for the patient subgrouping in this analysis, even though it is the only available one for these datasets. The lack of subgrouping that stratifies patients adequately may undermine the analysis of our results and could explain the *Proliferation* and *Apoptosis* scores of high-grade and low-grade Gleason patients.

Moreover, the behaviours observed in the simulations of the cell line-specific models do not always correspond to what is reported in the literature. The differences between simulation results and biological characteristics could be addressed in further studies by including other pathways, for example, better describing the DNA repair mechanisms, or by tailoring the model with different

sets of data, as the data used to personalise these models do not allow for clustering these cell lines according to their different characteristics (Appendix 1, Section 5, *Appendix 1—figures 24 and 25*). In this sense, another limitation is that we use static data or a snapshot of dynamic data to build dynamic models and to study its stochastic results. Thus, these personalised models would likely improve their performance if they were fitted to dynamic data (*Saez-Rodriguez and Blüthgen, 2020*) or quantitative versions of the models were built, such as ODE-based, that may capture more fine differences among cell lines. As perspectives, we are working on integrating these models in multiscale models to study the effect of the tumour microenvironment (*Ponce-de-Leon et al., 2021*; *Ponce-de-Leon et al., 2022*), on including information to simulate multiple reagents targeting a single node of the model, on scaling these multiscale models to exascale high-performance computing clusters (*Montagud et al., 2021*; *Saxena et al., 2021*), and on streamlining these studies using workflows in computing clusters to fasten the processing of new, bigger cohorts, as in the PerMedCoE project (https://permedcoe.eu/).

The present work contributes to efforts aimed at using modelling (*Eduati et al., 2020*; *Rivas-Barragan et al., 2020*; *Gómez Tejeda Zañudo et al., 2017*) and other computational methods (*Madani Tonekaboni et al., 2018*; *Menden et al., 2019*) for the discovery of novel drug targets and combinatorial strategies. Our study expands the prostate drug catalogue and improves predictions of the impact of these in clinical strategies for prostate cancer by proposing and grading the effectiveness of a set of drugs that could be used off-label or repurposed. The insights gained from this study present the potential of using personalised models to obtain precise, personalised drug treatments for cancer patients.

## Materials and methods

### Data acquisition

Publicly available data of 489 human prostate cancer patients from TCGA described in *Hoadley et al., 2018* were used in the present work. We gathered mutations, CNA, RNA and clinical data from cBioPortal (https://www.cbioportal.org/study/summary?id=prad_tcga_pan_can_atlas_2018) for all of these samples resulting in 488 with complete omics datasets.

Publicly available data of cell lines used in the present work were obtained from the Genomics of Drug Sensitivity in Cancer database (GDSC) (*Iorio et al., 2016*). Mutations, CNA and RNA data, as well as cell lines descriptors, were downloaded from (https://www.cancerrxgene.org/downloads). In this work, we have used 3- and 5-stage Gleason grades. Their correspondence is the following: GG Low is GG 1, GG Intermediate is GG 2 and 3, and GG High is GG 4 and 5.

All these data were used to personalise Boolean models using our PROFILE method (*Béal et al., 2019*).

### Prior knowledge network construction

Several sources were used in building this prostate Boolean model and, in particular, the model published by *Fumiã and Martins, 2013*. This model includes several signalling pathways such as the ones involving receptor tyrosine kinase (RTKs), phosphatidylinositol 3-kinase (PI3K)/AKT, WNT/β-Catenin, transforming growth factor-β (TGF-β)/Smads, cyclins, retinoblastoma protein (Rb), hypoxia-inducible transcription factor (HIF-1), p53 and ataxia-telangiectasia mutated (ATM)/ataxia-telangiectasia and Rad3-related (ATR) protein kinases. The model includes these pathways as well as the substantial cross-talks among them. For a complete description of the process of construction, see Appendix 1, Section 1.

The model also includes several pathways that have a relevant role in our datasets identified by ROMA (*Martignetti et al., 2016*), a software that uses the first principal component of a PCA analysis to summarise the coexpression of a group of genes in the gene set, identifying significantly overdispersed pathways with a relevant role in a given set of samples. This software was applied to the TCGA transcriptomics data using the gene sets described in the Atlas of Cancer Signaling Networks, ACSN (*Kuperstein et al., 2015*) (http://www.acsn.curie.fr/) and in the Hallmarks (*Liberzon et al., 2015*) (Appendix 1, Section 1.1.3, *Appendix 1—figure 1*) and highlighted the signalling pathways that show high variance across all samples, suggesting candidate pathways and genes. Additionally, OmniPath (*Türei et al., 2021*) was used to extend the model and complete it, connecting the nodes from Fumiã

and Martins and the ones from ROMA analysis. OmniPath is a comprehensive collection of literature-curated human signalling pathways, which includes several databases such as Signor (*Perfetto et al., 2016*) or Reactome (*Fabregat et al., 2016*) and that can be queried using pypath, a Python module for molecular networks and pathways analyses.

Fusion genes are frequently found in human prostate cancer and have been identified as a specific subtype marker (*Cancer Genome Atlas Research Network, 2015*). The most frequent is TMPRSS2:ERG, as it involves the transcription factor ERG, which leads to cell-cycle progression. ERG fuses with the AR-regulated TMPRSS2 gene promoter to form an oncogenic fusion gene that is especially common in hormone-refractory prostate cancer, conferring androgen responsiveness to ERG. A literature search reveals that ERG directly regulates EZH2, oncogene c-Myc and many other targets in prostate cancer (*Kunderfranco et al., 2010*).

We modelled the gene fusion with activation of ERG by the decoupling of ERG in a special node *AR_ERG* that is only activated by the *AR* when the *fused_event* input node is active. In the healthy case, *fused_event* (that represents TMPRSS2:ERG fusion event) is fixed to 0 or inactive. The occurrence of the gene fusion is represented with the model perturbation where *fused_event* is fixed to 1. This *AR_ERG* node is further controlled by tumour suppressor NKX3-1 that accelerates *DNA_repair* response, and avoids the gene fusion TMPRSS2:ERG. Thus, loss of NKX3-1 favours recruitment to the ERG gene breakpoint of proteins that promote error-prone non-homologous end-joining (*Bowen et al., 2015*).

The network was further documented using up-to-date literature and was constructed using GINsim (*Chaouiya et al., 2012*), which allowed us to study its stable states and network properties.

## Boolean model construction

We converted the network to a Boolean model by defining a regulatory graph, where each node is associated with discrete levels of activity (0 or 1). Each edge represents a regulatory interaction between the source and target nodes and is labelled with a threshold and a sign (positive or negative). The model is completed by logical rules (or functions), which assign a target value to each node for each regulator level combination (*Abou-Jaoudé et al., 2016*; *Chaouiya et al., 2012*). The regulatory graph was constructed using GINsim software (*Chaouiya et al., 2012*) and then exported in a format readable by MaBoSS software (see below) in order to perform stochastic simulations on the Boolean model.

The final model has a total of 133 nodes and 449 edges (*Supplementary file 1*) and includes pathways such as androgen receptor and growth factor signalling, several signalling pathways (Wnt, NFkB, PI3K/AKT, MAPK, mTOR, SHH), cell cycle, epithelial-mesenchymal transition (EMT), Apoptosis, DNA damage, etc. This model has nine inputs (*EGF, FGF, TGF beta, Nutrients, Hypoxia, Acidosis, Androgen, TNF alpha,* and *Carcinogen* presence) and six outputs (*Proliferation, Apoptosis, Invasion, Migration,* (bone) *Metastasis,* and *DNA repair*). Note that a node in the network can represent complexes or families of proteins (e.g. AMPK represents the genes PRKAA1, PRKAA2, PRKAB1, PRKAB2, PRKAG1, PRKAG2, PRKAG3). The correspondence can be found in "Montagud2022_interactions_sources.xlsx" and "Montagud2022_nodes_in_pathways.xlsx" in *Supplementary file 1*.

This model was deposited in the GINsim Database with identifier 252 (http://ginsim.org/model/signalling-prostate-cancer) and in BioModels (*Malik-Sheriff et al., 2020*) with identifier MODEL2106070001 (https://www.ebi.ac.uk/biomodels/MODEL2106070001). *Supplementary file 1* is provided as a zipped folder with the model in several formats: MaBoSS, GINsim, SBML, as well as images of the networks and their annotations. An extensive description of the model construction can be found in the Appendix 1, Section 1.

## Stochastic Boolean model simulation

MaBoSS (*Stoll et al., 2017*; *Stoll et al., 2012*) is a C++ software for stochastically simulating continuous/discrete-time Markov processes defined on the state transition graph (STG) describing the dynamics of a Boolean model (for more details, see *Abou-Jaoudé et al., 2016*; *Chaouiya et al., 2012*). MaBoSS associates transition rates to each node's activation and inhibition, enabling it to account for different time scales of the processes described by the model. Probabilities to reach a phenotype (to have value ON) are thus computed by simulating random walks on the probabilistic STG. Since a state in the STG can combine the activation of several phenotypic variables, not all phenotype probabilities

are mutually exclusive (like the ones in Appendix 1, Section 6.1, *Appendix 1—figure 28*). Using MaBoSS, we can study an increase or decrease of a phenotype probability when the model variables are altered (nodes status, initial conditions and transition rates), which may correspond to the effect of particular genetic or environmental perturbation. In the present work, the use of MaBoSS was focused on the readouts of the model, but this can be done for any node of the model.

MaBoSS applies Monte-Carlo kinetic algorithm (i.e. Gillespie algorithm) to the STG to produce time trajectories (*Stoll et al., 2017*; *Stoll et al., 2012*), so time evolution of probabilities are estimated once a set of initial conditions are defined and a maximum time is set to ensure that the simulations reach asymptotic solutions. Results are analysed in two ways: (1) the trajectories for particular model states (states of nodes) can be interpreted as the evolution of a cell population as a function of time and (2) asymptotic solutions can be represented as pie charts to illustrate the proportions of cells in particular model states. Stochastic simulations with MaBoSS have already been successfully applied to study several Boolean models (*Calzone et al., 2010*; *Cohen et al., 2015*; *Remy et al., 2015*). A description of the methods we have used for the simulation of the model can be found in the Appendix 1, Section 2.

## Data tailoring the Boolean model

Logical models were tailored to a dataset using PROFILE to obtain personalised models that capture the particularities of a set of patients (*Béal et al., 2019*) and cell lines (*Béal et al., 2021*). Proteomics, transcriptomics, mutations and CNA data can be used to modify different variables of the MaBoSS framework, such as node activity status, transition rates and initial conditions. The resulting ensemble of models is a set of personalised variants of the original model that can show great phenotypic differences. Different recipes (use of a given data type to modify a given MaBoSS variable) can be tested to find the combination that better correlates to a given clinical or otherwise descriptive data. In the present case, TCGA patient-specific models were built using mutations, CNA and/or RNA expression data. After studying the effect of these recipes in the clustering of patients according to their Gleason grading (Appendix 1, Section 4.1, *Appendix 1—figures 10–14*), we chose to use mutations and CNA as discrete data and RNA expression as continuous data.

Likewise, we tried different personalisation recipes to personalise the GDSC prostate cell lines models, but as they had no associated clinical grouping features, we were left with the comparison of the different values for the model's outputs among the recipes (Appendix 1, Section 5, *Appendix 1—figure 23*). We used mutation data as discrete data and RNA expression as continuous data as it included the most quantity of data and reproduced the desired results (Appendix 1, Section 5, *Appendix 1—figure 23*). We decided not to include CNA as discrete data as it forced LNCaP proliferation to be zero by forcing the E2F1 node to be 0 and the SMAD node to be 1 throughout the simulation (for more details, refer to Appendix 1, Section 5).

More on PROFILE's methodology can be found in its own work (*Béal et al., 2019*) and at its dedicated GitHub repository (https://github.com/sysbio-curie/PROFILE; *Béal, 2022*). A description of the methods we have used for the personalisation of the models can be found in the Appendix 1, Section 3. The analysis of the TCGA personalisations and their patient-specific drug treatments can be found in Appendix 1, Section 4. The analysis of the prostate cell lines personalisations can be found in Appendix 1, Section 5, with a special focus on the LNCaP cell line model analysis in Section 6.

## High-throughput mutant analysis of Boolean models

MaBoSS allows the study of knock-out or loss-of-function (node forced to 0) and gain-of-function (node forced to 1) mutants as genetic perturbations and of initial conditions as environmental perturbations. Phenotypes' stabilities against perturbations can be studied and allow to determine driver mutations that promote phenotypic transitions (*Montagud et al., 2019*).

Genetic interactions were thoroughly studied using our pipeline of computational methods for Boolean modelling of biological networks (available at https://github.com/sysbio-curie/Logical_modelling_pipeline; *Montagud, 2022b*). The LNCaP-specific Boolean model was used to perform single and double knock-out (node forced to 0) and gain-of-function (node forced to 1) mutants for each one of the 133 nodes, resulting in a total of 32,258 models. These were simulated under the same initial conditions, their phenotypic results were collected, and a PCA was applied on the wild type-centred matrix (Appendix 1, Section 6.1, *Appendix 1—figures 28 and 29*). In addition, we

found that the LNCaP model is very robust against perturbations of its logical rules by systematically changing an AND for an OR gate or vice versa in all of its logical rules (Appendix 1, Section 6.2, *Appendix 1—figures 30 and 31*).

The 488 TCGA patient-specific models were studied in a similar way, but only perturbing 16 nodes from *Table 1* shortlisted for their therapeutic target potential (AKT, AR, Caspase8, cFLAR, EGFR, ERK, GLUT1, HIF-1, HSPs, MEK1_2, MYC_MAX, p14ARF, PI3K, ROS, SPOP, and TERT). Then, the nodes that mostly contributed to a decrease of *Proliferation* (Appendix 1, Section 4.2, *Appendix 1—figure 19*) or an increase in *Apoptosis* (Appendix 1, Section 4.2, *Appendix 1—figure 20*) were gathered from the 488 models perturbed.

Additionally, the results of the LNCaP model's double mutants were used to quantify the level of genetic interactions (epistasis or otherwise) (*Drees et al., 2005*) between two genetic perturbations (resulting from either the gain-of-function mutation of a gene or from its knock-out or loss-of-function mutation) with respect to wild type phenotypes' probabilities (*Calzone et al., 2015*). The method was applied to the LNCaP model studying *Proliferation* and *Apoptosis* scores (Appendix 1, Section 7.3.2, *Appendix 1—figures 34 and 35*).

This genetic interaction study uses the following equation for each gene pair, which is equation 2 in *Calzone et al., 2015*:

$$\epsilon_\phi\left(A, B\right) = f_\phi^{AB} - \psi\left(f_\phi^A, f_\phi^B\right) \tag{1}$$

where $f_\phi^A$ and $f_\phi^B$ are phenotype $\phi$ fitness values of single gene defects, $f_\phi^{AB}$ is the phenotype $\phi$ fitness of the double mutant, and $\psi\left(x, y\right)$ is one of the four functions:

$$\begin{aligned} \psi^{ADD}(x, y) &= x + y \text{ (additive)} \\ \psi^{LOG}(x, y) &= log_2\left(\left(2^x - 1\right)\left(2^y - 1\right) + 1\right) \text{ (log)} \\ \psi^{MLT}(x, y) &= x * y \text{ (multiplicative)} \\ \psi^{MIN}(x, y) &= min(x, y) \text{ (min)} \end{aligned} \tag{2}$$

To choose the best definition of $\psi\left(x, y\right)$, the Pearson correlation coefficient is computed between the fitness values observed in all double mutants and estimated by the null model (more information on *Drees et al., 2005*). Regarding the $f_\phi^X$ fitness value, to a given phenotype $\phi$, $f_\phi^X < 1$ represents deleterious, $f_\phi^X > 1$ beneficial and $f_\phi^X \approx 1$ neutral mutation.

## Drug simulations in Boolean models

Logical models can be used to simulate the effect of therapeutic interventions and predict the expected efficacy of candidate drugs on different genetic and environmental backgrounds by using our PROFILE_v2 methodology. MaBoSS can perform simulations changing the proportion of activated and inhibited status of a given node. This can be determined in the configuration file of each model (see, for instance, the 'istate' section of the CFG files in the *Supplementary files 1; 3 and 5*). For instance, out of 5,000 trajectories of the Gillespie algorithm, MaBoSS can simulate 70% of them with an activated *AKT* and 30% with an inhibited *AKT* node. The phenotypes' probabilities for the 5000 trajectories are averaged, and these are considered to be representative of a model with a drug that inhibits 30% of the activity of *AKT*. The same applies for a combined drug inhibition: a simulation of 50% *AKT* activity and 50% *PI3K* will have 50% of them with an activated *AKT* and 50% with an activated *PI3K*. Combining them, this will lead to 25% of the trajectories with both *AKT* and *PI3K* active, 25% with both nodes inactive, 25% with *AKT* active and 25% with *PI3K* active.

In the present work, the LNCaP model has been simulated with different levels of node activity, with 100% of node inhibition (proper knock-out), 80%, 60%, 40%, 20%, and 0% (no inhibition), under four different initial conditions, a nutrient-rich media that simulates RPMI Gibco media with DHT (androgen), with EGF, with both and with none. In terms of the model, the initial conditions are *Nutrients* is ON and *Acidosis*, *Hypoxia*, *TGF beta*, *Carcinogen*, and *TNF alpha* are set to OFF. *EGF* and *Androgen* values vary upon simulations. We simulated the inhibition of 17 nodes of interest. These were the 16 nodes from *Table 1* with the addition of the fused AR-ERG (Appendix 1, Section 7.3.1, *Appendix 1—figures 34 and 35*) and their 136 pairwise combinations (Appendix 1, Section 7.3.2, *Appendix 1—figures 36 and 37*). As we used six different levels of activity for each node, the

resulting *Appendix 1—figures 36 and 37* comprise a total of 4,998 simulations for each phenotype (136 × 6 x 6 + 17 x 6).

Drug synergies have been studied using Bliss Independence. The Combination Index was calculated with the following equation (*Foucquier and Guedj, 2015*):

$$CI = \left(E_a + E_b - E_a * E_b\right)/E_{ab} \tag{3}$$

where $E_a$ and $E_b$ is the efficiency of the single drug inhibitions and $E_{ab}$ is the inhibition resulting from the double drug simulations. A Combination Index (*CI*) below 1 represents synergy among drugs (Appendix 1, Section 7.3.2, *Appendix 1—figures 36 and 37*).

This methodology can be found in its own repository: https://github.com/ArnauMontagud/PROFILE_v2.

## Identification of drugs associated with proposed targets

To identify drugs that could act as potential inhibitors of the genes identified with our models (*Table 1*), we explored the drug-target associations in DrugBank (*Wishart et al., 2018*). For those genes with multiple drug-target links, only those drugs that are selective and known to have relevance in various forms of cancer are considered here.

In addition to DrugBank searches, we also conducted exhaustive searches in ChEMBL (*Gaulton et al., 2017*) (http://doi.org/10.6019/CHEMBL.database.23) to suggest potential candidates for genes whose information is not well documented in Drug Bank. From the large number of bioactivities extracted from ChEMBL, we filtered human data and considered only those compounds whose bioactivities fall within a specific threshold (IC50/Kd/ Ki <100 nM).

We performed a target set enrichment analysis using the *fgsea* method (*Korotkevich et al., 2016*) from the *piano* R package (*Väremo et al., 2013*). We targeted pathway information from the GDSC1 and GDSC2 studies (*Iorio et al., 2016*) as target sets and performed the enrichment analysis on the normalised drug sensitivity profile of the LNCaP cell line. We normalised drug sensitivity across cell lines in the following way: cells were ranked from most sensitive to least sensitive (using ln(IC50) as the drug sensitivity metrics), and the rank was divided by the number of cell lines tested with the given drug. Thus, the most sensitive cell line has 0, while the most resistant cell line has 1 normalised sensitivity. This rank-based metric made it possible to analyse all drug sensitivities for a given cell line without drug-specific confounding factors, like mean IC50 of a given drug, etc. (Appendix 1, Sections 7.1 and 7.2).

## Cell culture method

For the in vitro drug perturbation validations, we used the androgen-sensitive prostate adenocarcinoma cell line LNCaP purchased from American Type Culture Collection (ATCC, Manassas, WV, USA). ATCC found no *Mycoplasma* contamination and the cell line was identified using STR profiling. Cells were maintained in RPMI-1640 culture media (Gibco, Thermo Fisher Scientific, Waltham, MA, USA) containing 4.5 g/L glucose, 10% foetal bovine serum (FBS, Gibco), 1 X GlutaMAX (Gibco), 1% PenStrep antibiotics (Penicillin G sodium salt, and Streptomycin sulfate salt, Sigma-Aldrich, St. Louis, MI, USA). Cells were maintained in a humidified incubator at 37 °C with 5% $CO_2$ (Sanyo, Osaka, Japan).

## Drugs used in the cell culture experiments

We tested two drugs targeted at Hsp90 and two targeted at PI3K complex. 17-DMAG is an Hsp90 inhibitor with an IC50 of 62 nM in a cell-free assay (*Pacey et al., 2011*). NMS-E973 is an Hsp90 inhibitor with DC50 of <10 nM for Hsp90 binding (*Fogliatto et al., 2013*). Pictilisib is an inhibitor of PI3K $\alpha/\delta$ with IC50 of 3.3 nM in cell-free assays (*Zhan et al., 2017*). PI-103 is a multi-targeted PI3K inhibitor for p110 $\alpha/\beta/\delta/\gamma$ with IC50 of 2–3 nM in cell-free assays and less potent inhibitor to mTOR/DNA-PK with IC50 of 30 nM (*Raynaud et al., 2009*). All drugs were obtained from commercial vendors and added to the growth media to have concentrations of 2, 8, 32, 128, and 512 nM for NMS-E973 and 1, 5, 25, 125, and 625 nM for the rest of the drugs in the endpoint cell viability and of 3.3, 10, 30 μM for all the drugs in the RT-CES cytotoxicity assay.

## Endpoint cell viability measurements

In vitro toxicity of the selected inhibitors was determined using the viability of LNCaP cells, determined by the fluorescent resazurin (Sigma-Aldrich, Germany) assay as described previously (*Szebeni et al., 2017*). Briefly, the ~10,000 LNCaP cells were seeded into 96-well plates (Corning Life Sciences, Tewksbury, MA, USA) in 100 μL RPMI media and incubated overnight. Test compounds were dissolved in dimethyl sulfoxide (DMSO, Sigma-Aldrich, Germany), and cells were treated with an increasing concentration of test compounds: 2, 8, 32, 128, and 512 nM for NMS-E973 and 1, 5, 25, 125, and 625 nM for the rest of the drugs. The highest applied DMSO content of the treated cells was 0.4%. Cell viability was determined after 48 hours of incubation. Resazurin reagent (Sigma–Aldrich, Budapest, Hungary) was added at a final concentration of 25 μg/mL. After 2 hr at 37 °C 5%, $CO_2$ (Sanyo) fluorescence (530 nm excitation/580 nm emission) was recorded on a multimode microplate reader (Cytofluor4000, PerSeptive Biosystems, Framingham, MA, USA). Viability was calculated with relation to blank wells containing media without cells and to wells with untreated cells. Each treatment was repeated in two wells per plate during the experiments, except for the PI-103 treatment with 1 nM in which only one well was used.

In these assays, a deviation of 10–15% for in vitro cellular assays is an acceptable variation as it is a fluorescent assay that detects the cellular metabolic activity of living cells. Thus, in our analyses, we consider changes above 1.00 to be the same value as the controls.

## Real-time cell electronic sensing (RT-CES) cytotoxicity assay

A real-time cytotoxicity assay was performed as previously described (*Ozsvári et al., 2010*). Briefly, RT-CES 96-well E-plate (BioTech Hungary, Budapest, Hungary) was coated with gelatin solution (0.2% in PBS, phosphate buffer saline) for 20 min at 37 °C; then gelatin was washed twice with PBS solution. Growth media (50 μL) was then gently dispensed into each well of the 96-well E-plate for background readings by the RT-CES system prior to the addition of 50 μL of the cell suspension containing 2 × $10^4$ LNCaP cells. Plates were kept at room temperature in a tissue culture hood for 30 min prior to insertion into the RT-CES device in the incubator to allow cells to settle. Cell growth was monitored overnight by measurements of electrical impedance every 15 min. The next day cells were co-treated with different drugs with concentrations of 3.3, 10 and 30 μM. Treated and control wells were dynamically monitored over 72 hr by measurements of electrical impedance every 5 min. Each treatment was repeated in two wells per plate during the experiments, except for the 3.3 μM ones in which only one well was used. Continuous recording of impedance in cells was used as a measurement of the cell growth rate and reflected by the Cell Index value (*Solly et al., 2004*).

Note that around hour 15, our RT-CES reader had a technical problem caused by a short blackout in our laboratory and the reader detected a minor voltage fluctuation while the uninterruptible power supply (UPS) was switched on. This caused differences that are consistent across all samples and replicates: all wild type and drug reads decrease at that time point, except Pictilisib that slightly increases. For the sake of transparency and as the overall dynamic was not affected, we decided not to remove these readings.

## Acknowledgements

The authors acknowledge the help provided by Jelena Čuklina at ETH Zurich, Vincent Noël at Institut Curie, Annika Meert at Barcelona Supercomputing Center and Aurélien Naldi at INRIA Saclay. The authors acknowledge the reviewers for their comments and suggestions that helped improve and clarify this article. The authors acknowledge the technical expertise and assistance provided by the Spanish Supercomputing Network (Red Española de Supercomputación), as well as the computer resources used: the LaPalma Supercomputer, located at the Instituto de Astrofísica de Canarias and MareNostrum4, located at the Barcelona Supercomputing Center. This work has been partially supported by the European Commission under the PrECISE project (H2020-PHC-668858), the INFORE project (H2020-ICT-825070) and the PerMedCoE (H2020-ICT-951773).

# Additional information

## Competing interests

Luis Tobalina: is a full-time employee and shareholder of AstraZeneca. Vigneshwari Subramanian: is a full-time employee of AstraZeneca. Róbert Alföldi: is CEO of Astridbio Technologies Ltd. László Puskás: is a scientific advisor of Astridbio Technologies Ltd. Alfonso Valencia: Reviewing editor, eLife. Julio Saez-Rodriguez: reports funding from GSK and Sanofi and fees from Travere Therapeutics and Astex. The other authors declare that no competing interests exist.

## Funding

| Funder | Grant reference number | Author |
| --- | --- | --- |
| European Commission | H2020-PHC-668858 | Arnau Montagud<br>Jonas Béal<br>Luis Tobalina<br>Pauline Traynard<br>Vigneshwari Subramanian<br>Bence Szalai<br>Róbert Alföldi<br>László Puskás<br>Emmanuel Barillot<br>Julio Saez-Rodriguez<br>Laurence Calzone |
| European Commission | H2020-ICT-825070 | Arnau Montagud<br>Alfonso Valencia |
| European Commission | H2020-ICT-951773 | Arnau Montagud<br>Alfonso Valencia<br>Emmanuel Barillot<br>Julio Saez-Rodriguez<br>Laurence Calzone |

The funders had no role in study design, data collection and interpretation, or the decision to submit the work for publication.

## Author contributions

Arnau Montagud, Data curation, Formal analysis, Investigation, Methodology, Project administration, Resources, Software, Validation, Visualization, Writing – original draft, Writing – review and editing; Jonas Béal, Formal analysis, Investigation, Methodology, Software, Visualization, Writing – review and editing; Luis Tobalina, Data curation, Investigation, Resources, Writing – original draft, Writing – review and editing; Pauline Traynard, Data curation, Investigation, Methodology, Resources, Writing – original draft, Writing – review and editing; Vigneshwari Subramanian, Data curation, Formal analysis, Investigation, Writing – review and editing; Bence Szalai, Formal analysis, Methodology, Validation, Visualization, Writing – review and editing; Róbert Alföldi, Data curation, Formal analysis, Investigation, Resources, Validation, Writing – review and editing; László Puskás, Funding acquisition, Project administration, Supervision, Validation, Writing – review and editing; Alfonso Valencia, Funding acquisition, Supervision, Writing – review and editing; Emmanuel Barillot, Conceptualization, Funding acquisition, Project administration, Supervision, Writing – review and editing; Julio Saez-Rodriguez, Conceptualization, Funding acquisition, Project administration, Supervision, Writing – original draft, Writing – review and editing, Co-senior authors; Laurence Calzone, Conceptualization, Funding acquisition, Project administration, Supervision, Visualization, Writing – original draft, Writing – review and editing, Co-senior authors

## Author ORCIDs

Arnau Montagud  http://orcid.org/0000-0002-7696-1241
Jonas Béal  http://orcid.org/0000-0003-1949-9801
Luis Tobalina  http://orcid.org/0000-0002-1947-8309
Pauline Traynard  http://orcid.org/0000-0002-4835-9114
Vigneshwari Subramanian  http://orcid.org/0000-0002-7319-8885
Bence Szalai  http://orcid.org/0000-0002-9320-5704
Alfonso Valencia  http://orcid.org/0000-0002-8937-6789

Emmanuel Barillot http://orcid.org/0000-0003-2724-2002
Julio Saez-Rodriguez http://orcid.org/0000-0002-8552-8976
Laurence Calzone http://orcid.org/0000-0002-7835-1148

**Decision letter and Author response**
Decision letter https://doi.org/10.7554/eLife.72626.sa1
Author response https://doi.org/10.7554/eLife.72626.sa2

## Additional files

### Supplementary files

• Supplementary file 1. A zipped folder with the generic prostate model in several formats: MaBoSS, GINsim, SBML, as well as images of the networks and their annotations.

• Supplementary file 2. A jupyter notebook to inspect Boolean models using MaBoSS. This notebook can be used as source code with the model files from *Supplementary file 1* to generate *Figure 3*.

• Supplementary file 3. A zipped folder with the TCGA-specific personalised models and their *Apoptosis* and *Proliferation* phenotype scores.

• Supplementary file 4. A TSV file with all the phenotype scores, including *Apoptosis* and *Proliferation,* of the TCGA patient-specific mutations. In the mutation list "_oe" stands for an overexpressed gene and "_ko" for a knocked out gene.

• Supplementary file 5. A zipped folder with the cell line-specific personalised models.

• Supplementary file 6. A TSV file with all the phenotype scores, including *Apoptosis* and *Proliferation,* of all 32,258 LNCaP cell line-specific mutations and the wild type LNCaP model. In the mutation list "_oe" stands for an overexpressed gene and "_ko" for a knocked out gene.

• Transparent reporting form

### Data availability

Code (and processed data) to reproduce the analyses can be found in a dedicated GitHub (https://github.com/ArnauMontagud/PROFILE_v2 copy archived at swh:1:rev:cdea0bbfa0e7791c-15c0dc452134f1196b4c1b09). Some of the code used in the work can be found in other GitHub repositories (https://github.com/sysbio-curie/PROFILE copy archived at swh:1:rev:2e0e74b21e7e-ac53dbedc46f350511b6558bf75b; https://github.com/sysbio-curie/Logical_modelling_pipeline copy archived at swh:1:rev:5524aae3eece3de1311a1724bd4c6452f0be0542). The model built can be accessed on the Supplementary File 1 and on BioModels and GINsim model repositories (https://www.ebi.ac.uk/biomodels/MODEL2106070001; http://ginsim.org/model/signalling-prostate-cancer). The papers associated with Prostate Adenocarcinoma and Genomics of Drug Sensitivity in Cancer datasets can be found at https://doi.org/10.1016/j.cell.2018.03.022 and https://doi.org/10.1016/j.cell.2016.06.017 respectively.

The following previously published datasets were used:

| Author(s) | Year | Dataset title | Dataset URL | Database and Identifier |
|---|---|---|---|---|
| Hoadley KA, Yau C, Hinoue T | 2018 | Prostate Adenocarcinoma (TCGA, PanCancer Atlas) | https://www.cbioportal.org/study/summary?id=prad_tcga_pan_can_atlas_2018 | cBioPortal, prad_tcga_pan_can_atlas_2018 |
| Iorio F | 2016 | GDSC 1 and 2 | https://www.cancerrxgene.org/downloads/bulk_download | Genomics of Drug Sensitivity in Cancer, GDSC1/2 |

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

# Appendix 1

## 1. Prostate Boolean model construction

Building the model is done in three steps:

1. Identifying signalling pathways or particular genes and proteins that are especially relevant to describe the prostate cancer tumorigenesis and tumour growth. Most of them are components that are known to be frequently altered in cancers.
2. Building a regulatory network that includes simplified representations of pathways identified as relevant for prostate cancer, as well as all individually identified genes. Each pathway is characterised by the key players that regulate it. This network takes the form of a directed graph for which positive and negative influences between components are represented.
3. From this network, a logical model is derived describing the network dynamics in specific contexts (dependent on initial conditions or perturbations). To this end, logical rules are associated with each node of the network to indicate how it is activated or inhibited by different combinations of its regulators.

### 1.1. Prior knowledge network construction

We started by using a published logical model of human signalling network (*Fumiã and Martins, 2013*), which is based on integrated experimental evidence of signal transduction. This model integrates major signalling pathways that have a role in regulating cell death and proliferation in many tumours. They include those involving receptor tyrosine kinase (RTKs), phosphatidylinosital 3-kinase (PI3K)/AKT, WNT/β-Catenin, transforming growth factor-b (TGF-β)/Smads, cyclins, retinoblastoma protein (Rb), hypoxia-inducible transcription factor (HIF-1), p53 and ataxia-telangiectasia mutated (ATM)/ataxia-telangiectasia and Rad3-related (ATR) protein kinases. The pathways reveal substantial cross-talks.

This initial generic network was then extended to include prostate cancer-specific genes and proteins using several approaches presented below.

### 1.1.1. Definition of inputs and outputs

Our Boolean model aims at predicting prostate phenotypic behaviours for healthy and cancer cells in different "physiological" conditions. To account for these conditions, we considered nine inputs that represent different physiological conditions of interest. These are *EGF, FGF, TGF beta, Nutrients, Hypoxia, Acidosis androgen, TNF alpha, and Carcinogen* presence. These input nodes have no regulation and their values are fixed for each simulation, representing the cell's microenvironmental characteristics.

For simplicity, we choose to clearly define phenotype variables as output nodes allowing the integration of multiple phenotypic signals and obtaining a 0/1 value for each phenotype. Our model has a total of 11 outputs. We define three main phenotypes representing the growing status of the cell: *Proliferation, Apoptosis,* and *Quiescence. Apoptosis* is activated by Caspase 8 or Caspase 9, while *Proliferation* is activated by cyclins from the cell cycle. We define *Quiescence* as the absence of *Proliferation* and *Apoptosis* and these two, although not directly linked, are always mutually exclusive in simulations.

The proliferation output is sometimes described in already published models as specific stationary protein activation patterns, namely the following sequence of activation of cyclins: Cyclin D, then Cyclin E, then Cyclin A, then Cyclin B. This sequence can easily be detected in complex attractors in synchronous dynamics. However, since asynchronous dynamics was chosen for this work and it is more difficult to analyse complex attractors with it, we define *Proliferation* as activated by either of the four cyclins. Transient dynamics in MaBoSS simulations allow us to check the correct oscillation of cyclins.

Furthermore, we define several phenotypic outputs that are not mutually exclusive but detect the activation of some markers of cancer hallmarks: *Invasion, Migration,* (bone) *Metastasis,* and *DNA repair*.

### 1.1.2. Identification of new components based on literature search:

Several studies have focused on identifying main subtypes among the heterogeneous molecular abnormalities in prostate cancer. In particular, a TCGA study (*Cancer Genome Atlas Research*

*Network, 2015*) reported a comprehensive molecular analysis of 333 primary prostate carcinomas. Seven subtypes, containing 74% of these tumours, were defined by specific gene fusions (ERG, ETV1/4, and FLI1) or mutations (SPOP, FOXA1, and IDH1). Epigenetic profiles allowed us to identify a methylator phenotype in the IDH1 mutant subset. SPOP and FOXA1 mutant tumours show the highest levels of AR-induced transcripts. Lesions in the PI3K or MAPK signalling pathways are observed in 25% of the prostate cancers and DNA repair genes inactivation in 19%.

The following list of frequently mutated genes extracted from this study indicate components that could be included in the model, provided that enough information is available on their mechanistic roles:

- gene fusions: ERG, ETV1, ETV4, FLI1
- deletions: SPOP, FOXA1, IDH1, TP53, PTEN, PIK3CA, BRAF, CTNNB1, HRAS, MED12, ATM, CDKN1B, RB1, NKX3-1, AKT1, ZMYM3, KMT2C, KMT2D, ZNF595, CHD1, BRCA2, CDK12, SPINK1
- amplifications: CCND1, MYC, FGFR1, WHSC1L1.

Comparing with a published cohort of 150 castration-resistant metastatic prostate cancer samples (*Robinson et al., 2015*), the authors find a similar subtype distribution as in *Cancer Genome Atlas Research Network, 2015*, with increased alteration rates in the metastatic samples and more frequent amplification or mutation of AR, as well as DNA repair and PI3K pathway alterations.

Other studies such as (*Altieri et al., 2009*) focus on the role of specific pathways which play a critical role in prostate cancer maintenance, such as chaperone-mediated mitochondrial homeostasis (in particular with HSP90 found very abundant in prostate cancer), integrin-dependent cell signalling and RUNX2-regulated gene expression in the metastatic bone microenvironment.

Notably, a set of regulatory maps of signalling pathway maps and altered circuitries of various cell biological events associated with the pathogenesis of human prostate cancer have been published recently (*Datta et al., 2016*). The authors manually constructed networks based on the literature. These networks constitute an important resource for retrieving information on prostate cancer specific components. Although not exhaustive, these maps are synthetic pictures of the existing knowledge on molecular events involved in prostate cancer hallmarks.

The covered hallmarks include: (1) classical cancer hallmarks: insensitivity to anti-growth signal, self-sufficiency in growth signal, tumour promoting inflammation, genome instability, mutation and perturbation, angiogenesis, metastasis, cell death resistance, metabolic reprogramming, avoidance of immune destruction, enabling replicative immortality, tumour microenvironment; and (2) prostate cancer specific hallmarks: androgen receptor signalling androgen independence, castration resistance.

This study points toward some candidate nodes to extend our network in order to take into account, at least in a simplified way, most pathways present in the maps. In particular, it shows that the initial network obtained through combinations of published models ignore any pathways related to inflammation, metabolism, immune evasion, or the tumour microenvironment. However, the resource contains few mechanistic details for the interactions between its components, which are a mix of genes, proteins, molecules, processes and phenotypes.

Finally, among all these genes associated with prostate cancer, a subset has been chosen for further research: AR, PTEN, SPOP, TP53, EZH2, FOXA1, BRCA1, BRCA2, PIK3CA, AKT1, NCOA2, NCOR1, NCOR2, EP300, MYC, RB1, CHD1, CDKN1B, MED12, ZNF595, HOXB13.

### 1.1.3. Identification of new components/pathways based on data analysis

ROMA (*Martignetti et al., 2016*) is a software package written in Java for the quantification and representation of biological module activity using expression data. It uses the first principal component of a PCA analysis to summarise the coexpression of a group of genes in the gene set.

We apply ROMA analysis on the transcriptomics data of TCGA. We define gene sets as they are described in the atlas of cancer signalling networks, ACSN (*Kuperstein et al., 2015*) (http://www.acsn.curie.fr/) and in the Hallmarks (*Liberzon et al., 2015*). ACSN is centred on signalling pathways such as DNA repair, cell death, EMT, cell adhesion, cell cycle, etc. and the Hallmarks gene sets provide a list of genes that participate in biological processes integrating information from other pathway databases.

Using ROMA, we are able to identify some pathways significantly overdispersed over the samples that should have relevant roles in prostate cancer and need therefore to be correctly described in the model.

The results show that, for ACSN database, among the 140 pathways from the database, 65 modules reveal a high variance of protein expression across all samples (*Appendix 1—figure 1*). The gene sets linked to the cell cycle seem to show a progressive activation from normal to high grade tumours, so does the DNA repair pathway with differences in the mechanisms that participate in DNA repair, whereas some gene sets such as the one related to immunosuppressive cytokine pathway show opposite behaviour. We performed the same analysis with the Hallmarks database and found 16 out of the 50 pathways that showed high variance. We can confirm the role of the cell cycle in tumour progression (E2F_targets and G2_M checkpoints).

Note that in both analyses, we see that group three is the most heterogeneous group, with a score that does not always follow the trend of increasing or decreasing pathway scores from group 1 to group 5.

ROMA provides some hints on where to extend the network to fully grab the alterations that are found in prostate cancer patients. For instance, the Hedgehog pathway was not described in the already published logical models that we used as a starting point of this model. Moreover, both the cell cycle and the DNA repair pathways were overly simplified, and were thus extended in this version.

Some pathways related to the immune response seem to be highly represented in ACSN and would need to be included in future extended versions of the prostate network, probably in the form of interacting networks of different cell types.

## 1.1.4. Model extension with Omnipath via pypath

OmniPath (*Türei et al., 2016*; *Türei et al., 2021*) is a comprehensive collection of high confidence, literature curated, human signalling pathways. It is accompanied and developed together with Pypath, a Python module for cellular signalling pathways analysis.

Pypath is a python module used to query the content of Omnipath in order to retrieve components and interactions in the human protein-protein signalling network associated with annotations, especially sources, literature references, direction, effect signs (stimulation/inhibition), and enzyme-substrate interactions.

The development of pypath allows us to build personalised queries. For instance, existing interaction paths between a protein of interest and a list of user-defined proteins can be found, with a given size for the paths. We use this in the extension process of our network to automatically find new interactions between a new gene and the genes already included in the network. We filter the interactions found to select the ones for which the direction and sign are known.

For example, when extending the network with the chaperone protein HSP90AA1, we generate the graph displayed in *Appendix 1—figure 2*, which shows all signed directed interactions linking HSP90AA1 to the network. The associated references given as annotations are useful to check the mechanism behind each interaction and manually infer a logical rule.

## 1.1.5. Model extension with the literature

Protein-protein interactions (PPI) and signalling databases are useful to find quickly established interactions between genes and proteins. However, they are not exhaustive and in particular they often lack recent findings. It is therefore necessary to rely on manual literature search to find information on specific prostate cancer components.

The roles of the fusion gene TMPRSS2:ERG and the tumour suppressor NKX3-1 are examples where the information from databases retrieved from Omnipath or PPI databases is lacking, and for which we found additional information from the literature.

Fusion genes are frequently found in human prostate cancer and have been identified as a specific subtype marker (*Cancer Genome Atlas Research Network, 2015*). The most frequent is TMPRSS2:ERG. It involves the transcription factor ERG, which leads to cell-cycle progression. ERG fuses with the AR-regulated TMPRSS2 gene promoter to form an oncogenic fusion gene that is especially common in hormone-refractory prostate cancer, conferring androgen responsiveness to ERG. This fusion is not found with Pypath, nor is any target of ERG (*Appendix 1—figure 3A*). However, literature search reveals that ERG directly regulates EZH2, oncogene c-Myc and tumour suppressor NKX3-1 and many other targets in prostate cancer (*Kunderfranco et al., 2010*).

We model the gene fusion with an activation of ERG by the decoupling of ERG in a special node AR_ERG that is only activated by the AR & fused_event node. In the healthy case, fused_event (that

represents TMPRSS2) is fixed to 0 or inactive. The occurrence of the gene fusion is represented with the model perturbation where fused_event is fixed to 1. Moreover, ERG expression has a major impact on cell invasion and epithelial-mesenchymal transition (EMT) through the upregulation of the FZD4 gene, a member of the frizzled family of receptors. In our model, we choose for simplicity to consider ERG as a marker of EMT, with a direct activation of the output node EMT by ERG (*Adamo and Ladomery, 2016*).

NKX3-1 has been identified as a tumour suppressor for prostate cancer. Since it is frequently mutated, it should be included in the model. Some of its regulations can be found with Pypath (*Appendix 1—figure 3B*), in particular its activation by AR and PKC. However, its role is not identified. The literature search highlighted its role in accelerating the DNA repair response and in particular in avoiding the gene fusion TMPRSS2:ERG. NKX3-1 binds to AR at the ERG gene breakpoint and inhibits both the juxtaposition of the TMPRSS2 and ERG gene loci and also their recombination, by influencing the recruitment of proteins that promote homology-directed DNA repair. Thus, loss of NKX3-1 favours recruitment to the ERG gene breakpoint of proteins that promote error-prone non-homologous end-joining (*Bowen et al., 2015*).

We therefore add the absence of the node NKX3-1 as a new requirement for the activation of ERG by AR and TMPRSS2 in the model. The effect of the gene fusion can be seen in combination with the perturbation that maintains NKX3-1 to the null level.

In contrast with these examples where some knowledge can be retrieved from the literature, some new nodes cannot be included in the model in a satisfactory manner, because of missing information about their regulation or role. High-throughput studies have allowed us to identify genes with mutations or expression levels associated with prostate cancer progression or prognosis. Nevertheless, for many of them, the precise mechanisms behind this association remains to be elucidated.

For example, IDH1 (isocitrate dehydrogenase 1) exhibits a recurrent mutation in 1% of primary prostate cancers that defines a specific subtype (*Cancer Genome Atlas Research Network, 2015*). This mutant status is associated with a DNA hypermethylation phenotype. Despite a lack of detailed mechanisms linking this gene to the regulation network, we can still reflect a candidate association in the model by including IDH1 as regulated by mTOR and MEK1_2, whose absence (level 0) induces the activation of the output node *Hypermethylation*. The regulation of both new nodes IDH1 and Hypermethylation should be refined when new knowledge is found.

In some cases, we cannot provide any link for a new node, either to an existing node or to a phenotypic output, even qualitatively. For example, ZNF595 has been linked to prostate cancer progression. However, this gene encodes a protein belonging to the Cys2His2 zinc finger protein family, whose members function as transcription factors that can regulate a broad variety of developmental and cellular processes. This knowledge is not detailed enough to add this node in the model yet. However, future mutation data from prostate cancer samples, associated with clinical data, will allow us to test several hypotheses.

This model includes several signalling pathways as well as the substantial cross-talks among them. These pathways range from receptors such as receptor tyrosine kinase (RTKs), androgen receptor (AR) and growth factors pathways (EGF, FGF, TGF-β); downstream gene regulation pathways such as phosphatidylinositol 3-kinase (PI3K)/AKT, Wnt/β-Catenin, NFkB, MAPK, mTOR, SHH, MYC, ETS1, p53, hypoxia-inducible transcription factor (HIF-1) and Smad pathways; cell cycle descriptions with cyclins, E2F1, retinoblastoma protein (Rb) and p21; epithelial-mesenchymal transition (EMT) and migration-related genes; DNA damage and apoptosis-related genes; as well as prostate cancer characteristic genes such as p53, ataxia-telangiectasia mutated (ATM)/ataxia-telangiectasia and Rad3-related (ATR) protein kinases, NKX3.1, TMPRSS2 and TMPRSS2:ERG fusion.

A complete list of the references for all the nodes and edges included in the model can be found in the XLS file of *Supplementary file 1*.

## 1.2 Boolean model construction
### 1.2.1. Primer on Boolean modelling
Boolean models are based on the logical formalism that relies on a regulatory graph and a list of logical rules associated with each of the nodes of the graph. We hereby present a small introduction

of the principal terms of this modelling. For further information, we refer readers to other works (*Béal et al., 2019*; *Saadatpour and Albert, 2013*; *Abou-Jaoudé et al., 2016*).

The aforementioned prior knowledge network is composed of nodes and edges, where nodes correspond to entities (e.g. genes, proteins, complexes, phenotypes or processes) and edges to influences, either positive or negative, which illustrate the possible interactions between two entities. Such regulatory networks are easily translatable to Boolean models. A node that has no regulator is denoted as *input* and a node that does not regulate another node is denoted as *output*. *Input* represent different physiological initial conditions and *outputs* represent biological read-outs.

Each node of the regulatory network has a corresponding Boolean variable associated that can take two values: 0 for inactive or OFF, and 1 for active or ON. These variables change their value according to a logical rule assigned to them. The state of a variable will thus depend on its logical rule, which is based on logical statements: a function of the node regulators linked with logical connectors AND, OR and NOT. More on this in Section 1.2.2 "Establishing the rules of the Boolean model".

These operators can account for what is known about the biology behind these edges. If two input nodes are needed for the activation of the target node, they will be linked by an AND gate; to list different means of activation of a node, an OR gate will be used. For negative influences, a NOT gate will be utilised.

Finally, the state transition graph (STG) is another network that recapitulates all the states of the nodes and the possible transitions from one model state to another depending on the logical rules. The form of the graph will depend on the updating strategy chosen -either all nodes are updated at once or nodes are updated one at a time. In addition, the state transition graph informs on the existence of the two types of attractors of the model: stable steady states or limit cycles. More on this in Section 1.2.3 "State transition graph and the update mechanism".

### 1.2.2. Establishing the rules of the Boolean model

When building our regulatory graph, there were many instances of concurrent activation and inhibition of a node. As a general rule, and unless evidence was found for the contrary, we decided to add the activators with OR gates and the inhibitors with AND NOT.

Usually the OR links activators from two different pieces of information extracted from different articles. For the inhibitors, the AND NOT allows to take into account their effect and overrule the activators.

This is an assumption that we make as a first try and when we have no further knowledge. If there is evidence that one of the activators is not affected when an inhibitor is present, then we adapt the logical formulas accordingly. For instance, if we know that two inhibitors only inhibit when both are present, we include that information and overwrite the previous formula.

Some of the possible combinations that we may find in Boolean models can be found in the following toy model (*Appendix 1—figure 4*). Node D and E are self-regulated, meaning they are inputs: their initial value will rule their activation. Node A can be activated by B and any combination of C and/or E. Node B is activated if D is not present and when A or C are present. C is activated by A and only when D and E are both not present. This means that C can still be activated when A and D are present, or A and E, but not D and E.

### 1.2.3. State transition graph and the update mechanism

In a Boolean framework, the variables associated to each node can take two values, either 0 or 1. We define a model state as a vector of all node states. All the possible transitions from any model state to another are dependent on the set of logical rules that define the model.

These transitions can be viewed into a graph called a state transition graph, or STG, where nodes are model states and edges are the transitions from one model state to another.

The resulting dynamics of the Boolean model can be represented in terms of a state transition graph (STG), where the nodes denote the states of the system (i.e. vectors giving the levels of activity of all the variables) and the arcs represent state transitions (i.e. changes in variable values, according to the corresponding logical functions). This way, trajectories from an initial condition to all the final states can be determined. The STG can contain up to $2^n$ model state nodes; thus, if $n$ is too big, the construction and the visualisation of the graph becomes resource consuming.

The attractors of the model are the long-term asymptotic behaviours of the system. We have two types: stable states, when the system has reached a model state whose successor in the transition graph is itself; and cyclic attractors, when trajectories in the transition graph lead to a group of model states that are cycling. For more details, see *Chaouiya et al., 2012*; *Abou-Jaoudé et al., 2016*.

When concurrent variable changes are enabled at a given state, the resulting state transition depends on the chosen updating assumptions. Numerous studies use the fully synchronous strategy where all variables are updated through a unique transition. This assumption leads to relatively simple transition graphs and deterministic dynamics. The proportion of initial conditions leading to given attractors is measured as the attractor landscape (*Helikar et al., 2008*; *Fumiã and Martins, 2013*; *Cho et al., 2016*). However, the synchronous updating assumption approximation often leads to spurious cyclic attractors. On the other hand, the fully asynchronous updating assumption considers separately all possible transitions and therefore allows the consideration of alternative dynamics in the absence of kinetic data. The resulting dynamics has a branching structure which makes it more difficult to evaluate. In this project, we consider asynchronous dynamics mixed with stochastic simulations.

The regulatory graph was constructed using GINsim software (*Chaouiya et al., 2012*) and then exported in a format readable by MaBoSS software (see below) in order to perform stochastic simulations on the Boolean model.

The final model accounts for 133 nodes and 449 edges (*Appendix 1—figure 1* and *Supplementary file 1*) and includes pathways such as androgen receptor and growth factor signalling, different signalling pathways (Wnt, NFkB, PI3K/AKT, MAPK, mTOR, SHH), cell cycle, epithelial-mesenchymal transition (EMT), Apoptosis, DNA damage, etc. This model has nine inputs (*EGF, FGF, TGF beta, Nutrients, Hypoxia, Acidosis, Androgen, TNF alpha* and *Carcinogen* presence) and six outputs (*Proliferation*, *Apoptosis*, *Invasion*, *Migration*, (bone) *Metastasis* and *DNA repair*).

## 2. Boolean model simulation

### 2.1. Primer on MaBoSS methodology

In the present study, all simulations have been performed with MaBoSS that stands for Markovian Boolean Stochastic Simulator. We hereby present a small introduction of the MaBoSS simulations. For further information, we refer readers to other works (*Béal et al., 2019*; *Stoll et al., 2012*; *Stoll et al., 2017*).

This framework is based on an asynchronous update scheme combined with a continuous time feature obtained with the Gillespie algorithm (*Gillespie, 1976*), allowing simulations to be continuous in time. This algorithm is particularly useful when the state transition graph is too big, as it allows to stochastically sample trajectories from a given initial condition to all possible asymptotic solutions and associate a probability to each model state and final stable states.

Gillespie algorithm provides a stochastic way to choose a specific transition among several possible ones and to infer a corresponding time for this transition. Thus, MaBoSS computation results in one stochastic trajectory as a function of time when objective transition rates, seen as qualitative activation or inactivation rates, are specified for each node. These transition rates can be set either all to the same value by default or in various levels reflecting different orders of magnitude of biological processes' time or due to difference among different patients' omics datasets (See Section 3.1 "Primer on PROFILE methodology"). These transition rates are translated as transition probabilities in order to determine the actual transition. All in all, this modelling framework is at the intersection of logical modelling and continuous dynamic modelling.

Since MaBoSS computes stochastic trajectories, it is highly relevant to compute several trajectories to get an insight of their average behaviour. In present work, all simulations have consisted on the average of 5,000 computed trajectories.

To capture the gradual inhibition of drugs (Section 7.3), we have taken advantage of the simulation of a population of trajectories, so initial values of each node can be defined with a continuous value between 0 and 1 representing the probability for the node to be defined to one for each new trajectory. For instance, a node with a 0.7 initial condition will be set to 1 in 70% of simulated trajectories and to 0 in 30% of the trajectories.

## 2.2. Wild type simulation

Our prostate Boolean model recapitulates known phenotypes of prostate cells by stochastic simulations in each of the studied "physiological" conditions. The model can be considered as a model of healthy prostate cells when no mutants or fused genes are present, called wild type model in present work. These healthy cells mostly exhibit quiescence in absence of any input. Because the initial conditions of all components of the model are set to random values and input nodes are OFF, there is a possibility to activate transiently the pathways but not to maintain them, and all pathways are eventually turned off.

Our prostate Boolean model was simulated using MaBoSS and asynchronous updates and recapitulates known phenotypes of prostate cells under physiological conditions (Main text, *Appendix 1—figure 2*). Model states distribution at the end of the simulation with growth factors, *Nutrients* and *Androgen* as inputs can be seen in *Appendix 1—figure 2B*. Note that some outputs are not mutually exclusive, therefore the presence of cells with *Invasion* and *Proliferation*. In *Appendix 1—figure 2C*, the same model with cell death factors ON.

In proliferating conditions, transient probabilities of the cyclins can be used to check that the order of activations of these nodes in the paths leading to the cyclic attractor is consistent with a proper cell cycle progression (*Appendix 1—figure 5*).

These analyses can be performed using model files from *Supplementary file 1* and the jupyter notebook from *Supplementary file 2*.

## 2.3. Mutants simulation

A mutant in the logical framework is simulated by setting the node corresponding to the gene mutated to 0 in the case of loss of function and to 1 in the case of gain of function. The effect of a mutation is assessed, like the change of initial conditions, by comparing the mutant's probabilities of reaching a phenotype with respect to the wild type model. Therefore, mutations change the model phenotypes: *Apoptosis, Proliferation, Invasion, Migration,* (bone) *Metastasis* and *DNA repair*.

### 2.3.1. Single mutations

The single mutations of some of the main nodes of the network show some changes in the probabilities of reaching the phenotypes when compared to wild type conditions.

The examples on *Appendix 1—figure 6* show that a loss-of-function mutation of FOXA1 in proliferative conditions (nutrients and growth factors) results in the activation of migration and invasion but not metastasis. A loss-of-function mutation of TP53 in the same condition with the addition of carcinogen does not lead to loss of the apoptosis induced by DNA damage because of the activation of caspase three pathway.

### 2.3.2. Multiple mutations

Cancer progression is characterised by the accumulation of genetic alterations that affect multiple pathways in the signalling network. The logical model allows to easily simulate all possible combinations of mutations and study the potential redundancy or synergy of alteration effects and the importance of order. An example of double mutation is shown in *Appendix 1—figure 7*, where the combination of the gene fusion TMPRSS2:ERG and the loss-of-function of NKX3-1 activates bone metastasis signals in proliferative conditions with androgen induction.

The model allows to study easily all possible associations of mutations to assess synergies or redundancies. It can also reproduce sets of mutations observed in tumours. Different sequences of possible acquired mutations can be simulated and compared to what is already known about patients harbouring these mutations.

## 3. Personalisation of Boolean models

### 3.1. Primer on PROFILE methodology

We give here an intuitive idea of how the personalization is done with PROFILE for both discrete data (mutation and copy number alteration data) and continuous data (RNAseq and/or proteomics data when available). For more thorough details on the methodology, readers can refer to *Appendix 1—figure 8* and the work described in *Béal et al., 2019*.

For discrete data: if the mutation is an activating mutation, the corresponding node will be set to 1; if the mutation is an inhibiting mutation, the corresponding node will be set to 0.

For continuous data, the data is normalised at first. Then, depending on the expression of the gene compared to others, the corresponding transition rate will be set to a high value if it is higher and to a low value if it is lower. A high transition rate will be favoured when travelling through the state transition graph. Initial values for these genes are also set accordingly. This personalisation can be observed in the CFG file for the LNCaP cell line (*Appendix 1—table 1*). The full file is available in the *Supplementary file 5*.

**Appendix 1—table 1.** Excerpt of the CFG file of the personalised LNCaP Boolean model.

| Transition rates for LNCaP personalised model | Initial conditions for LNCaP personalised model |
|---|---|
| $u_Acidosis = 1; | [Acidosis].istate = 0.5[1], 0.5 [0]; |
| $d_Acidosis = 1; | [Androgen].istate = 0.5[1], 0.5 [0]; |
| $u_AKT = 1.15285; | [Carcinogen].istate = 0.5[1], 0.5 [0]; |
| $d_AKT = 0.86742; | [Hypoxia].istate = 0.5[1], 0.5 [0]; |
| $u_AMP_ATP = 0.06407; | [Nutrients].istate = 0.5[1], 0.5 [0]; |
| $d_AMP_ATP = 15.60793; | [AKT].istate = 0.51544[1], 0.48456 [0]; |
| $u_AMPK = 0; | [AMP_ATP].istate = 0.20167[1], 0.79833 [0]; |
| $d_AMPK = 0.91263; | [ATR].istate = 0.32278[1], 0.677219 [0]; |
| $u_Androgen = 1; | [AXIN1].istate = 0.38829[1], 0.61171 [0]; |
| $d_Androgen = 1; | [BAD].istate = 0.65311[1], 0.34689 [0]; |
| $u_Angiogenesis = 1; | [Bak].istate = 0.32278[1], 0.677219 [0]; |
| $d_Angiogenesis = 1; | [Bcl_XL].istate = 0.36264[1], 0.637359 [0]; |
| $u_Apoptosis = 1; | [BCL2].istate = 1e-05[1], 0.99999 [0]; |
| $d_Apoptosis = 1; | [BIRC5].istate = 0.34426[1], 0.65574 [0]; |
| $u_AR = 100.0; | [BRCA1].istate = 0.42294[1], 0.57706 [0]; |
| $d_AR = 0; | [Caspase8].istate = 0.21981[1], 0.780189 [0]; |
| $u_AR_ERG = 1; | [Caspase9].istate = 0.32278[1], 0.677219 [0]; |
| $d_AR_ERG = 1; | [CDH2].istate = 0.0[1], 1.0 [0]; |
| $u_ATM = 0; | [cFLAR].istate = 0.5[1], 0.5 [0]; |
| $d_ATM = 5.81395; | [CyclinB].istate = 0.23353[1], 0.76647 [0]; |
| … | … |

## 3.2. Differences of PROFILE with the state of the art

Personalised models should be able to capture heterogeneity among cancer cell lines, cells of a tumour and cells from different patients. Until now, personalisation of models has used in vitro perturbation experiments, as studying this kind of cell-level heterogeneity between patients' responses to treatments is complicated in vivo. In vitro studies such as the ones from *Saez-Rodriguez et al., 2009* and *Dorier et al., 2016* showed how perturbation data could be used to capture differences in the models of different cell lines and patients.

Moreover, in vitro perturbation results are best when researchers can isolate the cells from their surrounding environment and study a small set of them, as happens with microfluidics techniques. *Eduati et al., 2020* showed a procedure in which cells from two cell lines and four biopsies were tested against a panel of 8 drugs and their combinations. These drug responses were then used to personalise a generic model.

Our PROFILE methodology does not use in vitro perturbation experiments, but rather bulk omics data. We are capable of having results specific for each cell line and patient without the need of in vitro testing. The perturbation data does not lack any kind of information to have these personalised models, but we consider that being able to personalise models without needing further experimentation is an asset of our method. In any case, note that the present PROFILE_v2 methodology and perturbation tools as the ones above are compatible and complementary as they use different kinds of data as inputs.

## 4. Personalised Boolean models of TCGA patients

Our prostate Boolean model was tailored to a set of 488 TCGA prostate cancer patients using our PROFILE personalisation method (*Béal et al., 2019*). The distribution of the 488 patients' Gleason score can be seen in *Appendix 1—figure 9*. The prostate cancer patients recipe that has a better correlation with their Gleason score was using mutations and copy number alterations (CNA) as node activity status and RNA as initial conditions and transition rates (*Appendix 1—figure 10*). All the 488 TCGA prostate cancer patients' models can be found in MaBoSS format in *Supplementary file 3*.

### 4.1. Phenotype distribution of TCGA patients

One of the quality checks performed in PROFILE is to build models using different recipes, i.e. using different data to modify different model variables, and to compare them to some clinical grouping or expression signature to rank them and select the most performing one. In our case, we used five different recipes (only mutations, mutations and CNA, mutations and RNA data, mutations, CNA and RNA data and only RNA data), we grouped the patients by their GG (either 3- or 5-stage) and studied the distributions of the different phenotypes scores: *Apoptosis* (*Appendix 1—figure 10*), *DNA repair* (*Appendix 1—figure 11*), *Invasion* (*Appendix 1—figure 12*), *Migration* (*Appendix 1—figure 13*) and *Proliferation* (*Appendix 1—figure 14*). Finally, we chose the recipe that uses mutations, CNA and RNA data as it included the most quantity of data and reproduced desired results (*Supplementary file 3*). Note that the correspondence between 3- and 5-stage GG is the following: GG Low is GG 1, GG Intermediate is GG 2 and 3 and GG High is GG 4 and 5. We used the Kruskal-Wallis rank sum test to identify if the phenotype distributions across 3- and 5-stage GG could originate from different distributions and, if significant, used the Dunn's nonparametric pairwise multiple comparisons test to identify which pairs of groups are statistically different.

Next, we took the personalised models that used mutations, CNA and RNA data and performed a PCA analysis on the 488 TCGA patients (*Supplementary file 3*) and their five phenotype scores that result from simulating them using MaBoSS. For these PCA, we grouped the patients by 3-stages GG (*Appendix 1—figure 15*) and 5-stages GG (*Appendix 1—figure 16*). In addition and for the sake of clarity, we reduced each of these groups to their barycenter (*Appendix 1—figure 17* for 3-stages GG and *Appendix 1—figure 18* for 5-stages GG), where we can see that higher GG move towards *Proliferation*, *Invasion* and *Migration* variables.

### 4.2. Analysis of drugs that inhibit the activity of genes of TCGA patients

Using our pipeline of tools (*Montagud et al., 2019*), we performed the analysis of all single perturbations that reduce *Proliferation* or increase *Apoptosis* together with the combined perturbations of a set of selected genes that are targets of already-developed drugs relevant in cancer progression (*Table 1*). Then, we aggregated the results of the 488 patients to identify which inhibitions affected *Proliferation* (*Appendix 1—figure 19*) and *Apoptosis* (*Appendix 1—figure 20*) the most in this cohort.

Interestingly, we found several genes that were found as suitable points of intervention in most of the patients (MYC_MAX complex and SPOP were identified in more than 80% of the cases) (*Appendix 1—figure 19* and *Appendix 1—figure 20*), but others were specific to only some of the patients (MXI1 was identified in only 4 patients, 1% of the total, GLI in only 7% and WNT in 8% of patients).

The inactivation of some of the targeted genes had greater effect in some patients than in others, suggesting the possibility for the design of personalised drug treatments (Main text). Nevertheless, knowing that some treatments that inhibit one gene are already able to reduce *Proliferation* phenotypes considerably, we explored the possibility of finding combinations of treatments that could lead to the same types of outcomes. One reason for searching for coupled drugs is that these combinations allow the use of lower doses of each of the two drugs and thus reduce their toxicity. It is important to note, though, that the analyses performed with the mathematical model do not aim at predicting drug dosages per se but to help in the identification of potential candidates.

The exhaustive search for combinations of drugs for each patient of the cohort requires an extensive amount of computation time (9 days and 7 hr on a personal computer or 3 hr on 20 nodes with 48 CPUs each, per model) as all variables of the model are automatically overexpressed and inhibited, one by one and in pairs, leading to a vast amount of simulations. For this reason, we have narrowed the list of potential candidates to reduce *Proliferation* or increase *Apoptosis* by

performing the analysis of all single perturbation and selecting the combined perturbations of a set of selected genes that are targets of already-developed drugs relevant in cancer progression (Main text, *Table 1*).

We used the models to grade the effects that the combined treatments would have in each one of the 488 TCGA patient-specific models. The resulting list of combinations vary greatly from patient to patient, making it infeasible economically for labs and companies to pursue true patient-specific treatments. It also poses challenges in clinical trial designs aimed at validating the model based on the selection of treatments. Because of these constraints, it is more interesting commercially to target group-specific treatments, which can be more easily related to clinical stages of the disease. Mathematical modelling of patient profiles would then help to classify them in these groups, providing, in essence, a grade-specific treatment.

The TCGA mutants and their normalised phenotype scores in regards to the wild type model can be found in *Supplementary file 4*.

## 5. Personalised Boolean models of prostate cell lines

We tailored our generic prostate model to eight prostate-specific cell lines: 22RV1 (*Sramkoski et al., 1999*), BPH-1 (*Hayward et al., 1995*), DU-145 (*Stone et al., 1978*), LNCaP-Clone-FGC (*Horoszewicz et al., 1983*), NCI-H660 (*Johnson et al., 1989*; *Lai et al., 1995*; *Castoria et al., 2011*), PC-3 (*Kaighn et al., 1979*), PWR-1E (*Webber et al., 1996*), and VCaP (*Korenchuk et al., 2001*). These cell lines had available datasets in the GDSC resource (*Iorio et al., 2016*) and these were used to personalise models using our PROFILE framework (*Béal et al., 2019*) and using mutation data as discrete data and RNA as continuous data (*Appendix 1—figure 21*).

We simulated the prostate cell line-specific models under random initial conditions and observed that they generated distinctive phenotype probabilities and captured some of the differences described in literature (*Appendix 1—figure 21* and *Appendix 1—figure 22*). For instance, it has been described that PC-3 cell line has high migratory potential compared to DU-145 cells, which have a moderate migratory potential, and to LNCaP cells, which have low migratory potential (*Cunningham and You, 2015*). In our simulations, we capture that PC-3 has greater invasiveness, migration and proliferation than DU-145. However, the invasiveness and proliferation potential of LNCaP is much higher than PC3. Note that these results come from a collection of datasets from GDSC and a Boolean model that includes a subset of the interactions of 312 proteins. Distortions from real-life behaviour are expected and will be the focus of further research, such as the high LNCaP invasiveness or the lack of difference of the benign cell lines (BPH-1 and PWR-1E) with the rest of the cell lines.

As we did for the TCGA patients' study, we tried different personalisation recipes to personalise these cell lines, but as they had no associated clinical features, we were left with the comparison of the different values for the model's outputs among the recipes. We chose the aforementioned recipe as it included two different data types (RNAseq and mutations) and reproduced desired results (*Appendix 1—figure 21* and *Appendix 1—figure 22*). Nevertheless, we could have considered using mutation and CNA as discrete data and RNA as continuous data, but the inclusion of CNA data forced LNCaP proliferation to be zero (*Appendix 1—figure 23*). This is due to the fact that CNA data used as discrete data forces several nodes to be active or inactive throughout the simulation, as if they were mutants. Notably, CNA data forces E2F1 node to be 0, which forces Cyclin B to be 0 and it forces SMAD node to be 1, which forces MYC_MAX node to be 0 and p21 node to be 1, forcing Cyclin D to be 0. Without either Cyclin B or D, the model cannot activate the *Proliferation* node.

In addition, we wanted to study these different personalisation recipes to try to better match simulated phenotypes and cell line phenotypes described experimentally, but we had similar results (Appendix *Appendix 1—figure 23*). Furthermore and due to the mismatches of cell line models with their described biology characteristics, we went back to the source data to study if these mismatches were something we could correct on the model or a problem of the dataset we used to personalise the model. We performed principal component analysis (PCA, using *FactoMineR* R package) (*Lê et al., 2008*; *Appendix 1—figure 24*) on the dataset used to personalise the models: an RNAseq dataset of 111 genes. We found that the cell lines do not cluster by their characteristics: DU-145, an invasive cell line, is close to BPH-1 and PWR-1E, non-invasive cell lines.

Furthermore, we digged into the pathways that are characteristic of each of these cell lines by using single sample GSEA (using *ssGSEA* 2.0 R package) (*Krug et al., 2019*; *Appendix 1—figure*

*25*) on the same RNA dataset using the Hallmarks molecular signatures. We found that out of the 50 Hallmarks, 21 have an overlap of more than five genes with the model's genes. Thus, we set to cluster the cell lines by using the signatures of each one of them in these 21 pathways. The results are quite telling of the lack of clear clustering of these cell lines with their different characteristics (Appendix *Appendix 1—figures 24 and 25*): invasive and non-invasive cell lines have similar signature values in EMT or G2M checkpoint pathways, BPH-1 clusters with NCI-H660 and PWR-1E with DU-145, etc.

All in all, it is unrealistic to expect that a model of different cellular behaviours will match all biological aspects and characteristics as models are, by definition, abstractions of reality (*Rosenblueth and Wiener, 1945*; *Korzybski, 1995*). For instance, if one were to match the cell lines' doubling times, of which *Proliferation* phenotype should be a good proxy (*St John et al., 2012*; *Cunningham and You, 2015*), such a study would need a deeper understanding of the cell's biology, the modelling of many more processes, with many more parameters, and a more complete simulation framework both multi-scaled and finer-grained, which is beyond the scope of the present work.

All the cell line-specific personalised models are publicly available in *Supplementary file 5*.

## 6. Personalised LNCaP Boolean model

LNCaP model was selected to study its genetic interactions and its uses for drug discovery. The simulation of the LNCaP-specific model under random initial conditions leads to four most probable phenotypes: *Invasion-Migration*, *Invasion-Migration-Proliferation*, *Invasion-Proliferation* and *Invasion*. Using MaBoSS software, we were able to assign probabilities to each one of these phenotypes (*Appendix 1—figure 26* and *Supplementary files 1 and 2*).

Additionally, we studied the LNCaP model under four different growth conditions that could be reproduced in experiments. These are a nutrient-rich media that mimics the RPMI supplemented with glucose and foetal bovine serum with additional androgen, EGF, both or none (*Appendix 1—figure 27*).

### 6.1. High-throughput mutant analysis of LNCaP model

A mutant in the logical framework is simulated by setting the node corresponding to the gene mutated to 0 in the case of loss of function and to 1 in the case of gain of function. The effect of a mutation is assessed, likewise to the change of initial conditions, by comparing the mutant's probability of reaching a phenotype with respect to the wild type model. Therefore, mutations change the model phenotypes' probabilities and this can be compared to the wild type model.

The logical model allows us to easily simulate all possible combinations of mutations and study the potential redundancy or synergy of these perturbations. To perform this, tools like our high-throughput mutant analysis pipeline (*Montagud et al., 2019*) are ideally suited. This pipeline of tools was applied to the LNCaP-specific model in order to study all single and double mutants of the LNCaP model (32,258 mutants) and their probabilities of reaching all the phenotypes of the model.

The double mutants of the high-throughput mutant analysis were used to identify genetic interaction relationships, such as epistasis, among the single mutants. Phenotype probabilities' variations of all 32,258 models were compared to the wild type model and were used to identify relevant combinations of perturbations that affect phenotypes of interest (*Appendix 1—figure 28*) and single phenotypes (*Appendix 1—figure 29*). In these figures, a PCA was applied to the matrix of the seven phenotype probabilities of the 32,258 mutants and was then normalised with the PCA values of the wild type. The result is a PCA centred around the wild type using the phenotypes as variables, where the distance between a given point and the wild type orange point at the centre is representative of the distance in the phenotype scores among them.

We were particularly interested in identifying knock-out (KO) and over-expression (OE) mutants that depleted *Proliferation* and/or increased *Apoptosis* with regard to the wild type LNCaP model. Using MaBoSS, we were able to quantify and rank the effect of all the 32,258 mutants on the probabilities of reaching *Proliferation* and *Apoptosis* (*Supplementary file 6*).

The double mutants that mostly depleted *Proliferation* were combinations of p21_oe, MXI1_oe, HIF1_oe, AR_ko and E2F1_ko. Likewise, the double mutants that mostly increased *Apoptosis* were combinations of GLI_oe, Caspase3_oe, Caspase8_oe, Caspase9_oe and PTCH1_oe. The single mutants that mostly depleted *Proliferation* were HIF_oe, MXI_oe, p21_oe, Caspase3_oe and Caspase8_oe. Likewise, those that mostly increased *Apoptosis* were GLI_oe, Caspase3_oe, Caspase8_oe, Caspase9_oe and SMO_oe

It was in our interest to identify drugs that could inhibit some of these genes, thus, we filtered these lists to find the best single KO mutations. We found that the single KO that mostly depleted *Proliferation* were AR_ko, VHL_ko, AKT_ko, E2F1_ko, PIP3_ko, EGFR_ko, PI3K_ko, CDH2_ko, TWIST1_ko, ERK_ko. Likewise, the single KO that mostly increased *Apoptosis* were AKT_ko, AR_ko, ERK_ko, cFLAR_ko, SPOP_ko, PIP3_ko, PI3K_ko, EGFR_ko, HSPs_ko and ATR_ko. Another knockout, p53_ko, was identified in our analysis, but was later discarded upon closer analysis. From topological analyses, p53 deletion should increase *Proliferation*, as p21, a cyclin inhibitor, is therefore not transcribed. Nevertheless, p53 has a dual effect on *Apoptosis* in the network: p53 activates CytC and Apaf1, which activate *Apoptosis*, but p53 also inhibits BIRC5, an activator of *Apoptosis*. The model should be closely inspected to correct this mismatch in future works. In any case, the effects of p53's mutations are not further analysed in present work, nor their results are further discussed.

We gathered the 20 top nodes from each of those lists and ended with 29 nodes that could be knocked out to deplete *Proliferation* and/or increase *Apoptosis* (AKT, AR, ATR, AXIN1, Bak, BIRC5, CDH2, cFLAR, CyclinB, CyclinD, E2F1, eEF2, eEF2K, EGFR, ERK, HSPs, MED12, mTORC1, mTORC2, MYC, MYC_MAX, PHDs, PI3K, PIP3, SPOP, TAK1, TWIST1, VHL). We used this ranking, the genes corresponding to these nodes and known drugs that target these genes to shortlist potential therapeutic target candidates tailored to LNCaP cell line (Main text, *Table 1*).

## 6.2. Robustness analysis of the logical model

We performed a perturbation on the logical rules stability of the LNCaP model, following our previous work (*Montagud et al., 2019*). In Section 6.1 we forced the value of a node to be 0 or 1 throughout the simulation. Now, we have changed one and two logical gates from each logical rule of the LNCaP model and studied the effects on the phenotype scores. In short, we have changed an AND in OR gate and vice versa in each logical rule (what we call level 1 analysis with 372 simulations in this model) or twice in the same rule (level 2 analysis with 1,263 simulations in this model).

Overall, we see that all of the most probable phenotypes (as the ones from *Appendix 1—figure 28*) are very robust to this kind of perturbation. Even the less stable phenotype, *Invasion-Migration-Proliferation*, only has 2.69% of the single (level 1) perturbations that reduce this phenotype's probability to zero (*Appendix 1—figure 30A*) and 3.33% of the double (level 2) perturbations (*Appendix 1—figure 30B*). Most of these perturbations were focused on HIF1 and AR_ERG nodes for single perturbations (*Appendix 1—figure 31A*) and HIF1 and p53 nodes for double perturbations (*Appendix 1—figure 31B*).

# 7. Drug studies in prostate cell lines

## 7.1. Drugs associated to genes included in the model

We tested if the drugs that targeted the genes included in the model allowed us to identify cell line specificities. We analysed drug sensitivity data from GDSC1 and GDSC2 studies (*Iorio et al., 2016*) and for each drug we calculated a normalised sensitivity of the eight prostate cell lines considered in present study (22RV1, BPH-1, DU-145, LNCaP-Clone-FGC, NCI-H660, PC-3, PWR-1E, and VCaP). We normalised drug sensitivity across cell lines in the following way: cells were ranked from most sensitive to least sensitive using ln(IC50) as the drug sensitivity metrics, and the rank was divided by the number of cell lines tested with the given drug. Thus, the most sensitive cell line scored 0, while the most resistant cell line scored one normalised sensitivity. This rank-based metric made it possible to analyse all drug sensitivities for a given cell line, without drug-specific confounding factors, such as the mean IC50 of a given drug or others.

We observed that cell lines described as resistant (DU-145 and PC-3) have a skewed distribution towards least sensitive values (*Appendix 1—figure 32D and E*), while cell lines such as LNCaP have a skewed distribution towards more sensitive values (*Appendix 1—figure 32A*). Meaning that the drugs that target the genes in the personalised model are not very effective against the resistant cell lines, but that LNCaP is significantly more sensitive to these. Additionally, we found that BPH-1 is generally sensitive to all drugs, let them be model-specific or not (*Appendix 1—figure 32C*). For the other cell lines, there is no significant difference between model-specific drugs or not.

In addition, we performed a target set enrichment analysis using the *fgsea* method (*Korotkevich et al., 2016*) from the *piano* R package (*Väremo et al., 2013*). Again, we targeted pathway information from the GDSC1 and GDSC2 studies (*Iorio et al., 2016*) as target sets, and performed the enrichment analysis on the aforementioned normalised drug sensitivity profile of the LNCaP cell

line. This target enrichment analysis showed that LNCaP cell lines are especially sensitive to PI3K/AKT/MTOR, hormone related (AR targeting) and Chromatin targeting (bromodomain inhibitors, regulating Myc) drugs (*Appendix 1—table 2*, adjusted p-values from target enrichment: 0.001, 0.001 and 0.037, respectively), which corresponds to the model predictions (Main text, *Table 1*).

**Appendix 1—table 2.** Target enrichment for LNCaP-specific drug sensitivities.
Drugs were sorted based on rank normalised drug sensitivity 0: most sensitive, 1 most resistant, based on GDSC AUC drug sensitivity metric for LNCaP. Target pathway enrichment analysis was performed based on the pathway membership of drug targets. Direction represents whether pathway-targeting drugs were enriched in sensitive or resistant drugs.

| Drug target pathway | p-value | adj. p-value | Direction |
| --- | --- | --- | --- |
| **PI3K/MTOR signalling** | 0.00011563 | **0.0011106** | sensitive |
| **Hormone-related** | 0.00014808 | **0.0011106** | sensitive |
| **Chromatin other** | 0.0065661 | **0.03283** | sensitive |
| **Chromatin histone methylation** | 0.01216 | **0.045601** | sensitive |
| p53 pathway | 0.079554 | 0.23866 | sensitive |
| DNA replication | 0.10466 | 0.26164 | sensitive |
| WNT signalling | 0.13583 | 0.29107 | sensitive |
| Unclassified | 0.20391 | 0.38233 | sensitive |
| Genome integrity | 0.54186 | 0.90311 | sensitive |
| Cytoskeleton | 0.63153 | 0.93981 | sensitive |
| Other, kinases | 0.81647 | 0.93981 | sensitive |
| RTK signalling | 0.85985 | 0.93981 | sensitive |
| Other | 0.87572 | 0.93981 | sensitive |
| Protein stability and degradation | 0.88166 | 0.93981 | sensitive |
| EGFR signalling | 0.93981 | 0.93981 | sensitive |
| Apoptosis regulation | 0.96036 | 0.96036 | resistant |
| Chromatin histone acetylation | 0.73164 | 0.83616 | resistant |
| JNK and p38 signalling | 0.63484 | 0.83616 | resistant |
| IGF1R signalling | 0.23538 | 0.37662 | resistant |
| Cell cycle | 0.19382 | 0.37662 | resistant |
| Metabolism | 0.053352 | 0.14227 | resistant |
| Mitosis | 0.027536 | 0.11014 | resistant |
| **ERK MAPK signalling** | 0.00050075 | **0.004006** | resistant |

## 7.2. Drugs associated to the proposed targets of LNCaP

We wanted to test if the LNCaP cell line is more sensitive than the rest of the prostate cell lines to the LNCaP-specific drugs identified in *Table 1* from the main text. We compared GDSC's Z-score of these drugs in LNCaP with their Z-scores in all GDSC cell lines (*Appendix 1—figure 5*). We observed that LNCaP is more sensitive to drugs targeting AKT or TERT than the rest of the studied prostate cell lines. In *Appendix 1—figure 33*, we can observe that trend in comparison to the other prostate cell lines and to the rest of the GDSC cell lines. In addition, we see that AKT sensibility in LNCaP is one of the highest in the GDSC records.

## 7.3. Gradual inhibition of genes in LNCaP model

Logical models can be used to simulate the effect of therapeutic interventions by using our PROFILE_v2 methodology. For this, we can take advantage of MaBoSS as it can perform simulations using a population of trajectories by changing the proportion of activated and inhibited status of a given

node. Using MaBoSS method (see Section 2.1), initial values of each node can be defined with a continuous value between 0 and 1 representing the probability for the node to be defined as 1 for each new trajectory. This can be determined in the configuration file of each model (see, for instance, 'istate' section of the CFG files in the *Supplementary files 13 and 5*). For instance, out of 5,000 trajectories of the Gillespie algorithm, MaBoSS can simulate 70% of them with an activated AKT and 30% with an inhibited AKT node. The phenotypes' probabilities for the 5,000 trajectories are averaged and these are considered representative of a model with a drug that inhibits 30% of the activity of AKT.

All these inhibitions were performed using our PROFILE_v2 framework (https://github.com/ArnauMontagud/PROFILE_v2) that allow to study the effect of single and double mutations (knock-out and overexpression) in the phenotypes' probabilities using MaBoSS as well as to study the Bliss Independence synergy score of these combinations.

### 7.3.1. Single inhibitions
We studied the variations of all the phenotype scores upon the 17 nodes' inhibitions under EGF growth condition (*Appendix 1—figure 34*) and under AR, EGF, 00 and AR_EGF growth conditions (*Appendix 1—figure 35*).

### 7.3.2. Double inhibitions
Thoroughly, we studied the effect of the inhibition of the 17 combined nodes under EGF growth condition in the *Proliferation* (*Appendix 1—figure 36*) and *Apoptosis* phenotype score (*Appendix 1—figure 37*).

This combined scores allowed us to study the Bliss Independence synergies scores and their variations in these combined nodes' inhibitions under EGF growth conditions. We studied *Proliferation* (*Appendix 1—figure 38*) and *Apoptosis* phenotypes (*Appendix 1—figure 39*).

## 8. Analyses of drug experiments
We present the dose-dependent changes in the LNCaP cell line growth upon drug addition of Hsp90 (*Appendix 1—figure 40*) and PI3K/AKT inhibitors (*Appendix 1—figure 41*) with insets to show the cytotoxicity assay results at 24, 48, and 72 hr after drug addition.

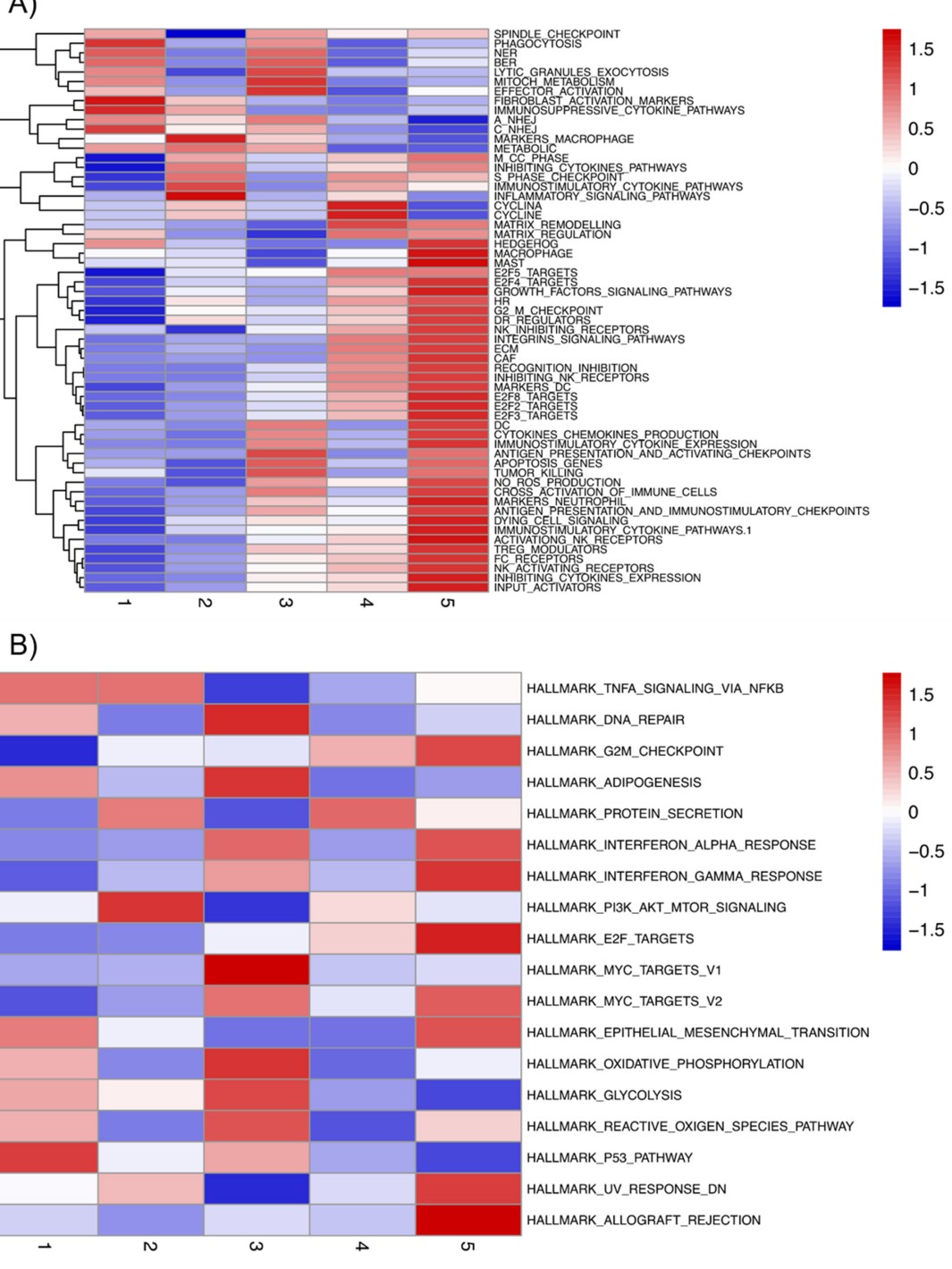

**Appendix 1—figure 1.** Mean activities by subgroups for gene modules defined from pathways described in ACSN. (**A**) And in Hallmarks' gene sets (**B**) and that are significantly overdispersed over all samples. Blue indicates low pathway activity, red indicates high pathway activity.

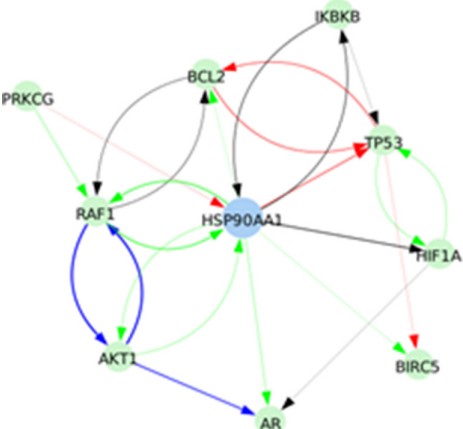

**Appendix 1—figure 2.** Signed directed interactions between HSP90AA1 and nodes already taken into account in the model.

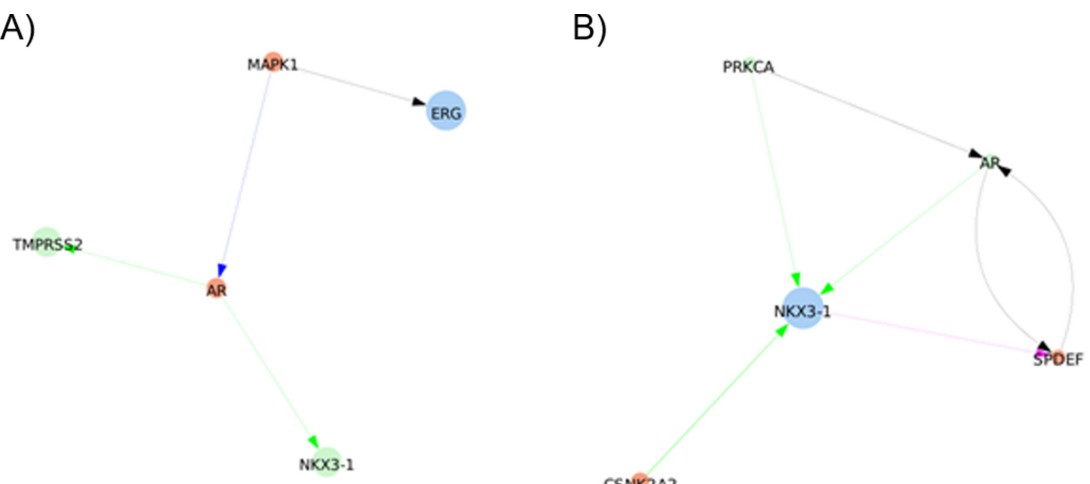

**Appendix 1—figure 3.** shortest paths found between ERG and TMPRSS2 or NKX3-1 by Pypath: no direct interaction is found.

A)

```
A = B AND (C OR E)
B = (A OR C) AND NOT D
C = A AND NOT (D AND E)
D = D
E = E
```

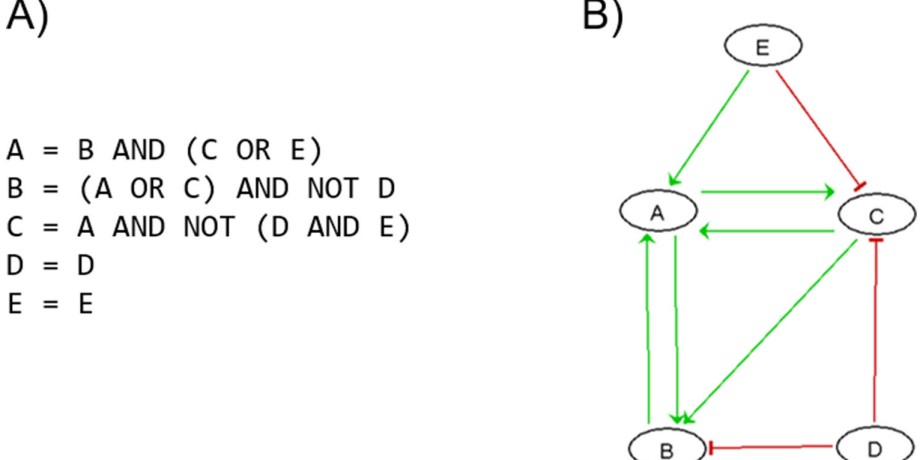

**Appendix 1—figure 4.** Boolean toy model to showcase different examples of Boolean formulas.

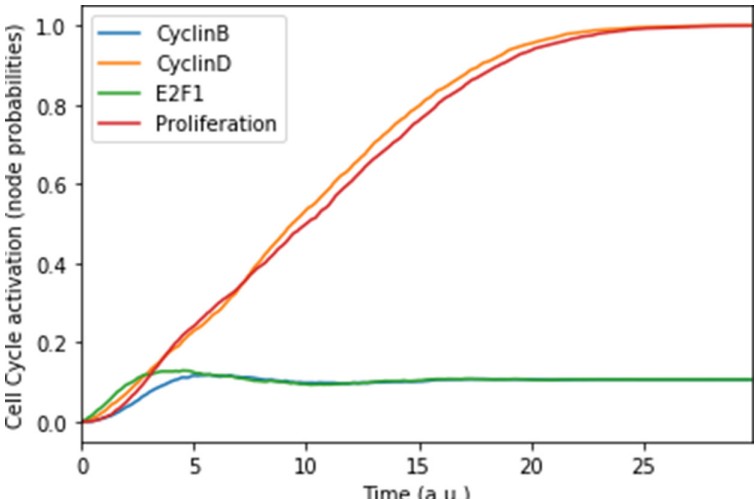

**Appendix 1—figure 5.** Mean probabilities of the nodes characterising the cyclins and proliferation, with nutrients and growth factors as inputs. We choose initial states for the nodes involved in the cell cycle that correspond to quiescence (cyclins OFF, cell cycle inhibitors Rb and p21 ON), in order to visualise the order of activation of the cyclins: first Cyclin D, then Cyclin B. The mean probabilities reach asymptotic levels because of the desynchronisation of stochastic trajectories in the population.

A)

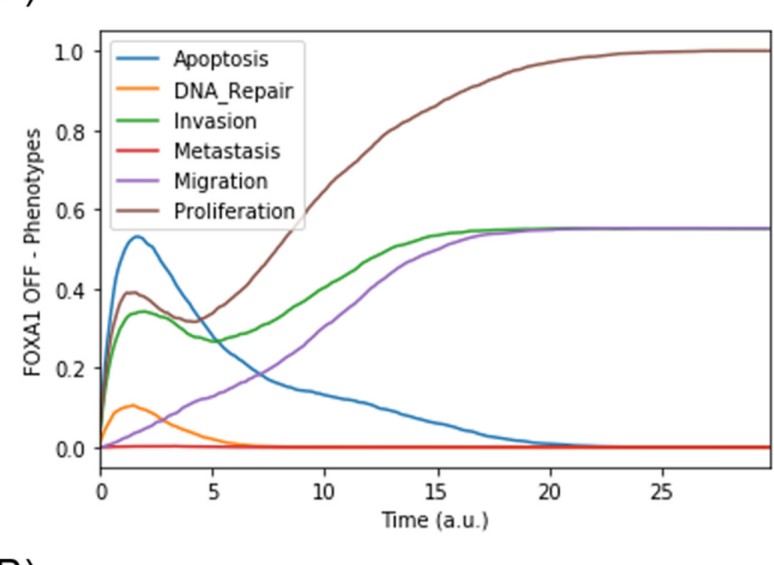

B)

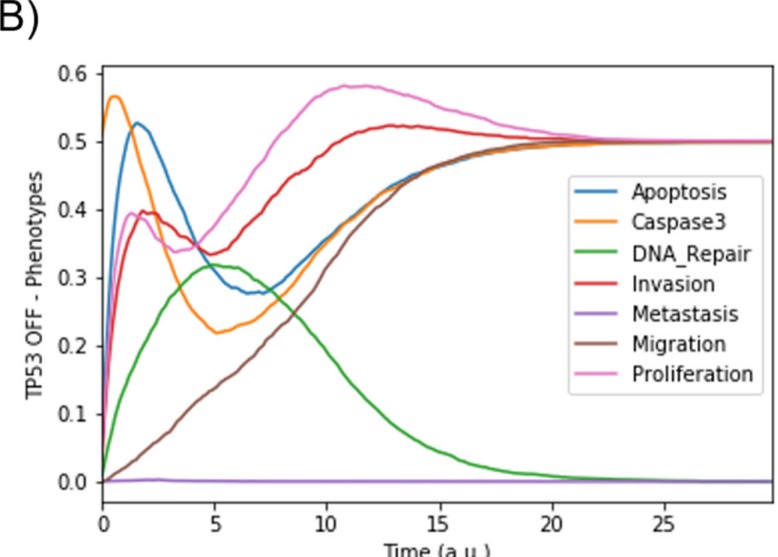

**Appendix 1—figure 6.** Mean probabilities in simulations of mutated models. (**A**) Loss-of-function mutation of FOXA1. (**B**) Loss-of-function mutation of TP53.

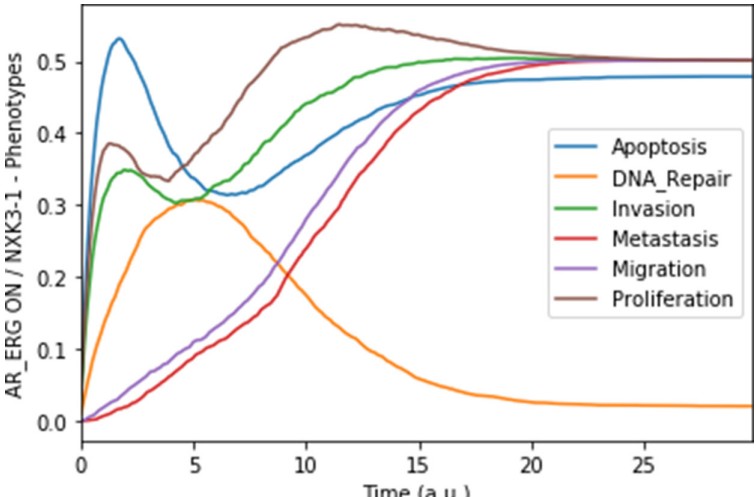

**Appendix 1—figure 7.** Mean probabilities in simulations of the model with a multiple simulation: the gene fusion TMPRSS2:ERG and a loss-of-function of NKX3-1.

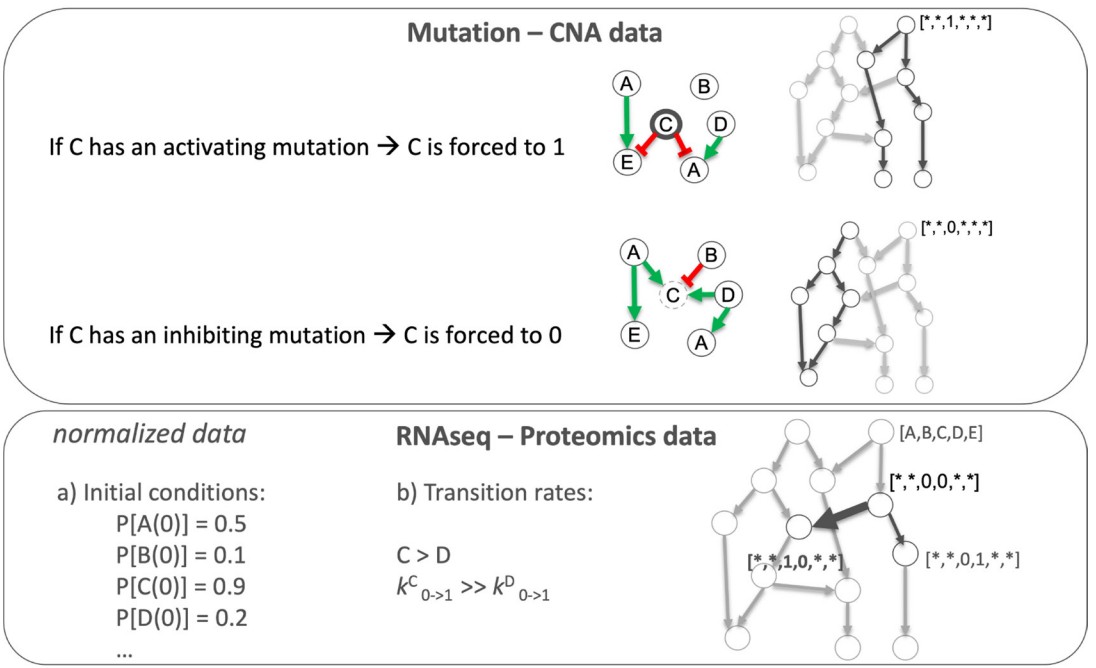

**Appendix 1—figure 8.** Data integration in Boolean models to have personalised Boolean models.

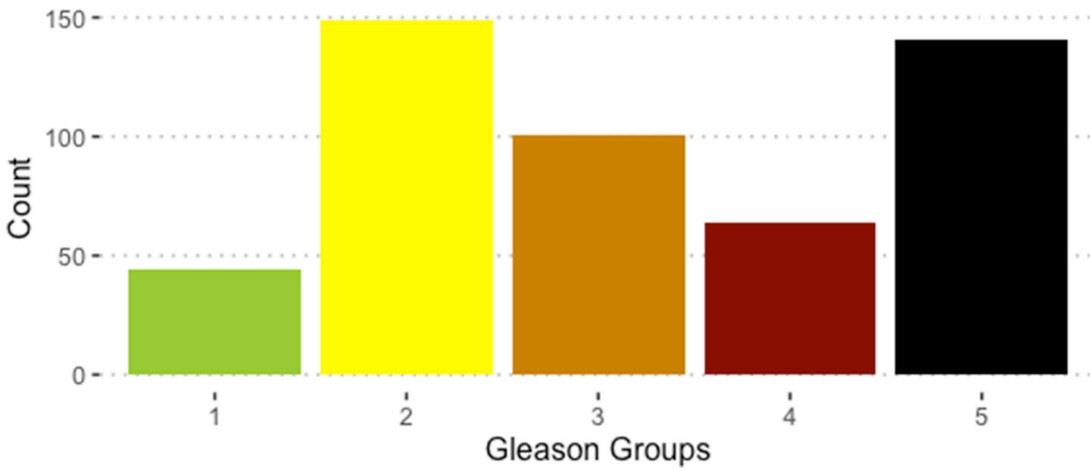

**Appendix 1—figure 9.** Distribution of 488 TCGA prostate cancer patients' samples per Gleason grade.

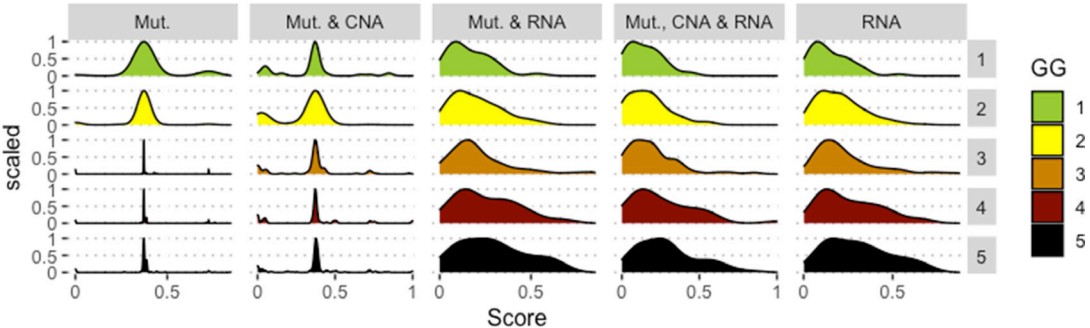

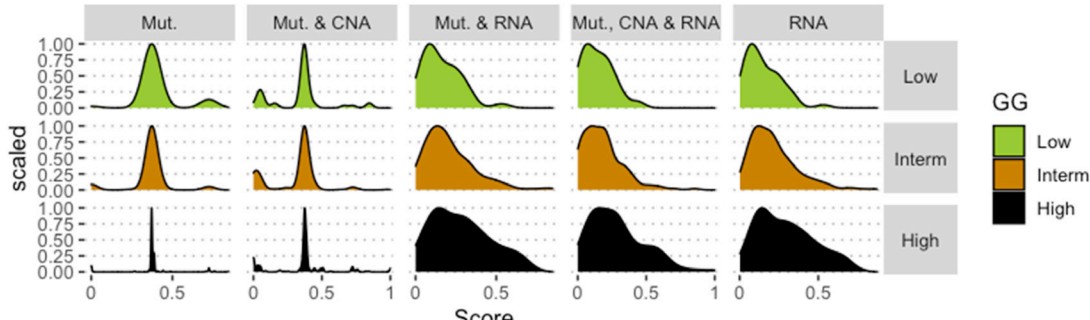

**Appendix 1—figure 10.** Associations between simulations and Gleason grades (GG). Distribution histograms of *Apoptosis* scores according to GG in three groups (**A**) and five groups (**B**). Columns correspond to different personalisation recipes (see ***Béal et al., 2019*** for more details). We found that across 3-stage GG Kruskal-Wallis rank sum test is significant for *Apoptosis* under the 'Mut, CNA, and RNA' recipe (p-value = 2.83E-6) and significant across 5-stage GG (p-value = 1.88E-5). Additionally, we used Dunn's test to identify which pairs of groups are statistically different focusing on the 3-stage GG and found that grade High is statistically different from grades Low (Bonferroni's adjusted p-value = 3.3E-3) and Intermediate (Bonferroni's adjusted p-value = 9.47E-6).

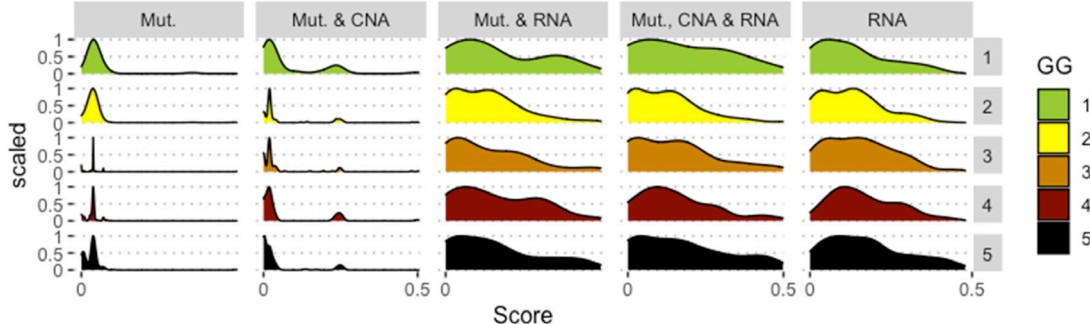

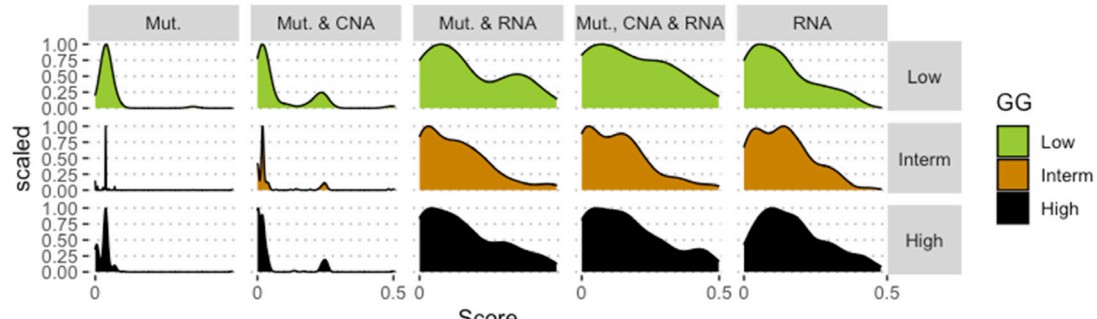

**Appendix 1—figure 11.** Associations between simulations and Gleason grades (GG). Distribution histograms of *DNA_repair* scores according to GG in three groups (**A**) and five groups (**B**). Columns correspond to different personalisation recipes (see ***Béal et al., 2019*** for more details). Kruskal-Wallis rank sum test across 3-stage GG is neither significant for *DNA_Repair* under the 'Mut, CNA and RNA' recipe (p-value = 0.217) nor across 5-stage GG (p-value = 0.0995).

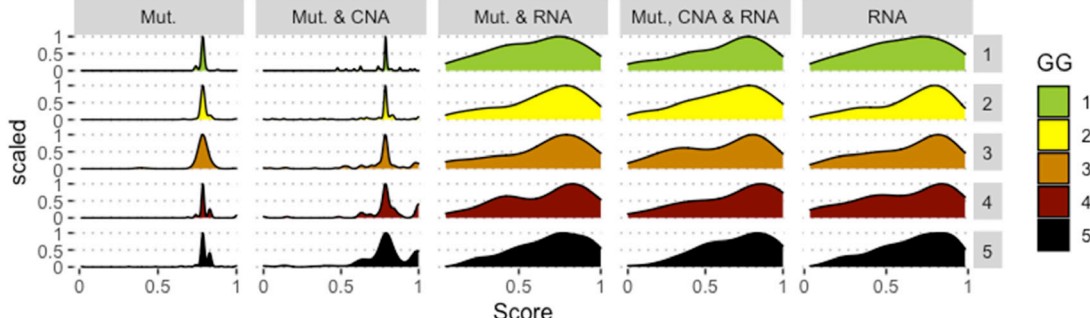

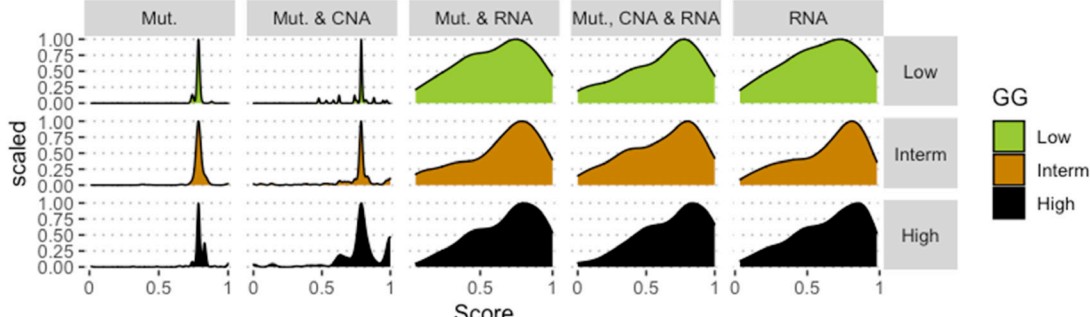

**Appendix 1—figure 12.** Associations between simulations and Gleason grades (GG). Distribution histograms of *Invasion* scores according to GG in three groups (**A**) and five groups (**B**). Columns correspond to different personalisation recipes (see ***Béal et al., 2019*** for more details). Kruskal-Wallis rank sum test across 3-stage GG is significant for *Invasion* under the 'Mut, CNA, and RNA' recipe (p-value = 0.0358), but not significant across 5-stage GG (p-value = 0.134). Using Dunn's test on the 3-stage GG, we found that grade High is statistically different from grade Intermediate (Bonferroni's adjusted p-value = 0.037).

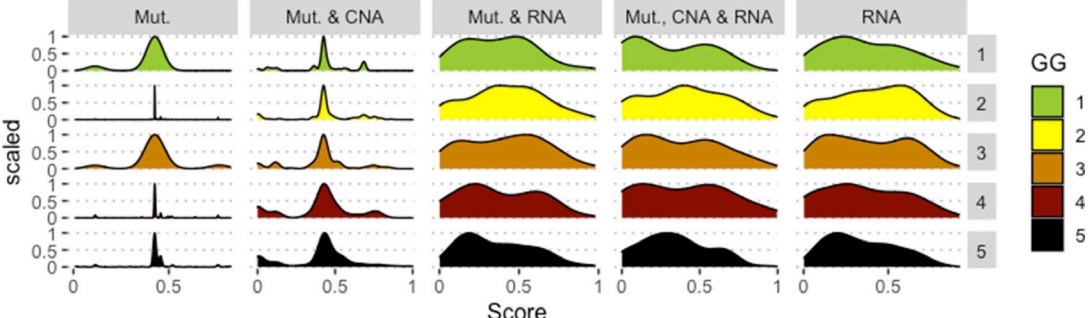

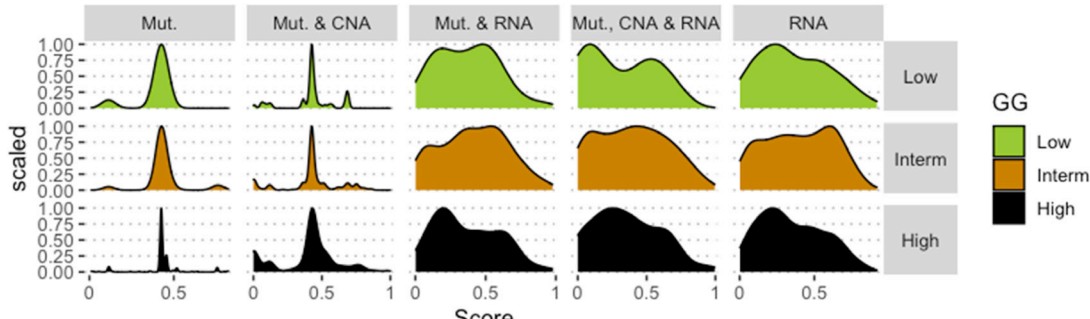

**Appendix 1—figure 13.** Associations between simulations and Gleason grades (GG). Distribution histograms of *Migration* scores according to GG in three groups (**A**) and five groups (**B**). Columns correspond to different personalisation recipes (see ***Béal et al., 2019*** for more details). Kruskal-Wallis rank sum test across 3-stage GG is neither significant for *Migration* under the 'Mut, CNA, and RNA' recipe (p-value = 0.173) nor across 5-stage GG (p-value = 0.275).

**A** Proliferation score in different simulation cases
(Classical 5-stage GG)

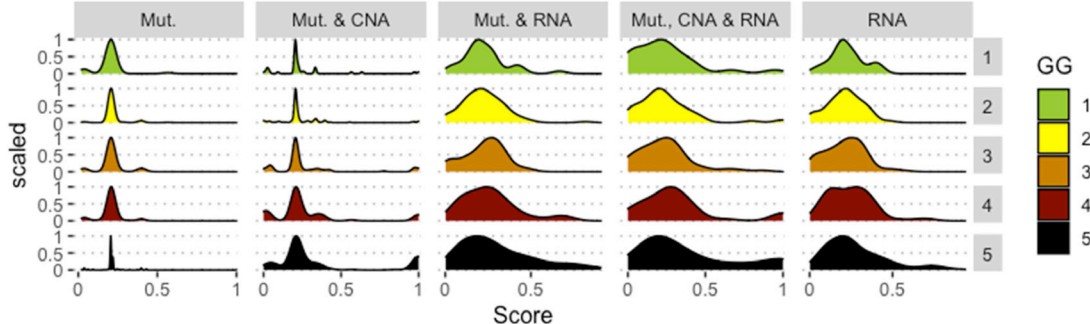

**B** Proliferation score in different simulation cases
(Merged 3-stage GG)

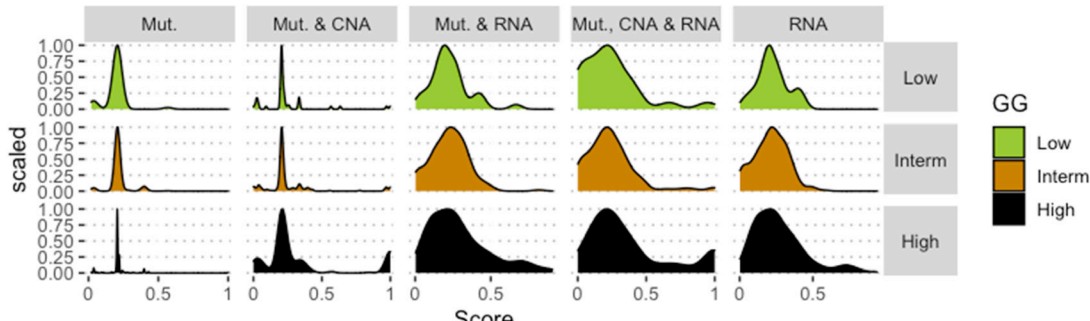

**Appendix 1—figure 14.** Associations between simulations and Gleason Grades (GG). Distribution histograms of *Proliferation* scores according to GG in three groups (**A**) and five groups (**B**). Columns correspond to different personalisation recipes (see *Béal et al., 2019* for more details). Kruskal-Wallis rank sum test across 3-stage GG is significant for *Proliferation* under the 'Mut, CNA, and RNA' recipe (p-value = 0.00207) and across 5-stage GG (p-value = 0.013). Using Dunn's test on the 3-stage GG, we found that grade High is statistically different from grade Intermediate (Bonferroni's adjusted p-value = 0.0023).

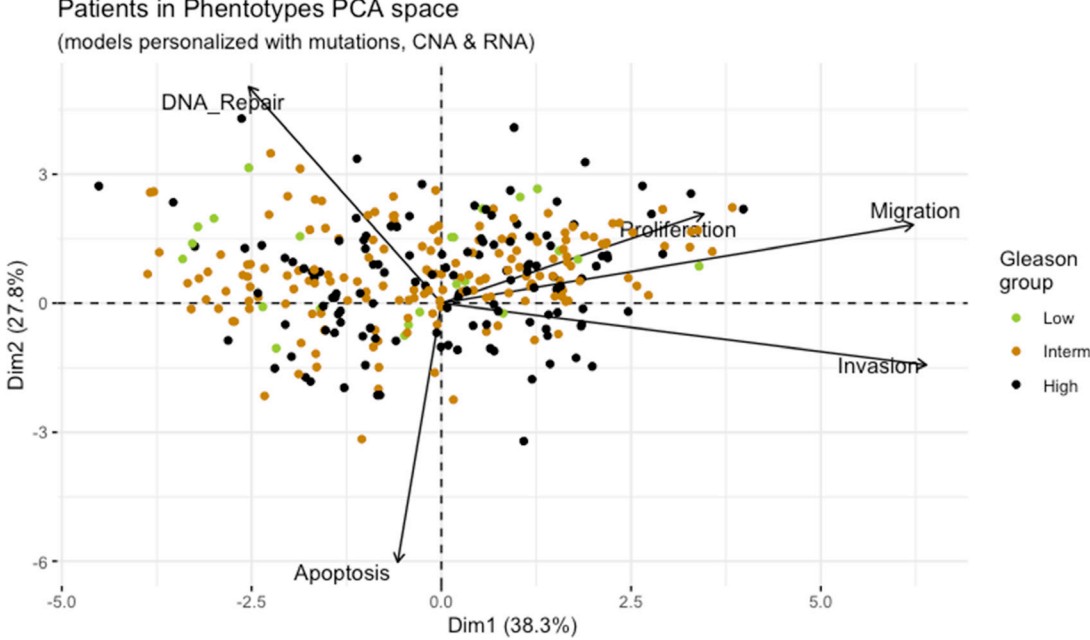

**Appendix 1—figure 15.** Principal Component Analysis of all 488 TCGA patients in 3 Gleason Grades using the vectors of all five phenotypes from the model.

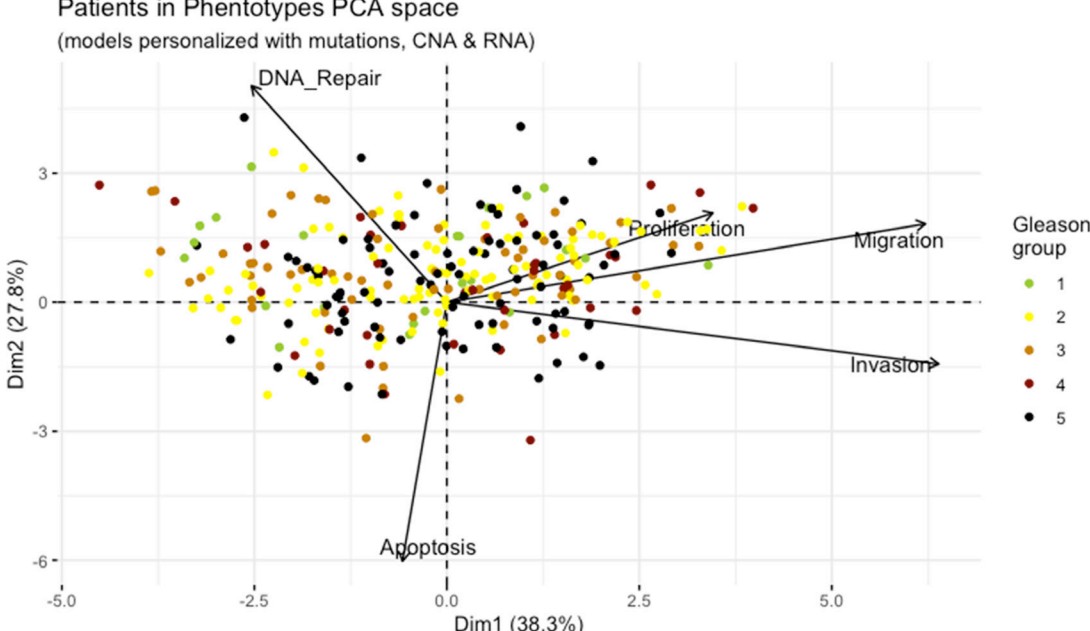

**Appendix 1—figure 16.** Principal Component Analysis of all 488 TCGA patients in 5 Gleason Grades using the vectors of all five phenotypes from the model.

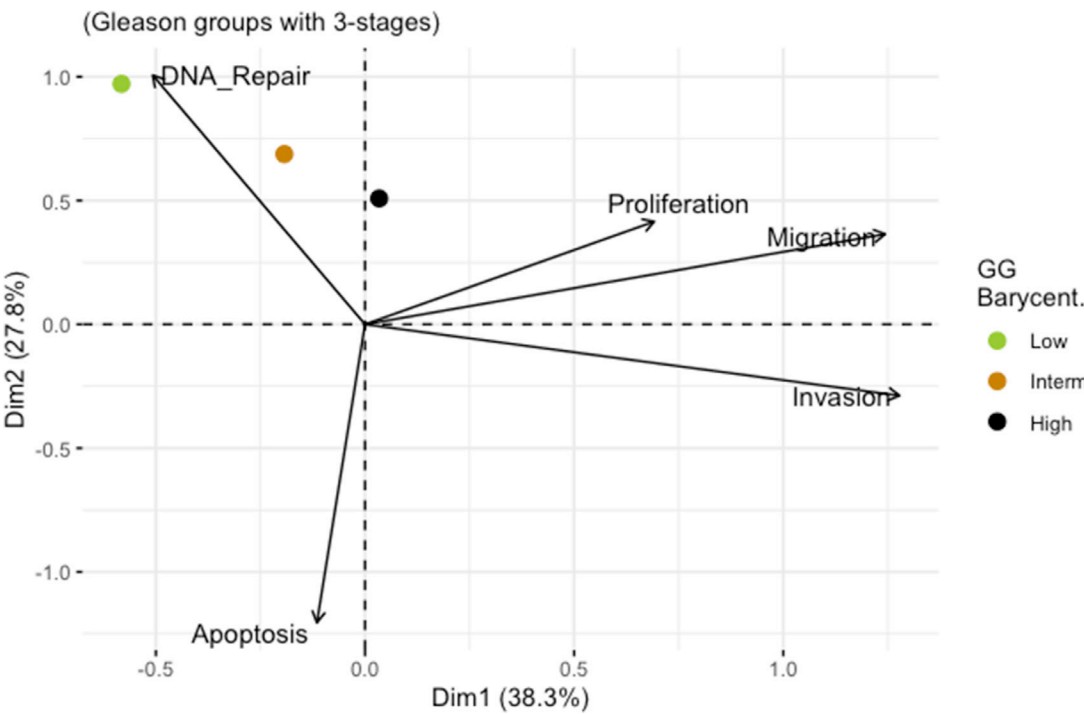

**Appendix 1—figure 17.** Principal Component Analysis' barycenters of all 488 TCGA patients grouped in 3 Gleason Grades using the vectors of all five phenotypes from the model. This is the same figure as *Appendix 1—figure 3* in the main text.

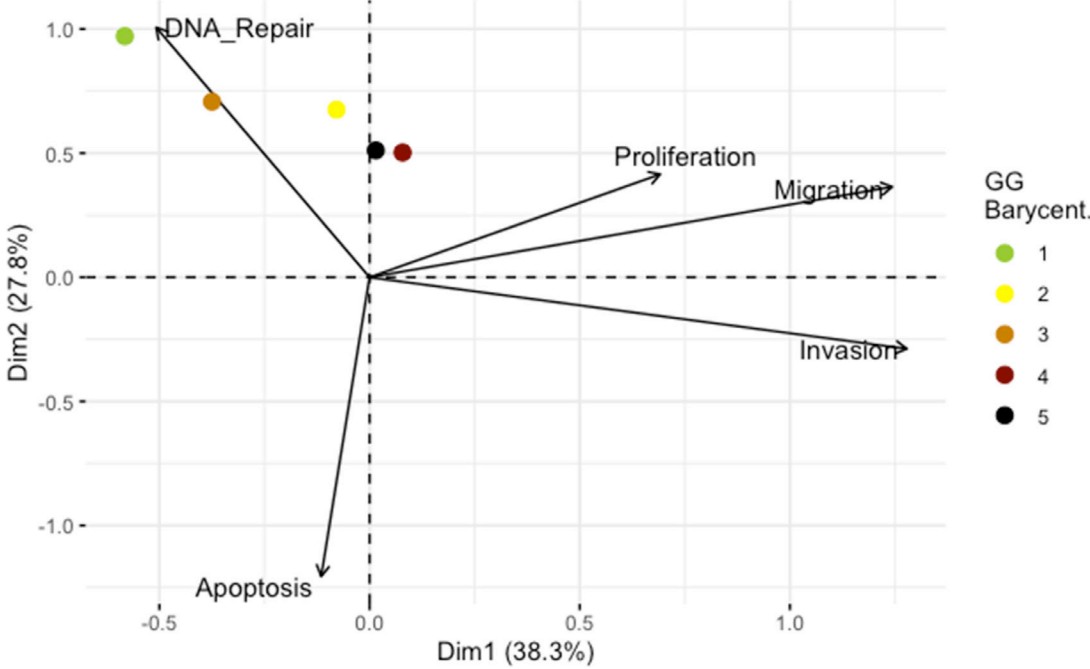

**Appendix 1—figure 18.** Principal Component Analysis' barycenters of all 488 TCGA patients grouped in 5 Gleason Grades using the vectors of all five phenotypes from the model.

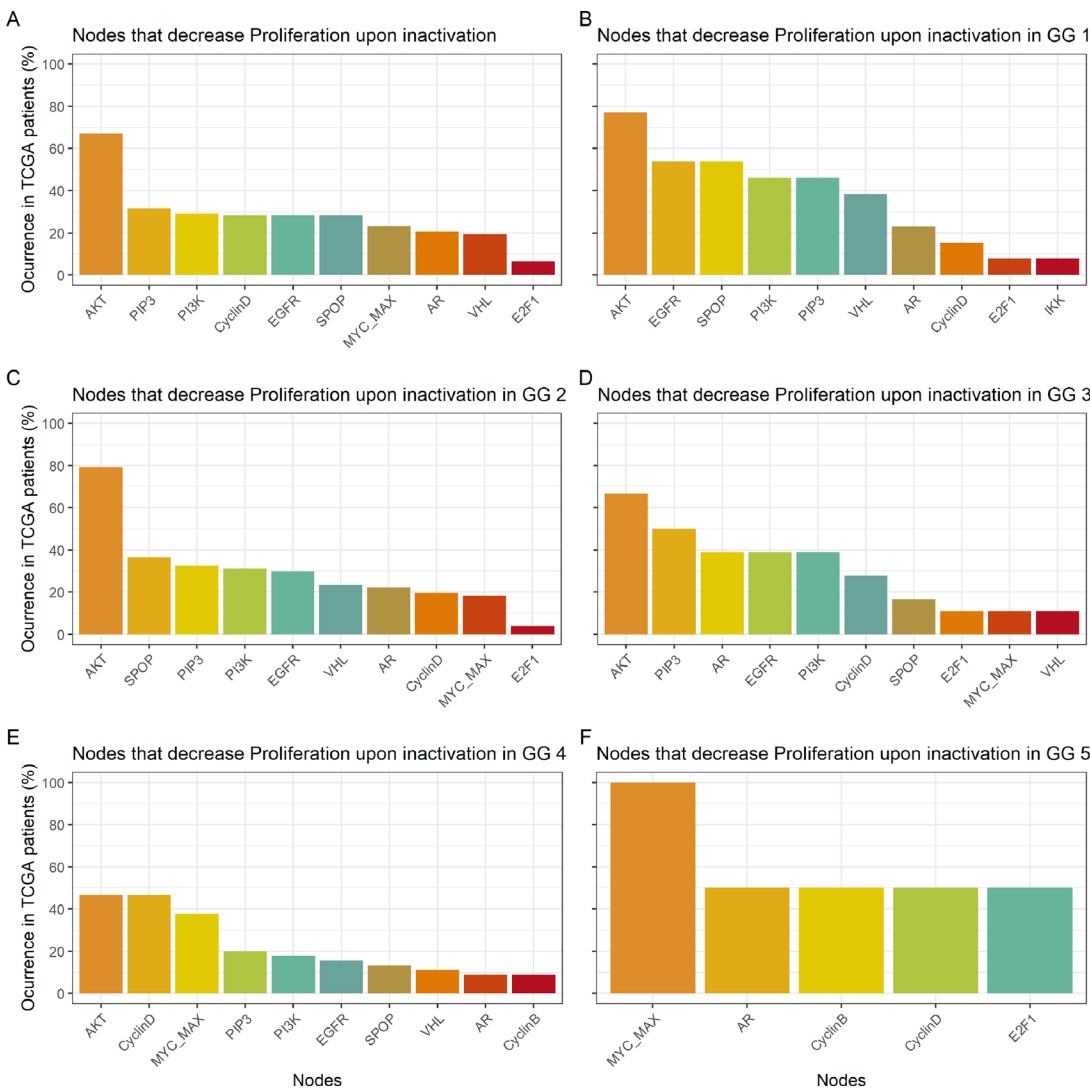

**Appendix 1—figure 19.** Nodes in the Boolean model that have a *Proliferation* value of at least 30% less the wild type value upon inactivation. (**A**) Nodes from aggregating all patient-specific results; (**B**) Nodes from patients from Gleason Grade 1; (**C**) Nodes from patients from Gleason Grade 2; (**D**) Nodes from patients from Gleason Grade 3; (**E**) Nodes from patients from Gleason Grade 4; (**F**) Nodes from patients from Gleason Grade 5.

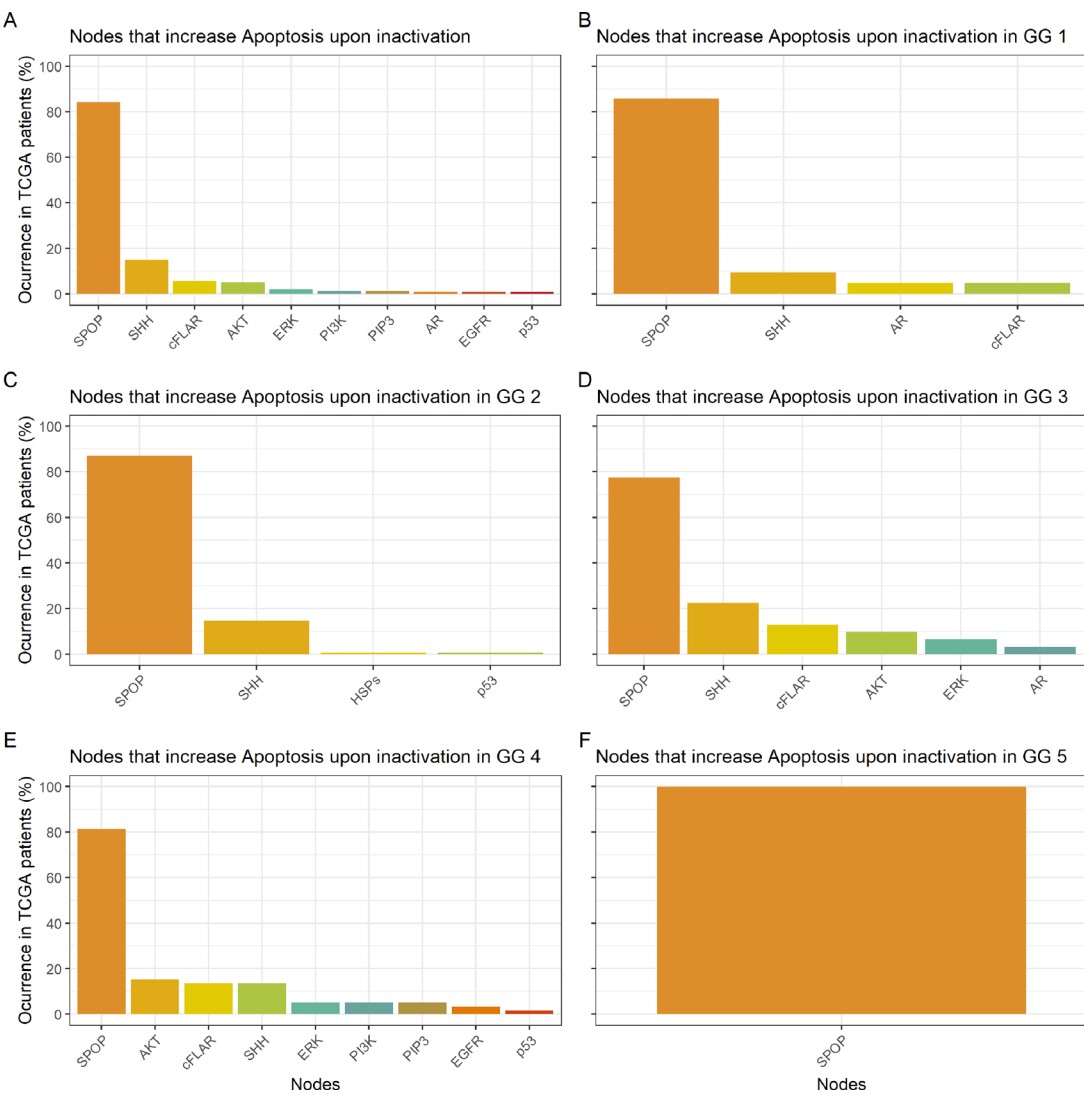

**Appendix 1—figure 20.** Nodes in the Boolean model that promote *Apoptosis* at least 30% more than the wild type value upon inactivation. (**A**) Nodes from aggregating all patient-specific results; (**B**) Nodes from patients from Gleason Grade 1; (**C**) Nodes from patients from Gleason Grade 2; (**D**) Nodes from patients from Gleason Grade 3; (**E**) Nodes from patients from Gleason Grade 4; (**F**) Nodes from patients from Gleason Grade 5.

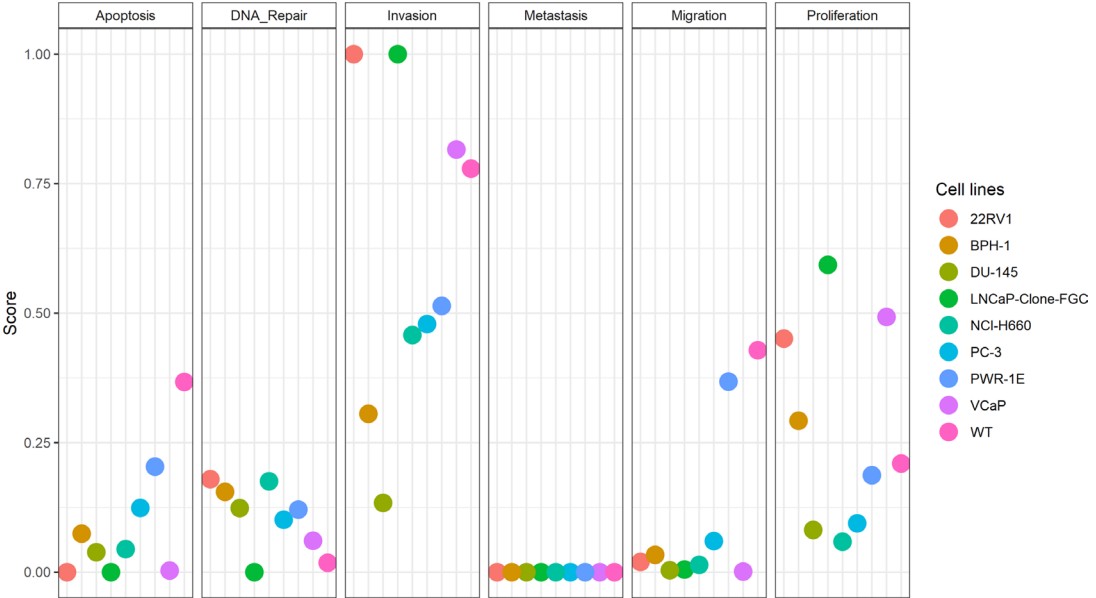

**Appendix 1—figure 21.** Phenotype simulation results across GDSC prostate cell line-specific Boolean models' simulation with random initial conditions. WT stands for wild type model, the original prostate model with no personalisation.

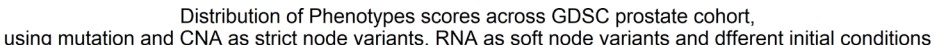

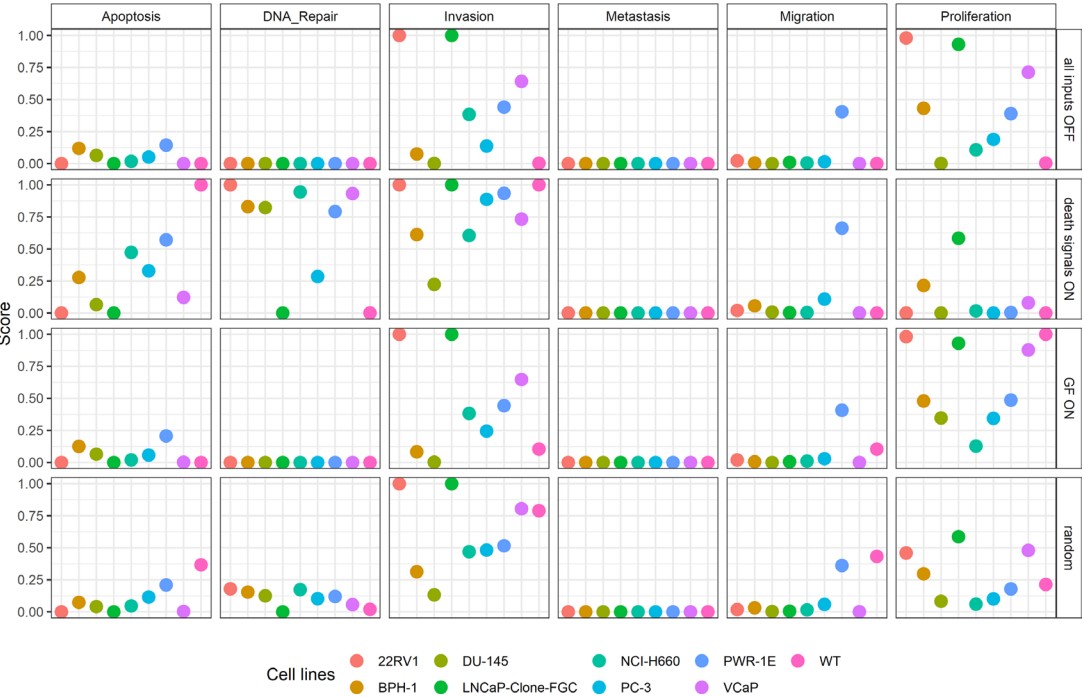

**Appendix 1—figure 22.** Phenotype simulation results across GDSC prostate cell line-specific Boolean models' simulation with different initial conditions. WT stands for wild type model, the original prostate model with no personalisation.

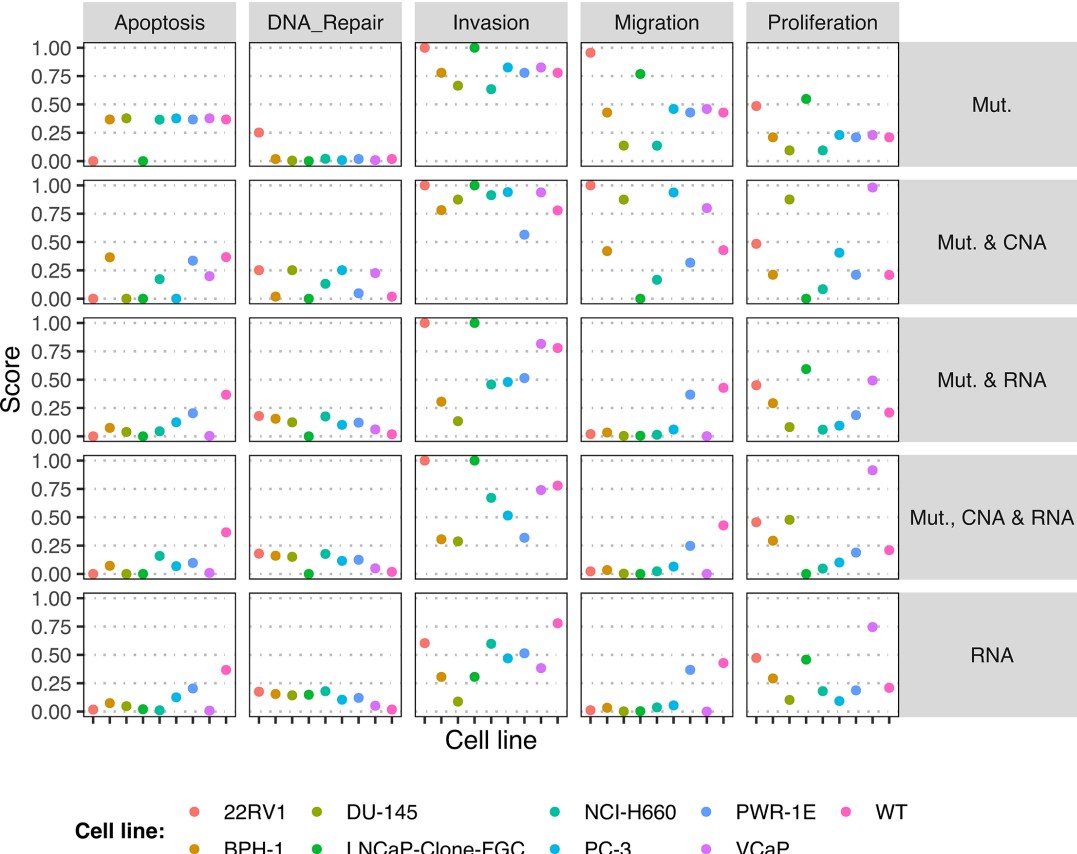

**Appendix 1—figure 23.** Phenotype simulation results across GDSC prostate cell line-specific Boolean models' simulation with random initial conditions under different personalisation recipes. Mutations and CNA are always considered as discrete data and RNA expression is always considered as continuous data.

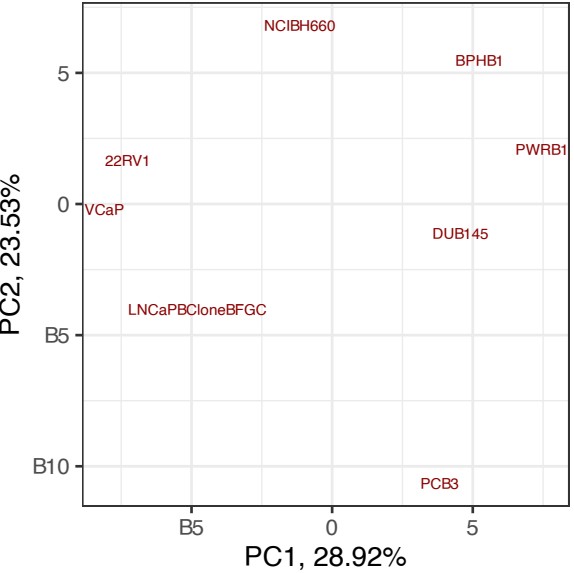

**Appendix 1—figure 24.** Principal Component Analysis (PCA) of the RNA dataset used to tailor the prostate cell lines.

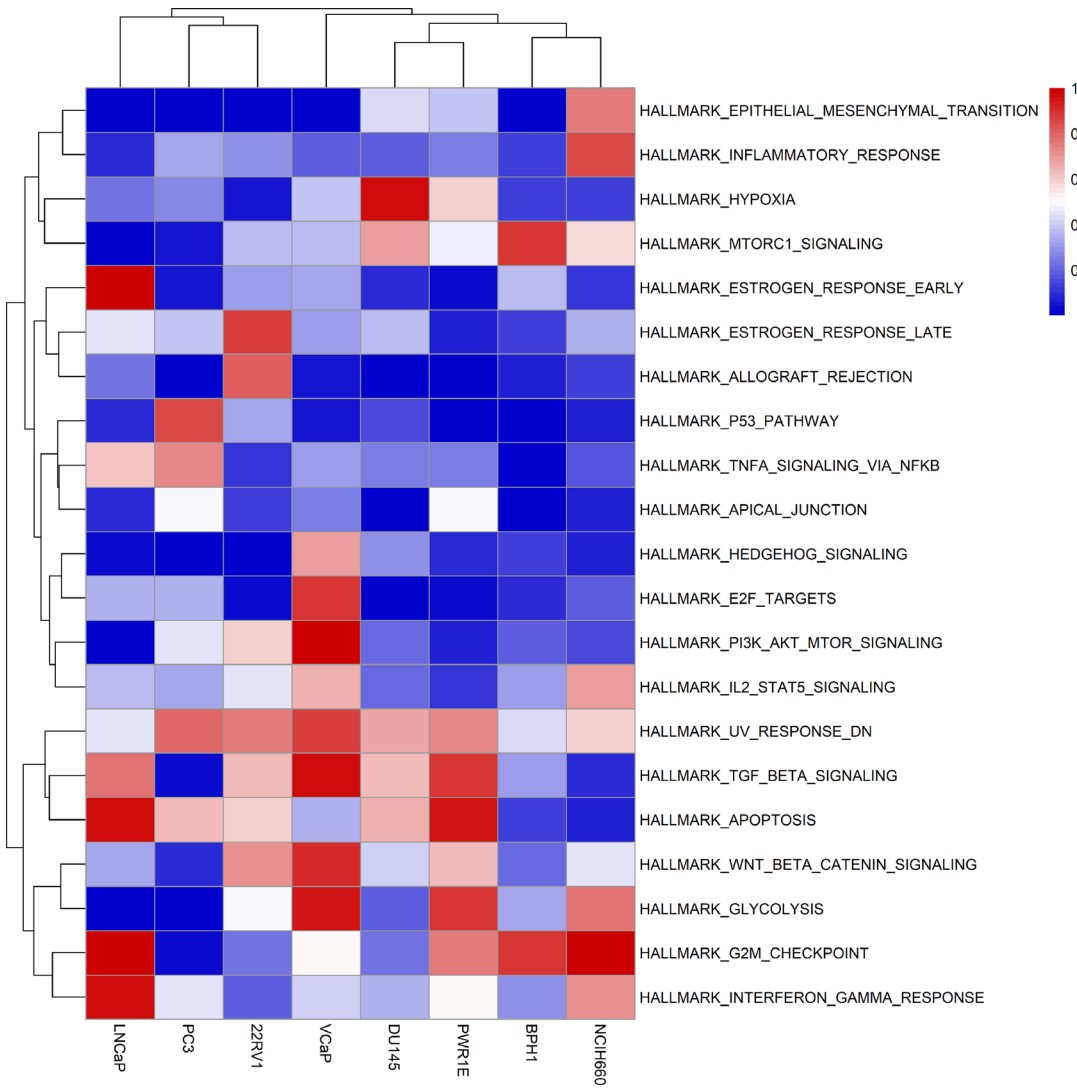

**Appendix 1—figure 25.** Results of the ssGSEA performed on the RNA dataset used to tailor the prostate cell lines.

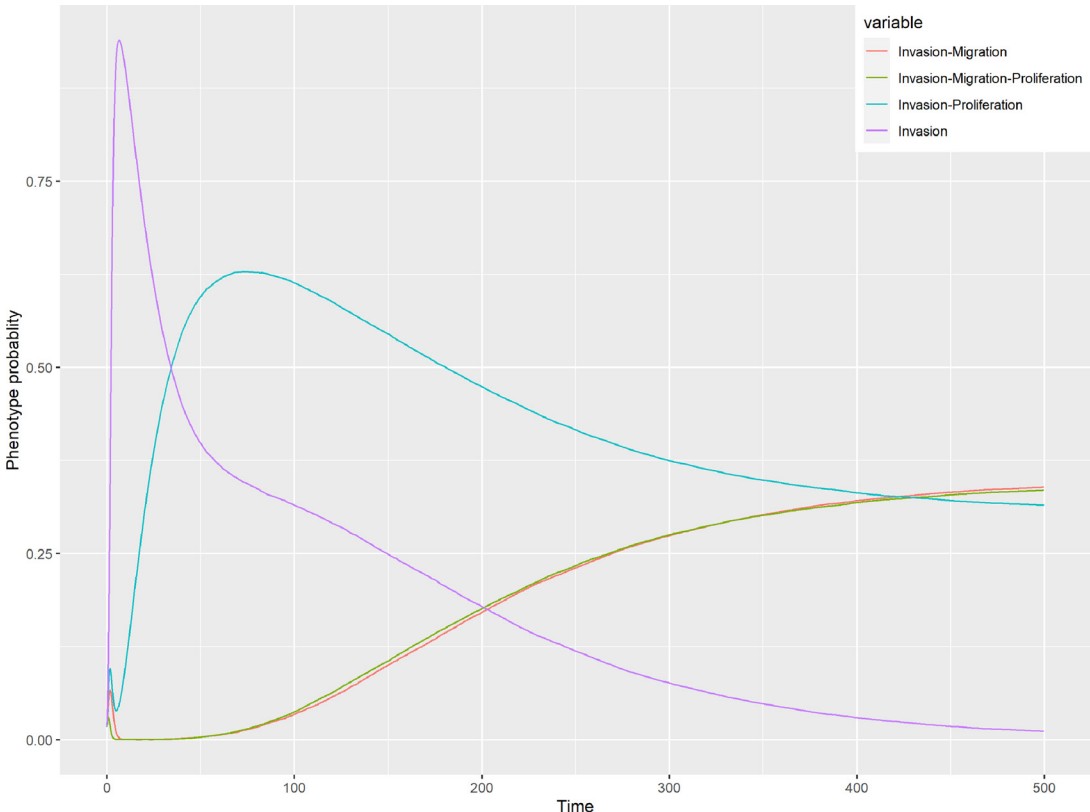

**Appendix 1—figure 26.** Phenotype probabilities of LNCaP model under random initial conditions.

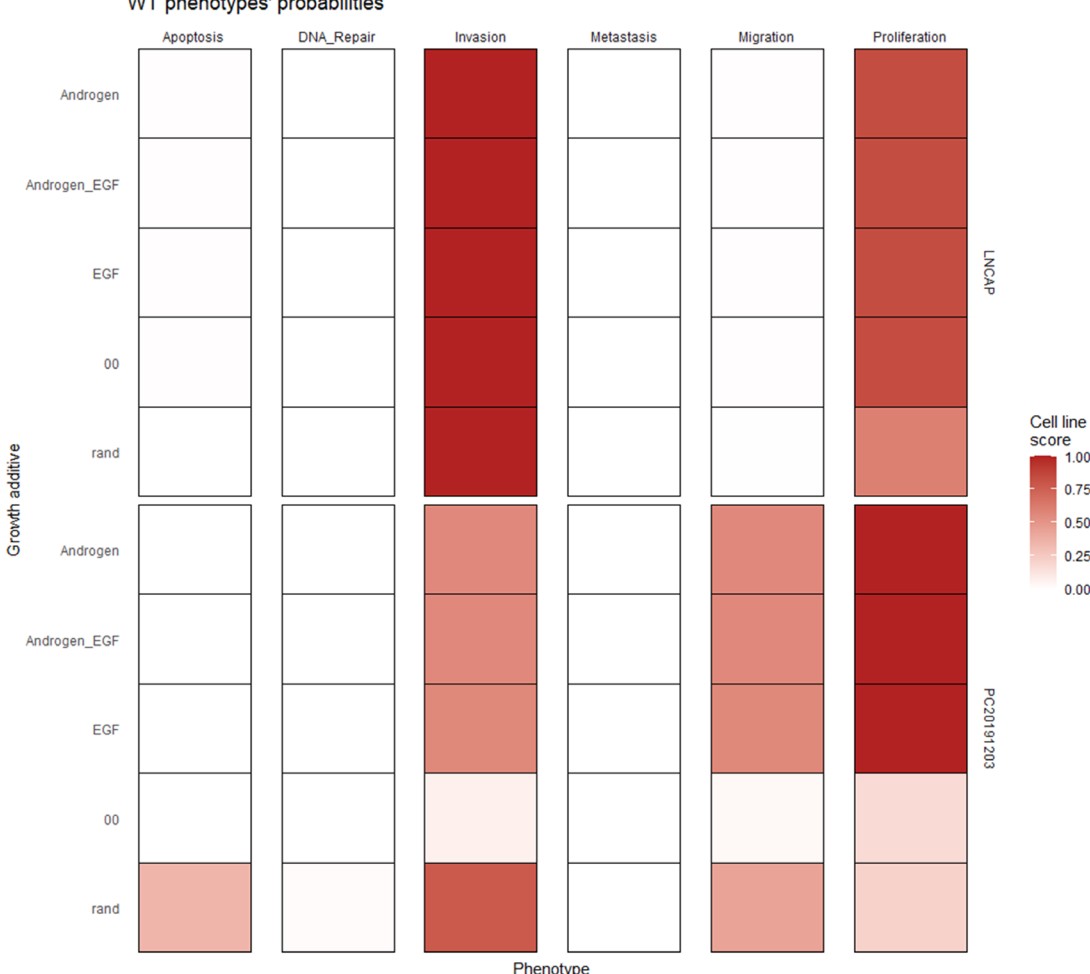

**Appendix 1—figure 27.** Wild type and LNCaP-specific model phenotype probability variations under four different growth conditions. Androgen stands for androgen presence, EGF for EGF presence, and 00 for lack of androgen and EGF.

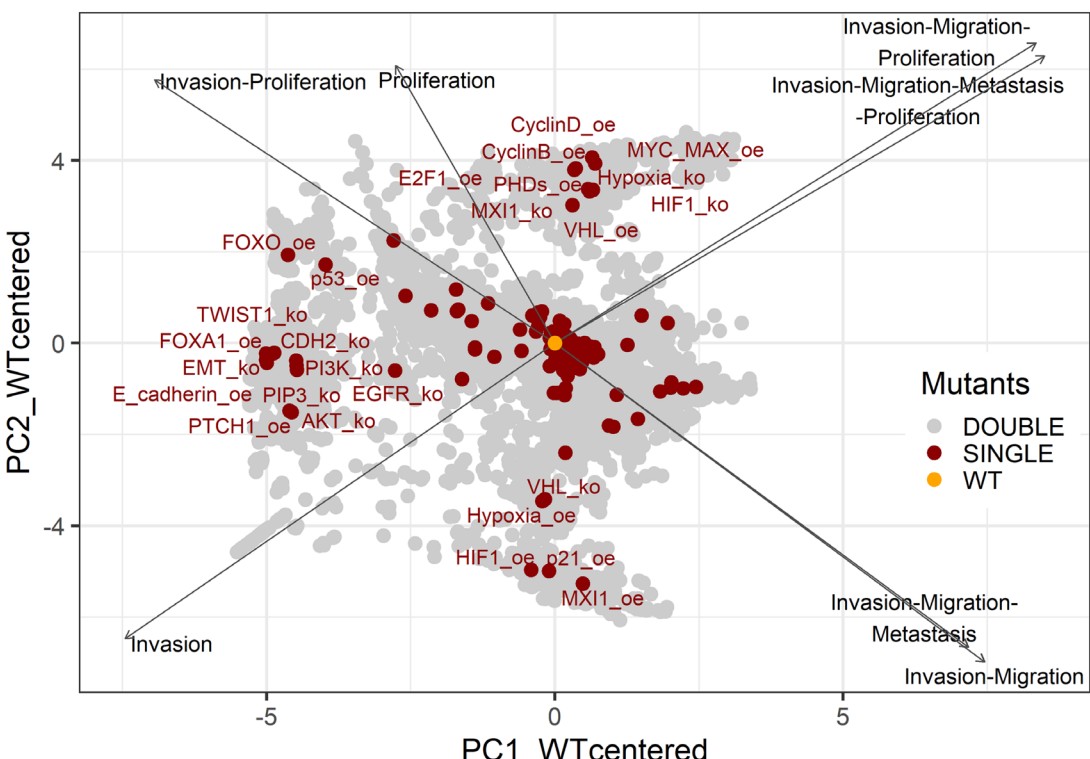

**Appendix 1—figure 28.** PCA of the 32,258 single and double LNCaP model mutants with combinations of the most probable phenotypes.

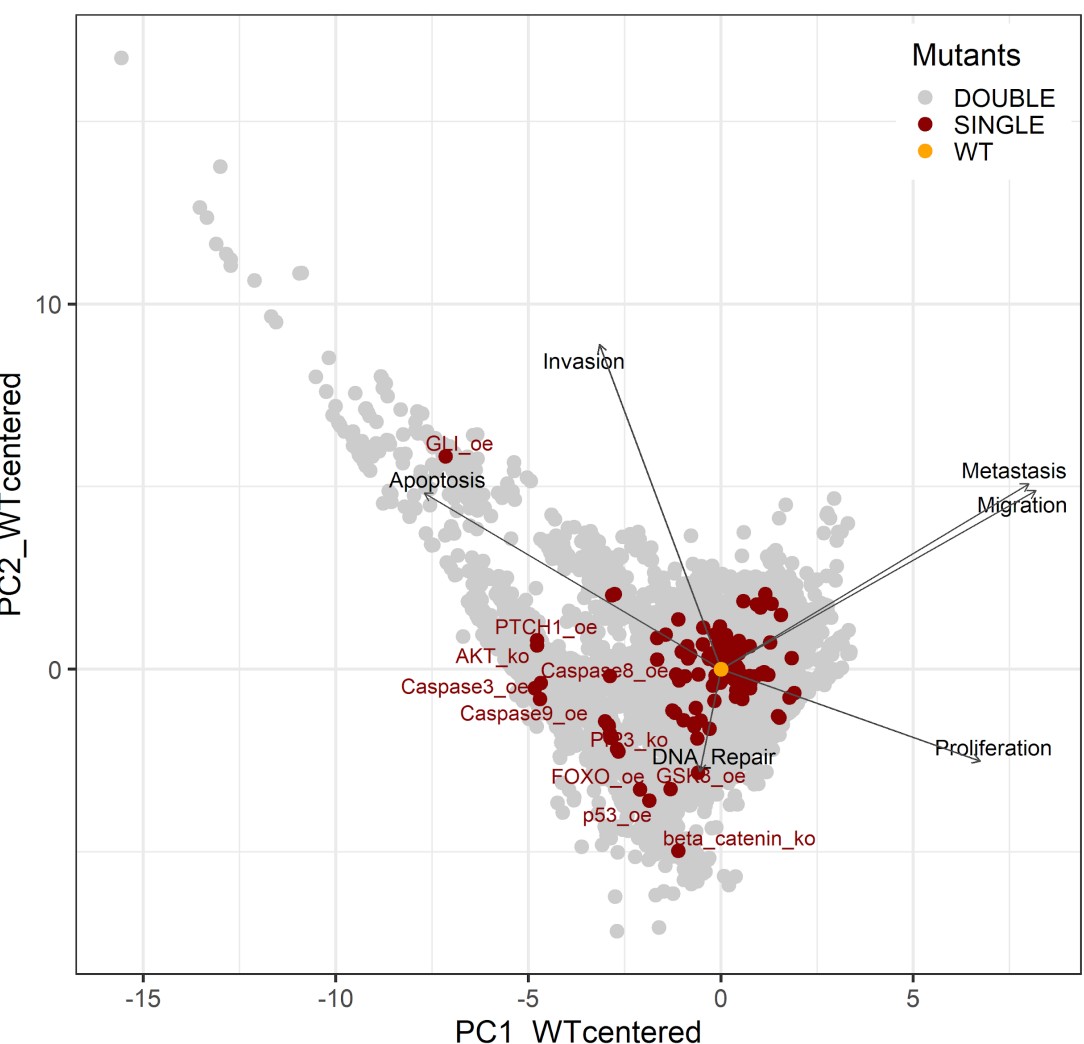

**Appendix 1—figure 29.** PCA of the 32,258 single and double LNCaP model mutants with the decomposition in single phenotypes.

A

Distribution of logical gates single perturbations of
Invasion_Migration_Proliferation phenotype

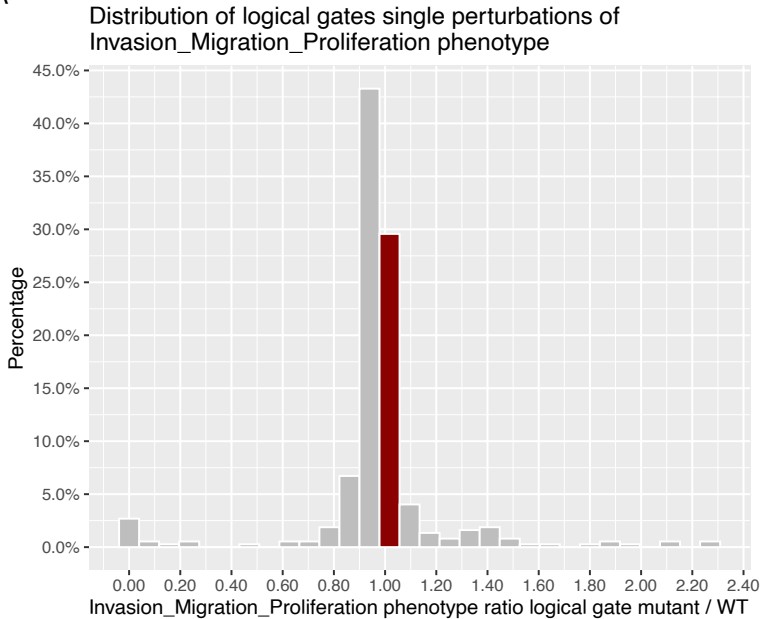

Distribution of logical gates ouble perturbations of
Invasion_Migration_Proliferation phenotype

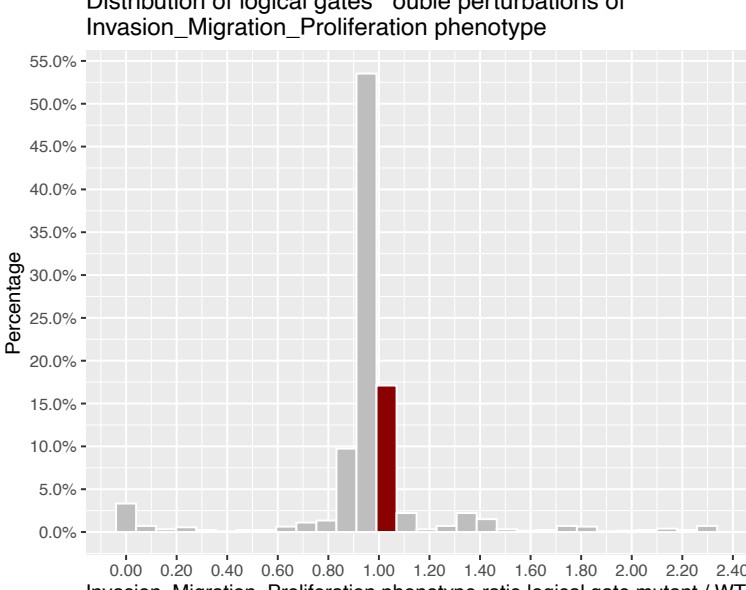

**Appendix 1—figure 30.** *Invasion-Migration-Proliferation* phenotype probability distribution across all mutants for logical gates. Bin where wild type value is found has been marked with dark red colour. (**A**) Phenotype probability using level one single perturbations; (**B**) Phenotype probability using level two double perturbations.

A

Distribution of double perturbations on nodes' logical gates that reduce
Invasion_Migration_Proliferation phenotype probability to zero

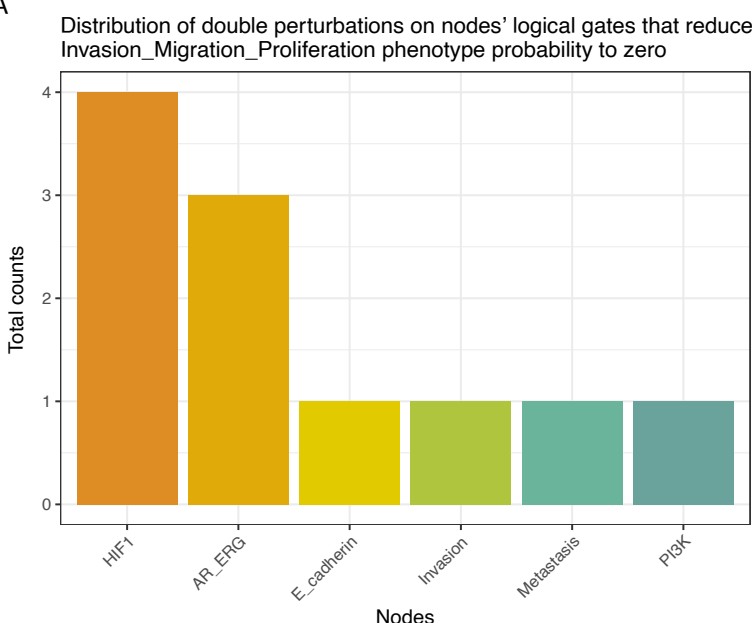

B

Distribution of single perturbations on nodes' logical gates that reduce
Invasion_Migration_Proliferation phenotype probability to zero

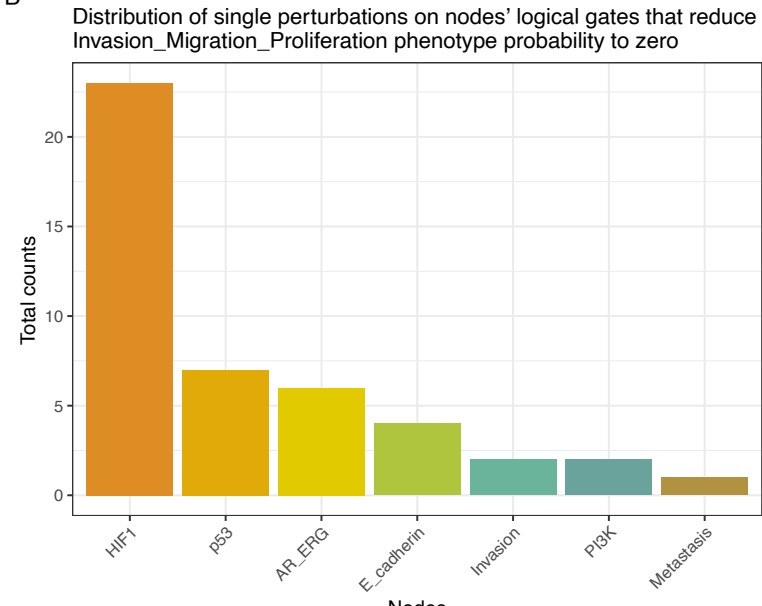

**Appendix 1—figure 31.** Distribution of perturbations on nodes' logical gates that reduce *Invasion-Migration-Proliferation* phenotype probability to zero. (**A**) Counts of level one single perturbations; (**B**) Counts of level two double perturbations.

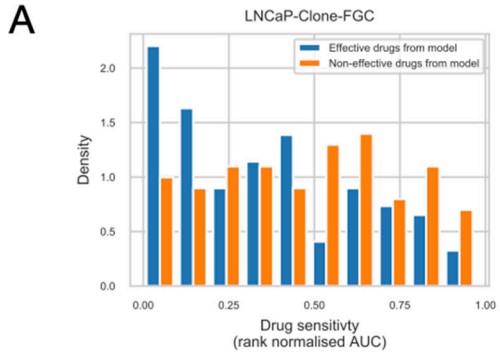

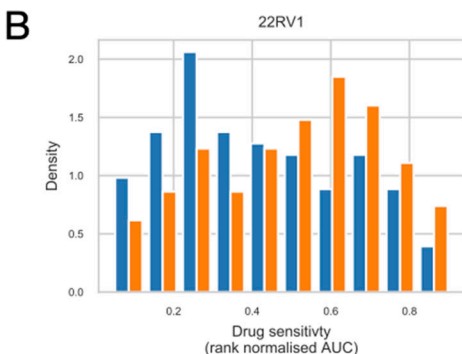

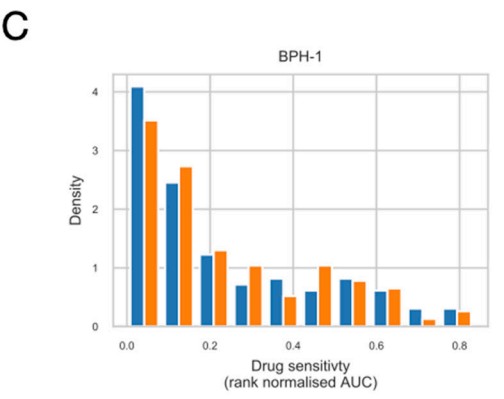

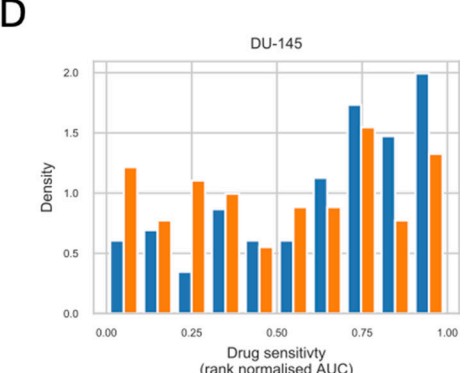

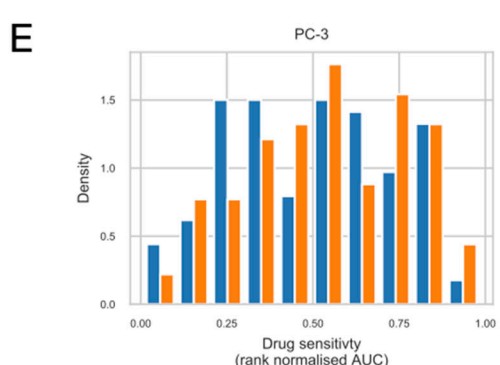

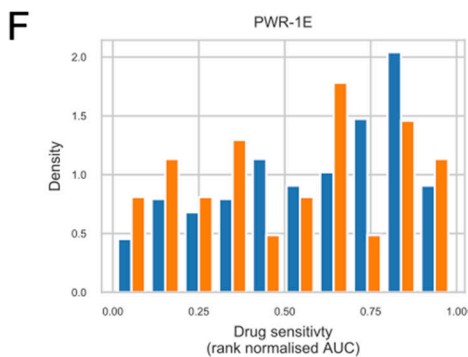

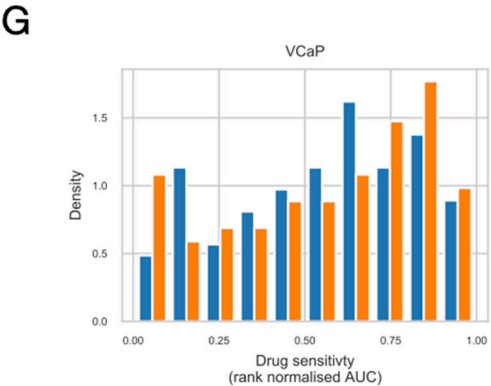

**Appendix 1—figure 32.** Drug sensitivity of the seven prostate cell lines. Rank normalised drug sensitivity (0: most

*Appendix 1—figure 32 continued on next page*

*Appendix 1—figure 32 continued*

sensitive; 1: most resistant, based on GDSC AUC drug sensitivity metric) for each GDSC drug across prostate cancer cell lines. Drugs are grouped to be predicted effective drugs based on the LNCaP Boolean model (orange) and predicted ineffective drugs (blue). Mann-Whitney U p-values for differences between the rank normalised drug sensitivity between predicted effective and ineffective drugs: (**A**) LNCaP, *P* = 0.00041 (more sensitive to LNCaP model-predicted drugs); (**B**) 22RV1, *P* = 0.0033 (more sensitive to LNCaP model-predicted drugs); (**C**) BPH-1, *P* = 0.31; (**D**) DU-145, *P* = 0.0026 (more resistant to LNCaP model-predicted drugs); (**E**) PC-3, *P* = 0.15; (**F**) PWR-1E, *P* = 0.075; (**G**) VCaP *P* = 0.38.

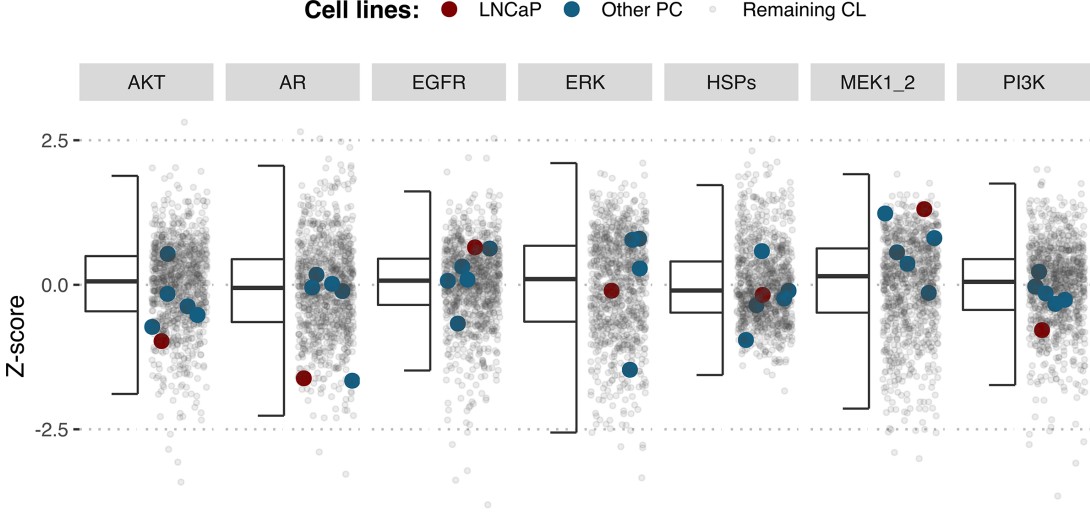

Different target nodes across all GDSC cell lines

**Appendix 1—figure 33.** Model-targeting drugs' sensitivities across prostate cell lines. GDSC Z-score was obtained for all the drugs targeting genes included in the model for all the prostate cell lines in GDSC. LNCaP is highlighted in red, the other seven prostate cell lines in blue and the rest of the GDSC cell lines are coloured in grey.

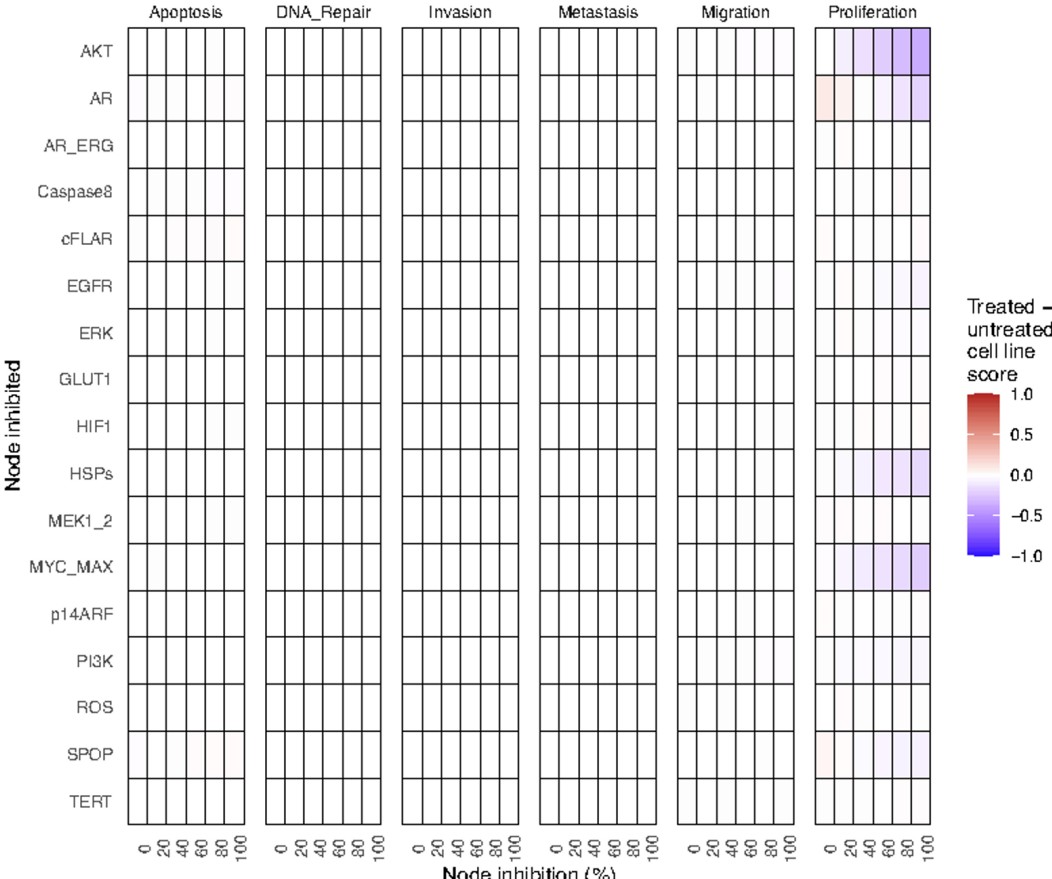

**Appendix 1—figure 34.** Phenotype score variations of the LNCaP model upon nodes' inhibition under *EGF* growth condition. Values of the scores are depicted with a colour gradient.

**Appendix 1—figure 35.** Phenotype score variations of the LNCaP model upon nodes inhibition under *AR, EGF, 00* and *AR_EGF* growth conditions. Values of the scores are depicted with a colour gradient.

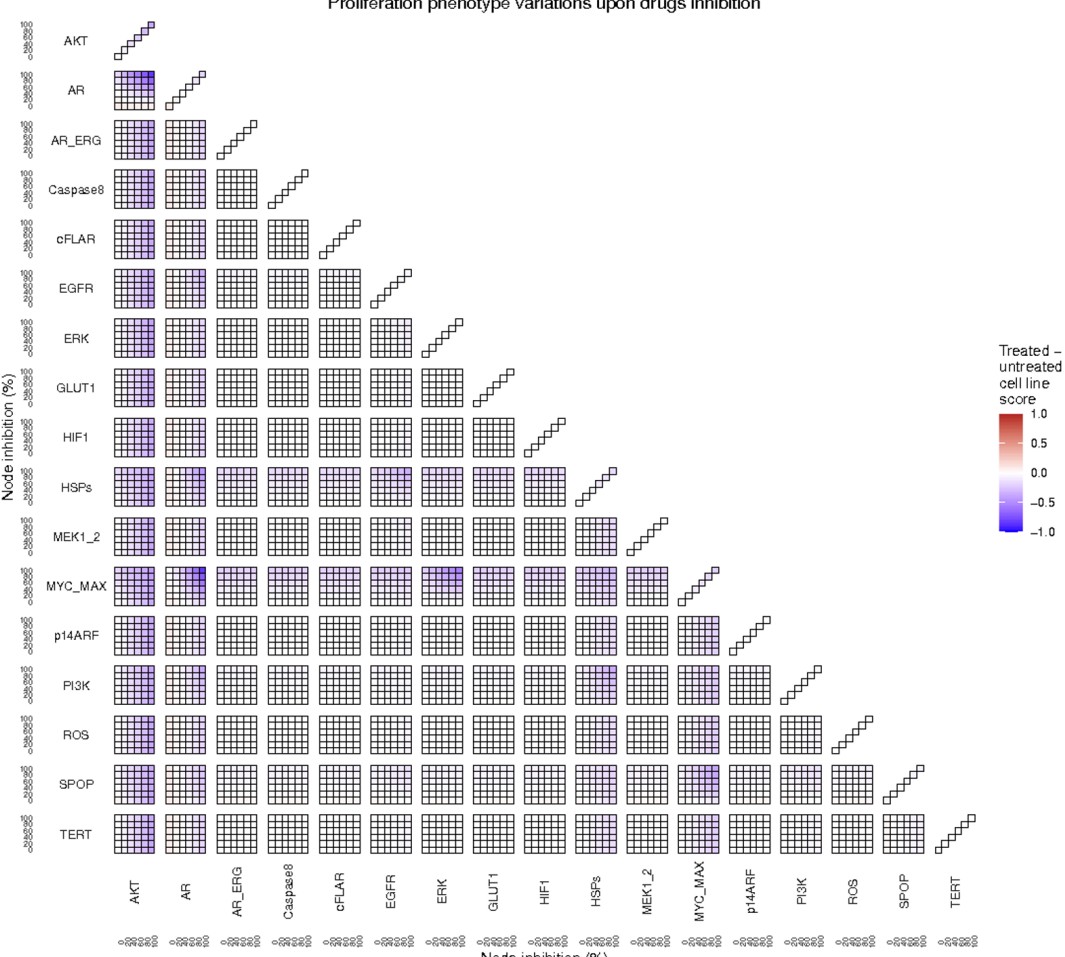

**Appendix 1—figure 36.** *Proliferation* phenotype score variations of the LNCaP model upon combined nodes inhibition under *EGF* growth condition. *Appendix 1—figure 4A* is a closer look at ERK and MYC_MAX combination and *Appendix 1—figure 4B* at HSPs and PI3K combination.

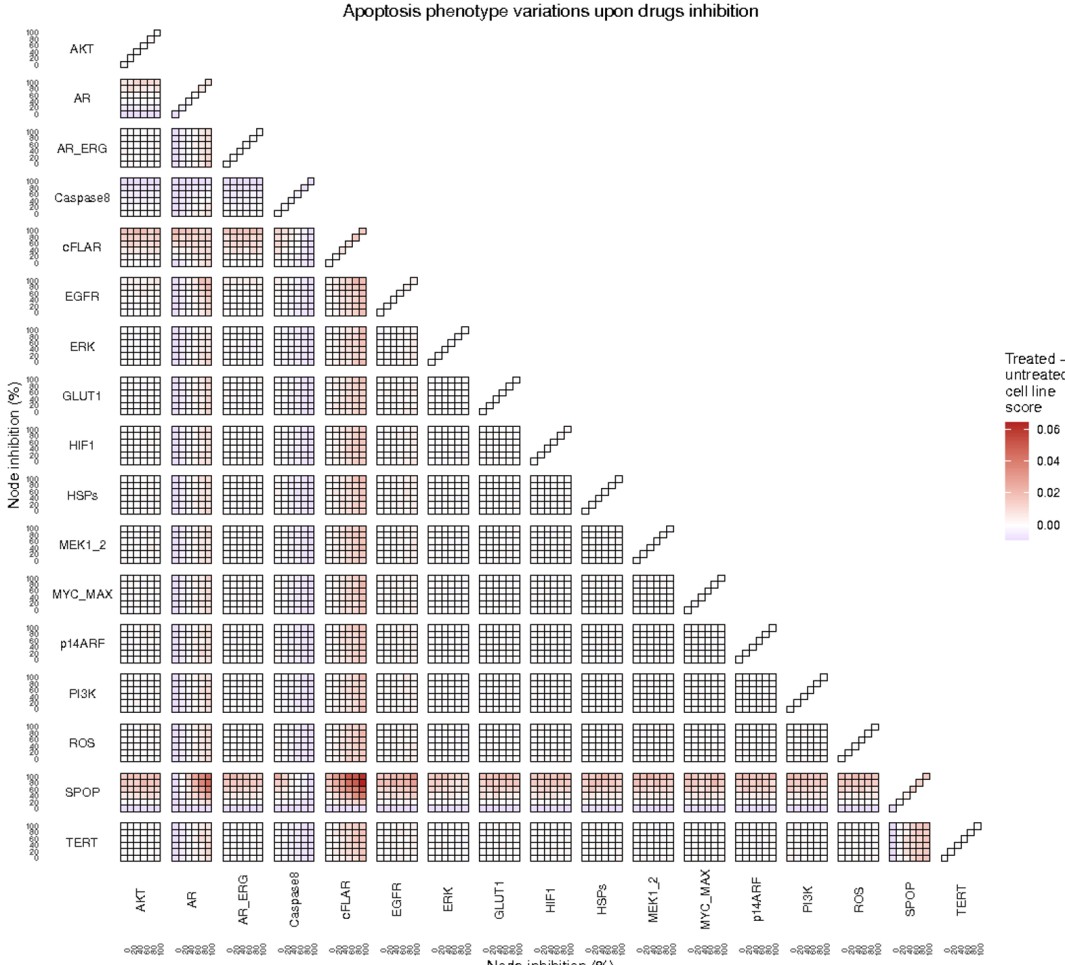

**Appendix 1—figure 37.** *Apoptosis* phenotype score variations of the LNCaP model upon combined nodes inhibition under *EGF* growth condition.

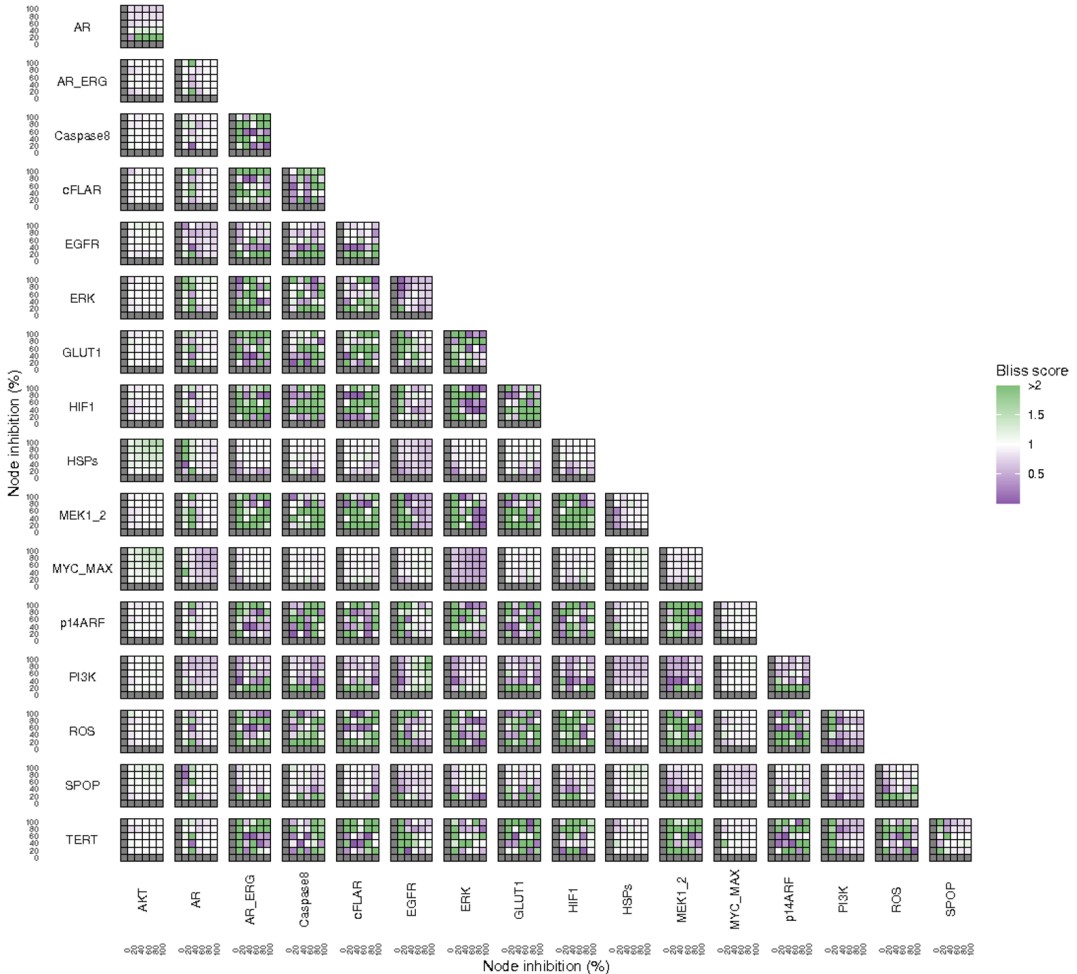

**Appendix 1—figure 38.** Bliss Independence synergies scores variations in *Proliferation* phenotype of the LNCaP model upon combined nodes inhibition under *EGF* growth conditions. Bliss Independence synergy score <1 is characteristic of drug synergy. *Appendix 1—figure 4C* is a closer look at ERK and MYC_MAX combination and *Appendix 1—figure 4D* at HSPs and PI3K combination, grey colour means one of the drugs is absent and thus no synergy score is available.

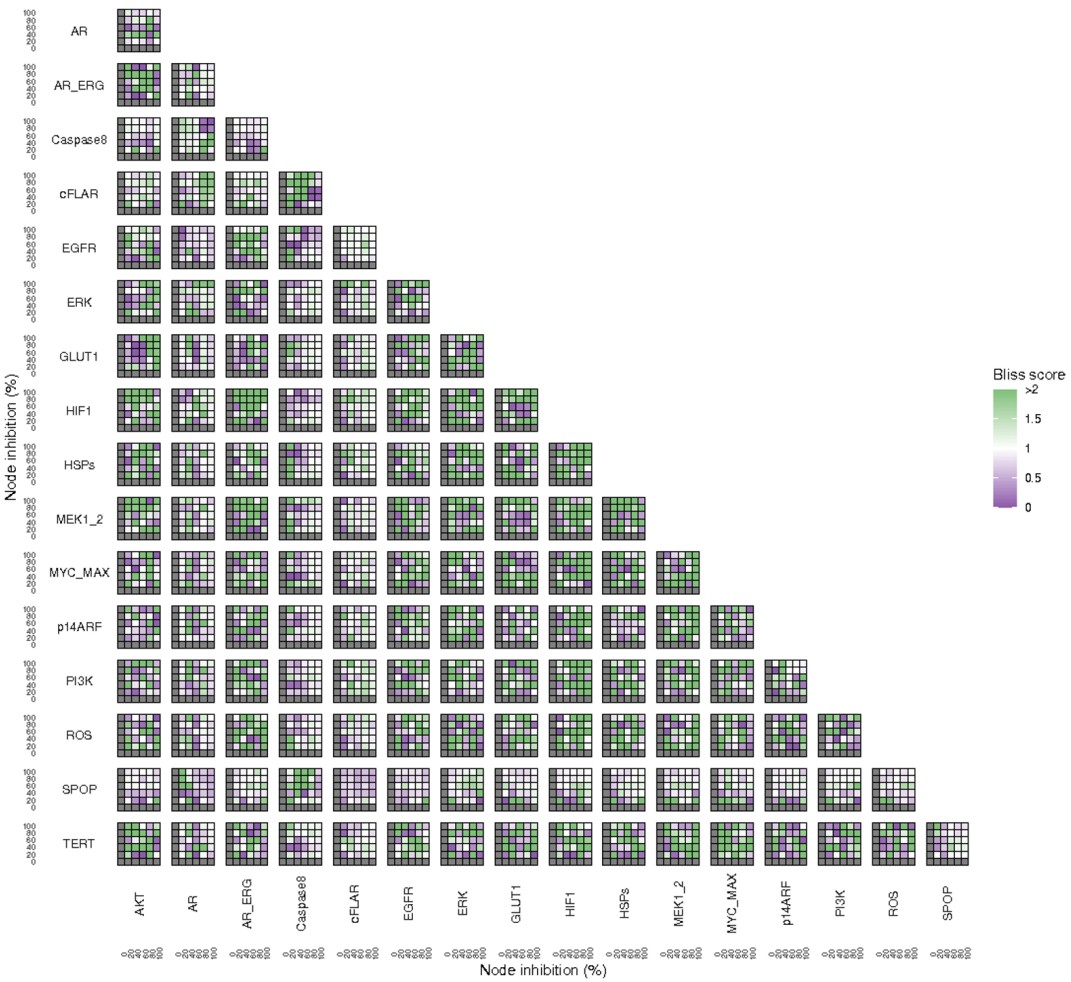

**Appendix 1—figure 39.** Bliss Independence synergies scores variations in *Apoptosis* phenotypes of the LNCaP model upon combined nodes inhibition under *EGF* growth conditions. Bliss Independence synergy score <1 is characteristic of drug synergy, grey colour means one of the drugs is absent and thus no synergy score is available.

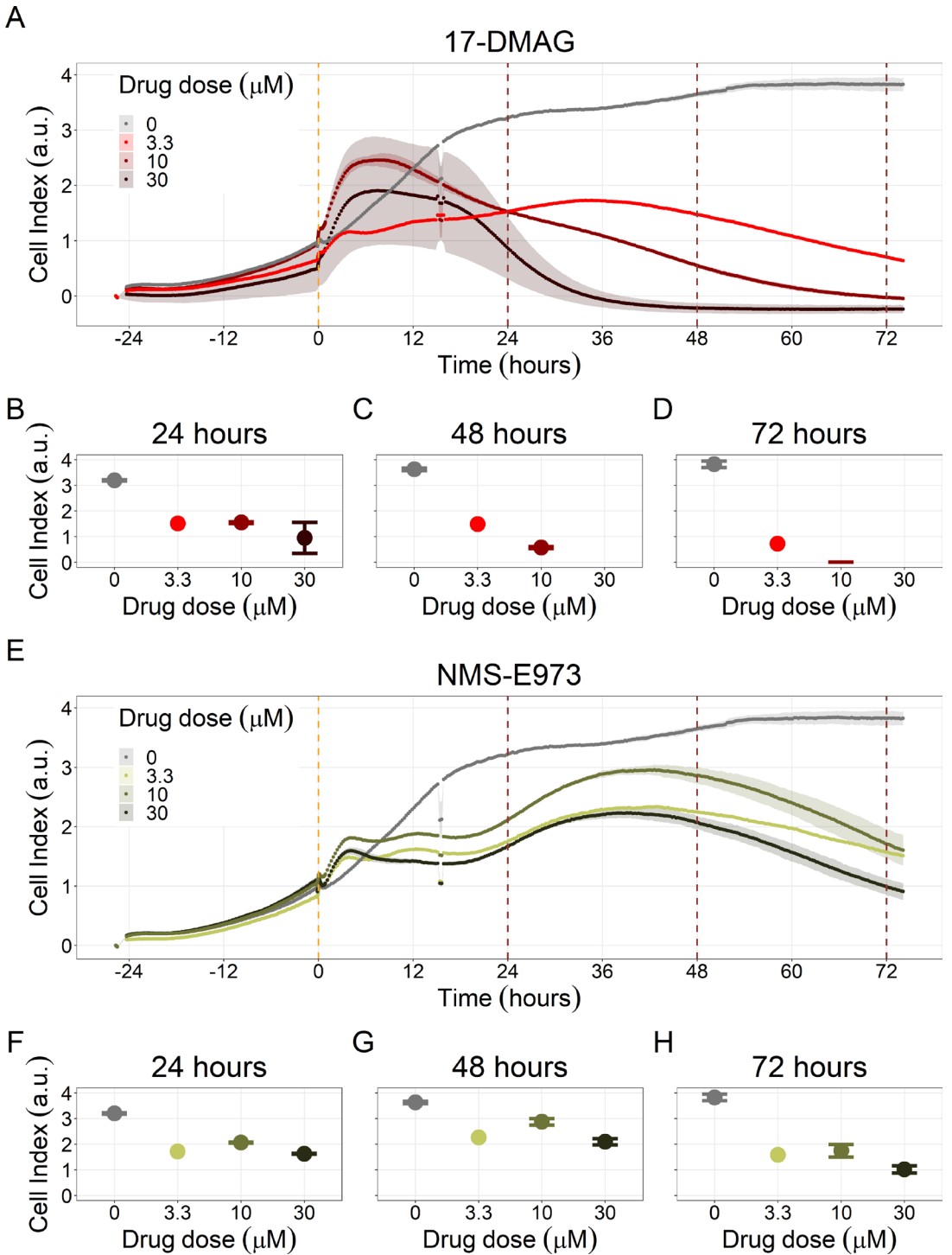

**Appendix 1—figure 40.** Hsp90 inhibitors resulted in dose-dependent changes in the LNCaP cell line growth.
(**A**) Real-time cell electronic sensing (RT-CES) cytotoxicity assay of Hsp90 inhibitor, 17-DMAG, that uses the Cell
Index as a measurement of the cell growth rate (see the Material and Methods section). The yellow dotted line
represents 17-DMAG addition. The brown dotted lines are indicative of the cytotoxicity assay results at 24 hours
(**B**), 48 hours (**C**) and 72 hours (**D**) after 17-DMAG addition. (**E**) RT-CES cytotoxicity assay of Hsp90 inhibitor,
NMS-E973. The yellow dotted line represents NMS-E973 addition. The brown dotted lines are indicative of the
cytotoxicity assay results at 24 hours (**F**), 48 hours (**G**) and 72 hours (**H**) after NMS-E973 addition.

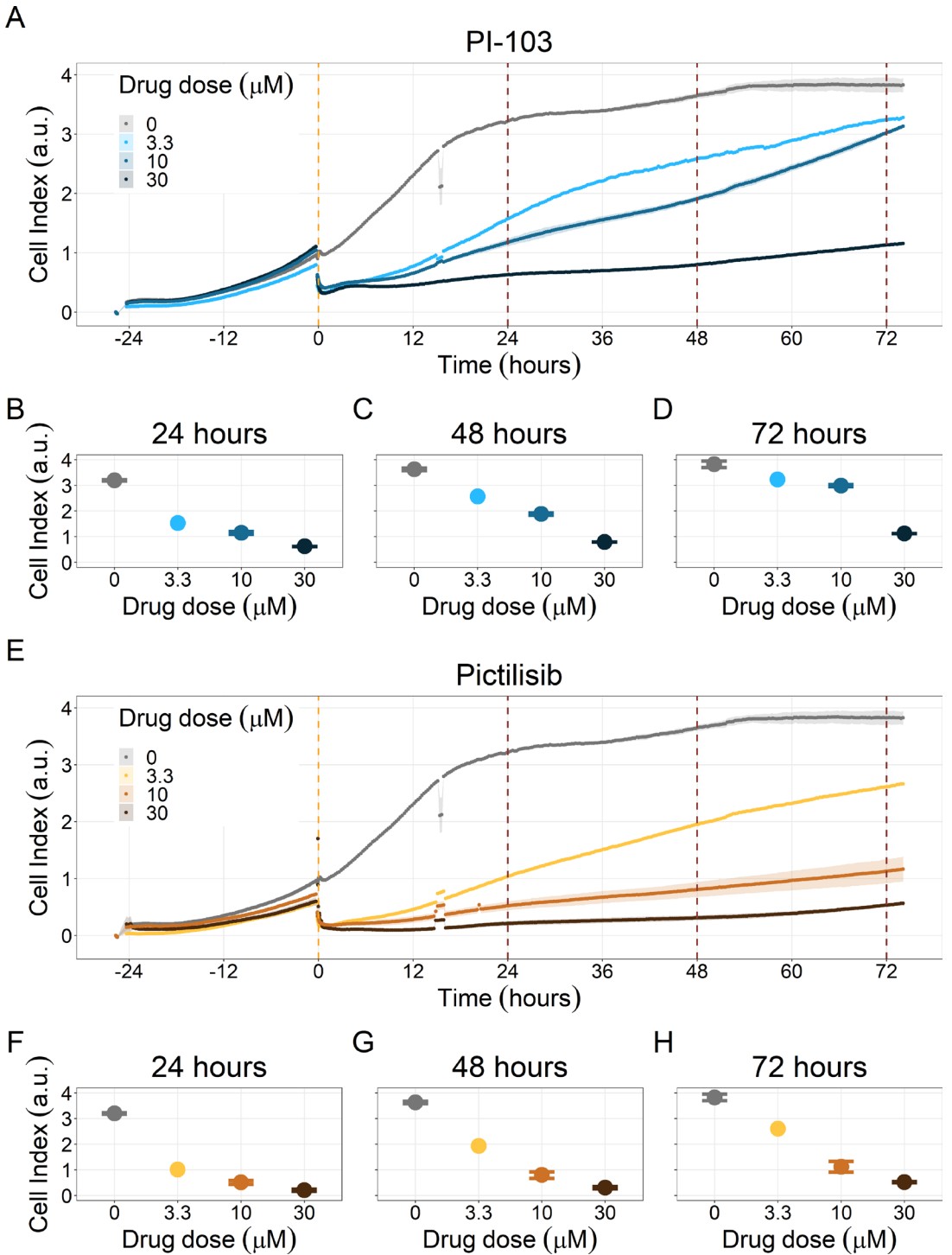

**Appendix 1—figure 41.** PI3K/AKT pathway inhibition with different PI3K/AKT inhibitors shows dose-dependent response in LNCaP cell line growth. (**A**) Real-time cell electronic sensing (RT-CES) cytotoxicity assay of PI3K/AKT pathway inhibitor, PI-103, that uses the Cell Index as a measurement of the cell growth rate (see the Material and Methods section). The yellow dotted line represents PI-103 addition. The brown dotted lines are indicative of the cytotoxicity assay results at 24 hours (**B**), 48 hours (**C**) and 72 hours (**D**) after PI-103 addition. (**E**) RT-CES cytotoxicity assay of PI3K/AKT pathway inhibitor, Pictilisib. The yellow dotted line represents Pictilisib addition. The brown dotted lines are indicative of the cytotoxicity assay results at 24 hours (**F**), 48 hours (**G**) and 72 hours (**H**) after Pictilisib addition.

**Appendix 1—key resources table**

| Reagent type (species) or resource | Designation | Source or reference | Identifiers | Additional information |
|---|---|---|---|---|
| gene (Homo-sapiens) | AKT1 | HGNC | HGNC:391 | |
| gene (Homo-sapiens) | AKT2 | HGNC | HGNC:392 | |
| gene (Homo-sapiens) | AKT3 | HGNC | HGNC:393 | |
| gene (Homo-sapiens) | AR | HGNC | HGNC:644 | |
| gene (Homo-sapiens) | CASP8 | HGNC | HGNC:1,509 | |
| gene (Homo-sapiens) | CFLAR | HGNC | HGNC:1,876 | |
| gene (Homo-sapiens) | EGFR | HGNC | HGNC:3,236 | |
| gene (Homo-sapiens) | MAPK1 | HGNC | HGNC:6,871 | |
| gene (Homo-sapiens) | MAPK3 | HGNC | HGNC:6,877 | |
| gene (Homo-sapiens) | SLC2A1 | HGNC | HGNC:11,005 | |
| gene (Homo-sapiens) | HIF1A | HGNC | HGNC:4,910 | |
| gene (Homo-sapiens) | HSP90AA1 | HGNC | HGNC:5,253 | |
| gene (Homo-sapiens) | HSP90AB1 | HGNC | HGNC:5,258 | |
| gene (Homo-sapiens) | HSP90B1 | HGNC | HGNC:12,028 | |
| gene (Homo-sapiens) | HSPA1A | HGNC | HGNC:5,232 | |
| gene (Homo-sapiens) | HSPA1B | HGNC | HGNC:5,233 | |
| gene (Homo-sapiens) | HSPB1 | HGNC | HGNC:5,246 | |

*Appendix 1 Continued on next page*

*Appendix 1 Continued*

| Reagent type (species) or resource | Designation | Source or reference | Identifiers | Additional information |
|---|---|---|---|---|
| gene (Homo-sapiens) | MAP2K1 | HGNC | HGNC:6,840 | |
| gene (Homo-sapiens) | MAP2K2 | HGNC | HGNC:6,842 | |
| gene (Homo-sapiens) | MYC | HGNC | HGNC:7,553 | |
| gene (Homo-sapiens) | MAX | HGNC | HGNC:6,913 | |
| gene (Homo-sapiens) | CDKN2A | HGNC | HGNC:1,787 | |
| gene (Homo-sapiens) | PIK3CA | HGNC | HGNC:8,975 | |
| gene (Homo-sapiens) | PIK3CB | HGNC | HGNC:8,976 | |
| gene (Homo-sapiens) | PIK3CG | HGNC | HGNC:8,978 | |
| gene (Homo-sapiens) | PIK3CD | HGNC | HGNC:8,977 | |
| gene (Homo-sapiens) | PIK3R1 | HGNC | HGNC:8,979 | |
| gene (Homo-sapiens) | PIK3R2 | HGNC | HGNC:8,980 | |
| gene (Homo-sapiens) | PIK3R3 | HGNC | HGNC:8,981 | |
| gene (Homo-sapiens) | PIK3R4 | HGNC | HGNC:8,982 | |
| gene (Homo-sapiens) | PIK3R5 | HGNC | HGNC:30,035 | |
| gene (Homo-sapiens) | PIK3R6 | HGNC | HGNC:27,101 | |
| gene (Homo-sapiens) | PIK3C2A | HGNC | HGNC:8,971 | |
| gene (Homo-sapiens) | PIK3C2B | HGNC | HGNC:8,972 | |

*Appendix 1 Continued*

| Reagent type (species) or resource | Designation | Source or reference | Identifiers | Additional information |
|---|---|---|---|---|
| gene (Homo-sapiens) | PIK3C2G | HGNC | HGNC:8,973 | |
| gene (Homo-sapiens) | PIK3C3 | HGNC | HGNC:8,974 | |
| gene (Homo-sapiens) | NOX1 | HGNC | HGNC:7,889 | |
| gene (Homo-sapiens) | NOX3 | HGNC | HGNC:7,890 | |
| gene (Homo-sapiens) | NOX4 | HGNC | HGNC:7,891 | |
| gene (Homo-sapiens) | NOX2 | HGNC | HGNC:2,578 | |
| gene (Homo-sapiens) | SPOP | HGNC | HGNC:11,254 | |
| gene (Homo-sapiens) | TERT | HGNC | HGNC:11,730 | |
| cell line (Homo-sapiens) | LNCaP clone FGC, prostate carcinoma (normal, Adult) | ATCC | CRL-1740 | RRID:CVCL_1379 |
| chemical compound, drug | RPMI 1640 Medium, GlutaMAX Supplement | Gibco | 61870–010 | |
| chemical compound, drug | 17-DMAG, an Hsp90 inhibitor | Sigma-Aldrich | 100,069 | *Pacey et al., 2011* |
| chemical compound, drug | NMS-E973, an Hsp90 inhibitor | MedChemExpress | HY-17547 | *Fogliatto et al., 2013* |
| chemical compound, drug | Pictilisib, an inhibitor of PI3Kα/δ | Thermo Scientific | 467861000 | *Zhan et al., 2017* |
| chemical compound, drug | PI-103, a multi-targeted PI3K inhibitor for p110α/β/δ/γ | Sigma-Aldrich | 528,100 | *Raynaud et al., 2009* |
| chemical compound, drug | Resazurin | Sigma-Aldrich | R7017 | *Szebeni et al., 2017* |
| chemical compound, drug | Dimethyl sulfoxide (DMSO) | Sigma-Aldrich | D8418 | |
| chemical compound, drug | Foetal bovine serum (FBS) | Gibco | 16140–071 | |

*Appendix 1 Continued on next page*

*Appendix 1 Continued*

| Reagent type (species) or resource | Designation | Source or reference | Identifiers | Additional information |
|---|---|---|---|---|
| chemical compound, drug | PenStrep antibiotics (Penicillin G sodium salt, and Streptomycin sulfate salt) | Sigma-Aldrich | P4333 | |
| software, algorithm | R language | https://www.R-project.org/ | | RRID:SCR_001905 |
| software, algorithm | Python language | https://www.python.org/ | | RRID:SCR_008394 |
| software, algorithm | MaBoS | https://github.com/maboss-bkmc/MaBoSS-env-2.0 | | *Stoll et al., 2017*; *Stoll et al., 2012* |
| software, algorithm | High-throughput mutant analysis | https://github.com/sysbio-curie/Logical_modelling_pipeline | | *Montagud et al., 2019* |
| software, algorithm | PROFILE | https://github.com/sysbio-curie/PROFILE | | *Béal et al., 2019* |
| software, algorithm | PROFILE_v2 | https://github.com/ArnauMontagud/PROFILE_v2 | | This work. Main text, Section "Personalisation of the prostate Boolean model" and Appendix 1, Sections 3,4,5 and 6. |
| software, algorithm | Prostate Boolean model | https://www.ebi.ac.uk/biomodels/MODEL2106070001; http://ginsim.org/model/signalling-prostate-cancer | | This work. Main text, Section "Boolean model construction" and Appendix 1, Section 1. |

