## [Editor Report]

This paper presents a mathematical model for prioritizing drugs for prostate cancer patients based on signal network database. The manuscript is of broad interest to the field of oncology and precision medicine. The methodology developed is sophisticated and relevant to real patient prostate cancer data. The predictions from the model are validated in an experimental setting and provide suggestions for the personalisation of prostate cancer treatment. The study can serve as a roadmap for future development of predictive, personalized models.

---

## [Decision Letter]

**Decision letter after peer review:**

Thank you for submitting your article "Patient–specific Boolean models of signaling networks guide personalized treatments" for consideration by *eLife*. Your article has been reviewed by 3 peer reviewers, and the evaluation has been overseen by a Reviewing Editor and Aleksandra Walczak as the Senior Editor. The following individual involved in review of your submission has agreed to reveal their identity: Réka Albert (Reviewer #2).

Essential revisions:

1. The main limitation is that the bioinformatics conclusion was validated using only one cell line experimentally. The authors might consider how to consolidate the general utility of this model.

2. Another major limitation is that this model points to the existing drug targets as highlighted in the signal databases. Is it possible to identify new drug targets? When there are multiple reagents hitting the same drug target, can it advise which chemical to use? The authors should comment on this limitation.

3. The authors should comment on the contribution of tumor microenvironment, and whether this model could address this issue since it seems mainly built on signaling in tumor cells.

4. The author used Monte–Carlo kinetic algorithm to generate the time trajectories for the state transition graph and set a maximum time to ensure an asymptotic solution. It would be better to provide more detail of situations if an asymptotic solution cannot be found or such solution can always be found and converge to the same result.

5. The author has not mentioned enough detail about the transition probability between states (e.g., does the transition probability based on prior knowledge, will the transition probability, will the transition probability change when constructing a personalized Boolean model, and how does it change if it changes). It will be good if the authors can provide a transition probability matrix as an example, and any influence from prior knowledge.

6. In the supplementary information, "Personalized Boolean model of prostate cell lines" section, the plots (S19/S20) used simulation result from random initial condition, does that mean the result is independent of the choice of initial condition? It would be helpful if the author could analyze the influence of the model output with different initial conditions (e.g., whether it will converge to the same results or different ones).

7. Why does the cell viability increase with drug (e.g. 2nM) compared to no drug (i.e. 0 nM) in Figure 7? Do the drugs promote proliferation in small doses?

8. It would be good to test the drug concentration cell viability experiment for larger doses of drug so we can actually see the maximal efficacy of these drugs. It seems for some of the drugs they don't even achieve their half–effect in the doses tested. Is there a reason for this choice?

9. Indicating the general criteria for the logical rules and giving an example in the Appendix would help a lot.

10. Text queries/suggestions:

– Figure 2, when the authors say phenotypes, is this also the 6 variable outputs that they mention in the text? If so, it might be good to say this in the caption so it's clear to readers.

– It's challenging to really fully grasp the full model, I think the readers would benefit from more reference to appropriate sections of the supplementary material.

– Can you elaborate on how the combined perturbations for already–developed drugs was performed? How was the model changed for a given drug combination?

– What is EGF, FGF, TGF etc? Are these acronyms? Since they are "physiological conditions of interest" it might be good for the readers to know what these represent.

– When you say "the final model accounts for 133 nodes and 449 edges" can this be linked to its biological counterpart. i.e. Is this then 133 proteins and 449 protein–protein pathways/interactions?

– '…such data can only be obtained with non–standard procedures such as microfluidics from patients' material ': The authors should make it clear about what kind of information is missing from those data making those mode unavailable.

– The author should clearly specify the number of pathways, genes, and cross–talks involved in their models. It is unclear how many components were integrated into the network to obtain the final network containing 133 nodes and 449 edges. Please also specify how many drugs and drug combinations participated in personalised drug prediction. The authors should clarify the number of drug and drug combination instead of the word of " several".

11. Figures queries/suggestions:

– Figure S1 the text cannot be read; it needs to be larger and the graphic needs to be higher quality.

– Figure 2 should also be much bigger, it's too difficult to make out the blue rectangles and in turn most of the paths are difficult to discern.

– Meaning for acronyms in Figure 4 should be given before they are used, i.e. PCA and GG.

– Provided more details of what is mean by Cell index (a.u.) in Figure 8 and Figure 9 in the caption.

– A figure summarising the combined treatment effects for the different patients with a figure to demonstrate how AKT was the top hit in Gleason Groups 1, 2 and 3 and so on, would be helpful.

– What is the strange dynamic occurring at about 15 hours where the points shift down and then jump back up?

– Figure3: Why there were a sharp increase of several elements before the decay for those output with 0 activities in the final states? Why the kinetics of decaying varied across different nodes? Any interpretation?

– Figure 4: The correlation between the Gleason scores and Proliferation score is not clear by the graphics. Any other means to show this?

– Figure 8/9 BCD/FGH seem redundant with Figure 8/9 A/E. You can combine the two types of figures. Also, there seems a discontinuous segment in Figure8/9A/E. Is it an editing error of images? You may consider integrate them as a whole panel.

*Reviewer #1 (Recommendations for the authors):*

Below are some queries/comments I have for the authors.

General queries:

– Why does the cell viability increase with drug (e.g. 2nM) compared to no drug (i.e. 0 nM) in Figure 7? Do the drugs promote proliferation in small doses?

– It would be good to test the drug concentration cell viability experiment for larger doses of drug so we can actually see the maximal efficacy of these drugs. It seems for some of the drugs they don't even achieve their half-effect in the doses tested. Is there a reason for this choice?

Text queries/suggestions:

– Figure 2, when the authors say phenotypes, is this also the 6 variable outputs that they mention in the text? If so, it might be good to say this in the caption so it's clear to readers

– It's challenging to really fully grasp the full model, I think the readers would benefit from more reference to appropriate sections of the supplementary material

– Can you elaborate on how the combined perturbations for already-developed drugs was performed? How was the model changed for a given drug combination?

– What is EGF, FGF, TGF etc? Are these acronyms? Since they are "physiological conditions of interest" it might be good for the readers to know what these represent

– When you say "the final model accounts for 133 nodes and 449 edges" can this be linked to its biological counterpart. i.e. Is this then 133 proteins and 449 protein-protein pathways/interactions?

Figures queries/suggestions:

– Figure S1 the text cannot be read; it needs to be larger and the graphic needs to be higher quality

– Figure 2 should also be much bigger, it's too difficult to make out the blue rectangles and in turn most of the paths are difficult to discern.

– Meaning for acronyms in Figure 4 should be given before they are used, i.e. PCA and GG.

– Provided more details of what is mean by Cell index (a.u.) in Figure 8 and Figure 9 in the caption

– A figure summarising the combined treatment effects for the different patients with a figure to demonstrate how AKT was the top hit in Gleason Groups 1, 2 and 3 and so on, would be helpful

– What is the strange dynamic occurring at about 15 hours where the points shift down and then jump back up?

*Reviewer #2 (Recommendations for the authors):*

Indicating the general criteria for the logical rules and giving an example in the Appendix would help a lot.

*Reviewer #3 (Recommendations for the authors):*

1. The main limitation is that the bioinformatics conclusion was validated using only one cell line experimentally. The authors might consider how to consolidate the general utility of this model.

2. Another major limitation is that this model points to the existing drug targets as highlighted in the signal databases. Is it possible to identify new drug targets? When there are multiple reagents hitting the same drug target, can it advise which chemical to use? The authors should comment on this limitation.

3. The authors should comment on the contribution of tumor microenvironment, and whether this model could address this issue since it seems mainly built on signaling in tumor cells.

4. The author used Monte-Carlo kinetic algorithm to generate the time trajectories for the state transition graph and set a maximum time to ensure an asymptotic solution. It would be better to provide more detail of situations if an asymptotic solution can not be found or such solution can always be found and converge to the same result.

5. The author has not mentioned enough detail about the transition probability between states (e.g., does the transition probability based on prior knowledge, will the transition probability, will the transition probability change when constructing a personalized Boolean model, and how does it change if it changes). It will be good if the authors can provide a transition probability matrix as an example, and any influence from prior knowledge.

6. In the supplementary information, "Personalized Boolean model of prostate cell lines" section, the plots (S19/S20) used simulation result from random initial condition, does that mean the result is independent of the choice of initial condition? It would be helpful if the author could analyze the influence of the model output with different initial conditions (e.g., whether it will converge to the same results or different ones).

---

## [Author Response]

Essential revisions:1. The main limitation is that the bioinformatics conclusion was validated using only one cell line experimentally. The authors might consider how to consolidate the general utility of this model.

We thank the editor and reviewers for their comments on how to improve the impact and clarity of our work. A convincing validation for patients would be to cross-validate the predictions of the model with already existing personalised treatments. This was done to some extent, with the example of the androgen receptor inhibitors that are more effective in high-grade tumour patients and that are indeed given in the clinics to prostate cancer patients with advanced tumour stages. We hope that a more systematic search will be done on clinical trials that are in progress.

The experimental validation on the LNCaP cell line proved that the candidates suggested by the personalised model were valid and can be seen as a proof of concept. Ideally, the drug targets would have been validated on more cell lines or more realistic models such as mice, but this validation would require another project, which is in our future plans.

2. Another major limitation is that this model points to the existing drug targets as highlighted in the signal databases. Is it possible to identify new drug targets? When there are multiple reagents hitting the same drug target, can it advise which chemical to use? The authors should comment on this limitation.

The results of our analyses are focused on existing drugs. Because of the size of the model, we chose to filter the "potential targeted nodes" with the list of existing drugs. In theory, we could provide a list of "potential targeted nodes", but we thought that focusing on the available drugs would increase the repurposing motivation of the work.

For the LNCaP example, we provide a complete list of potential targets without the filter in the supplemental information (Supplementary File 6 and Appendix 1, Section "High-Throughput mutant analysis of the LNCaP model"). This list is a ranked list of the effect of mutations on LNCaP behaviours. In this work, we have focused on mutants that knocked down the activity of the nodes and found drugs that cause this that are approved or in clinical trials (Table 1).

Even though our methodology can consider and simulate off-target effects of drugs, if these are known a priori and included in the list of potential targets, this work does not evaluate which one of the multiple reagents targeting a single node is better. The introduction of such an evaluation would be a good fit for follow-up work. We have added this to the Discussion.

3. The authors should comment on the contribution of tumor microenvironment, and whether this model could address this issue since it seems mainly built on signaling in tumor cells.

In the present work, we have not modelled the tumour microenvironment (TME) directly. We have only considered its indirect effects by modulating the activity of the model inputs that are representative of the status of the TME (e.g. the presence or absence of growth factors, oxygen, etc.). The focus of the model is intracellular, and it only addresses the effects that these environmental variables have on the signalling networks and how the cell reacts to these effects. We modified the first paragraph of the Results section to clarify this point: "These input nodes have no regulation. Their value is fixed according to the simulated experiment to represent the status of the microenvironmental characteristics (e.g. the presence or absence of growth factors, oxygen, etc.). A more complex multiscale approach would be required to consider the dynamical interaction with other cell types."

As mentioned in the modified text, we are expanding this work using a multiscale modelling framework (cf. with the tool PhysiBoSS (Letort et al. 2019. Bioinformatics)) to take into account these effects, but this is beyond the scope of the present work.

4. The author used Monte–Carlo kinetic algorithm to generate the time trajectories for the state transition graph and set a maximum time to ensure an asymptotic solution. It would be better to provide more detail of situations if an asymptotic solution cannot be found or such solution can always be found and converge to the same result.

The formalism used is based on a Boolean framework with stochastic simulations, which considers that solutions are estimated with probabilities of populations of trajectories, not calculated analytically. Provided that the maximum time is long enough and the number of trajectories high enough, we can ensure that the probabilities of the solutions will end up with similar values. We have proof in the supplementary material of the first article of MaBoSS that states that for the same initial conditions, it will always converge to the same stationary solution (Stoll *et al.* 2012). This is even more verified because MaBoSS uses a different seed for the random generator for each run. We tested several rounds of simulations and confirmed that the obtained probabilities are very close.

Another issue would be to characterise the types of attractors that we obtain: stable states or limit cycles. If there were some complex attractors such as limit cycles, they would eventually end up in an asymptotic solution anyway. However, another way to measure the "chaos" of the system is to compute the entropy and the transition entropy. In MaBoSS, we added this functionality to assess how deterministic the system is. A typical signature of the existence of a limit cycle is that the entropy is non-zero, whereas the transition entropy is zero, which is the case here. The existence of the limit cycle is confirmed by the probabilities of all the fixed points that do not add to 1. We have added two cells in the notebook (Supplementary File 2) to show these results.

5. The author has not mentioned enough detail about the transition probability between states (e.g., does the transition probability based on prior knowledge, will the transition probability, will the transition probability change when constructing a personalized Boolean model, and how does it change if it changes). It will be good if the authors can provide a transition probability matrix as an example, and any influence from prior knowledge.

The transition probabilities are based on omics data in the case of the personalised models rather than prior knowledge. To be more precise, for the generic prostate model, at first, we set all the probabilities to be equal to explore all the possible behaviours of the model; but for the personalised models, the values for the transition rates will depend on the transcriptomics data. Once they are fixed for a personalised model, they do not change anymore. The way they are set is done through our PROFILE methodology. We have added a short and intuitive description of the method in the Appendix, section "Primer on PROFILE". An example of the obtained transition rates and initial conditions is provided for the LNCaP model. These transition probabilities can be inspected in the corresponding CFG files ($u_NODE and $d_NODE) for the rest of the cell line models and the TCGA patients.

6. In the supplementary information, "Personalized Boolean model of prostate cell lines" section, the plots (S19/S20) used simulation result from random initial condition, does that mean the result is independent of the choice of initial condition? It would be helpful if the author could analyze the influence of the model output with different initial conditions (e.g., whether it will converge to the same results or different ones).

By "random initial condition", we mean that input nodes have a 50% chance of having an active value in the initial conditions. We ran 5000 trajectories and, using the Monte-Carlo kinetic algorithm, ended up with these phenotype scores for each personalised cell line. The choice of initial conditions is critical for the resulting probabilities, as can be seen in Figure 3 of the main text and in the jupyter notebook of the wild type model analysis.

For Figures S21 and S23, we decided to choose random initial conditions as a levelling ground so that all prostate cell lines would have as many non-zero probabilities as possible in their outputs. To show the effect of the initial conditions on the simulation outputs, we used the same 3 initial conditions from Figure 3 (all initial inputs OFF, growth factors ON and death signals ON) on the 7 prostate cells lines and the non-personalised wild type model (Figure S22). Here we see that different initial conditions can have a drastic effect on the outcome of the simulations: Apoptosis and DNA_Repair are fired up upon death signals are ON, and Proliferation and Invasion are activated upon growth factors are ON.

7. Why does the cell viability increase with drug (e.g. 2nM) compared to no drug (i.e. 0 nM) in Figure 7? Do the drugs promote proliferation in small doses?

This is a problem caused by the resazurin cell viability assay. In these assays, a deviation of 10-15% for in vitro cellular assays is an acceptable variation as it is a fluorescent assay that detects the cellular metabolic activity of living cells. Thus, in our analyses, we consider changes above 1.00 to be the same value as the controls.

We have added a sentence on the analysis of Figure 7 "This dose-dependent activity is more notable in Hsp90 drugs (NMS-E973,17-DMAG) than in the PI3K/AKT pathway (Pictilisib) ones and very modest for PI-103.".

We have also added a sentence on these deviations in the Methods section.

8. It would be good to test the drug concentration cell viability experiment for larger doses of drug so we can actually see the maximal efficacy of these drugs. It seems for some of the drugs they don't even achieve their half–effect in the doses tested. Is there a reason for this choice?

In our in vitro experiments, we first used the biologically relevant doses (in nanomolar concentrations) that were given in the literature for the respective agents. However, we found that we did not observe a proper dose-response for all substances in the endpoint measurements (resazurin assay, Figure 7). Thus, we repeated the experiments with real-time cytotoxicity assay (RT-CES) with higher drug concentrations (μM), which gave more accurate data with more than a hundred sampling points (Figure 8). The continuous recording of impedance in cells was used as a measurement of the cell growth rate and reflected the effects of different drugs.

9. Indicating the general criteria for the logical rules and giving an example in the Appendix would help a lot.

In general, and unless evidenced otherwise, we join the activators with OR gates and the inhibitors with AND NOT. This way of connecting the incoming arrows of a node is a strong assumption that we make as a first try and without any further knowledge, and which is often applied (Fumia and Martins, 2013), and other methods of Boolean rules' inferences (Lim et al. 2016, BMC Bioinformatics). Usually, the OR links information extracted from different articles. In addition, assigning the NOT gate to inhibitors is a way of increasing their weight so that they are, in fact, inhibiting. In any case, these assumptions are the first approach; if there is any information about any of the components, we modify the Boolean equation accordingly. For instance, if we know that two inhibitors only inhibit when both are present, we include that information and overwrite the previous formula.

We have included a new section in the Appendix called "Establishing the rules of the Boolean model", where we explain the choices that are made to write the logical rules. We exemplify it with a Boolean toy model.

10. Text queries/suggestions:– Figure 2, when the authors say phenotypes, is this also the 6 variable outputs that they mention in the text? If so, it might be good to say this in the caption so it's clear to readers.

Yes. The way we constructed the model is such that our outputs of the model are biological phenotypes or can be interpreted as phenotypic read-outs. We have changed the Figure 2 caption and included this description: "and dark blue rectangles [correspond] to outputs that represent biological phenotypes".

– It's challenging to really fully grasp the full model, I think the readers would benefit from more reference to appropriate sections of the supplementary material.

We have added references to the supplementary material sections in the main manuscript.

– Can you elaborate on how the combined perturbations for already–developed drugs was performed? How was the model changed for a given drug combination?

Drugs act as inhibitors of nodes of the models. As we explain in the "Drug simulations in Boolean models" section of Material and Methods, to have nodes that are inhibited at a certain level: "out of 1000 trajectories of the Gillespie algorithm, MaBoSS can simulate 70% of them with an activated AKT (fixed value to 1) and 30% with an inhibited AKT node (fixed value to 0). The phenotype scores for the 1000 trajectories are averaged, and these are considered to be representative of a model with a drug that partially inhibits 30% of the activity of AKT. The same applies for a combined drug inhibition: a simulation of 50% AKT activity and 50% PI3K will have 50% of them with an activated AKT and 50% with an activated PI3K. Combining them, this will lead to 25% of the trajectories with both AKT and PI3K active, 25% with both nodes inactive, 25% with AKT active and 25% with PI3K active."

The combinations of these single inhibitions were considered as a double inhibition of the model nodes. As explained above, we filtered the inhibited nodes to existing drugs to limit the space of search.

– What is EGF, FGF, TGF etc? Are these acronyms? Since they are "physiological conditions of interest" it might be good for the readers to know what these represent.

They are, in fact, acronyms for different growth factors: Epithelial Growth Factor (EGF), Fibroblast Growth Factor (FGF), Transforming Growth Factor β (TGFbeta), Tumour Necrosis Factor α (TNF α).

We have included these definitions in the manuscript in the Results section "Prostate Boolean model construction" and in the legend of Figure 2.

– When you say "the final model accounts for 133 nodes and 449 edges" can this be linked to its biological counterpart. i.e. Is this then 133 proteins and 449 protein–protein pathways/interactions?

A node in the model can represent one protein or several ones. "β_catenin" node in the model represents the *CTNNB1* gene, but the "AMPK" node represents genes *PRKAA1*, *PRKAA2*, *PRKAB1*, *PRKAB2*, *PRKAG1*, *PRKAG2*, *PRKAG3*. The correspondence list can be found in the file "Montagud2022_interactions_sources.xlsx" and "Montagud2022_nodes_in_pathways.xlsx" in Supplementary File 1.

We have added this info in the manuscript in the Materials and methods section, "Boolean model construction".

– '…such data can only be obtained with non–standard procedures such as microfluidics from patients' material ': The authors should make it clear about what kind of information is missing from those data making those mode unavailable.

Our PROFILE_v2 methodology does not use in vitro perturbation experiments such as the ones from Saez-Rodriguez et al. (2009) and Dorier et al. (2016), but rather bulk omics data. The perturbation data does not lack any kind of information to have these personalised models, but we consider that being able to personalise models without needing further experimentation is an asset of our method. In any case, PROFILE_v2 methodology and perturbation tools are compatible and complementary as they use different kinds of data as inputs.

We have extended the explanation in the Appendix and rewritten these sentences in the last paragraph of the Introduction section with the aim of presenting our claim in a more clear manner: "When summarising the biological knowledge into a network and translating it into logical terms, the obtained model is generic and cannot explain the differences and heterogeneity between patients' responses to treatments. To personalise models and capture these heterogeneities, models can be trained with dedicated perturbation experiments (Dorier et al., 2016; Saez-Rodriguez et al., 2009), but such data can only be obtained doing further experimentation on patients' material using, for instance, non-standard clinical procedures such as microfluidics (Eduati et al., 2020). To address this limitation, we developed a methodology to use different omics data that are more commonly available to personalise generic models to individual cancer patients or cell lines and verified that the obtained models correlated with clinical results such as patient survival information (Béal et al., 2019)."

– The author should clearly specify the number of pathways, genes, and cross–talks involved in their models. It is unclear how many components were integrated into the network to obtain the final network containing 133 nodes and 449 edges. Please also specify how many drugs and drug combinations participated in personalised drug prediction. The authors should clarify the number of drug and drug combination instead of the word of " several".

Regarding the number of pathways and genes, we have detailed all this information in two different files in the zipped folder Supplementary File 1:

­– Montagud2022_interactions_sources.xlsx details all the connections among nodes, the HUGO names corresponding to the target node, the interaction type, the source node of the interaction, the description of the interaction from the literature, the reference of the paper (in PMID or DOI), and the logical rule of the interaction.

– Montagud2022_nodes_in_pathways.xlsx details how we have organised the nodes in pathways for Figure 1.

We have added in the "Drug simulations in Boolean models" section of Material and Methods a paragraph to detail the amount of simulations performed: "We simulated the inhibition of 17 nodes of interest. These were the 16 nodes from Table 1 with the addition of the fused AR-ERG (Figures S32 and S33) and their 136 pairwise combinations (Figures S34 and S35). As we used 6 different levels of activity for each node, the resulting Figures S34 and S35 comprise a total of 4998 simulations for each phenotype (136 x 6 x 6 + 17 x 6)."

11. Figures queries/suggestions:– Figure S1 the text cannot be read; it needs to be larger and the graphic needs to be higher quality.

We improved the readability of the figure by increasing the font for the names of the pathways.

– Figure 2 should also be much bigger, it's too difficult to make out the blue rectangles and in turn most of the paths are difficult to discern.

We have improved the readability of the figure by increasing the font and its resolution. We can increase the size of the figure as much as the journal allows for it. In any case, the image is a vectorised SVG that, at least on a screen, can be zoomed-in to visualise in detail.

In addition, and apart from the SBML, MaBoSS and GINsim file formats, we have included the Cytoscape file of the network in the Supplementary File 1 so that readers can browse it.

– Meaning for acronyms in Figure 4 should be given before they are used, i.e. PCA and GG.

Figure 4 caption has been changed to include the definition of these acronyms: "(A) Centroids of the Principal Component Analysis of the samples according to their Gleason grades (GG)".

– Provided more details of what is mean by Cell index (a.u.) in Figure 8 and Figure 9 in the caption.

The Cell Index definition has been improved on the Material and Methods section, and we have added a reference to a technical paper on it: "Continuous recording of impedance in cells was used as a measurement of the cell growth rate and reflected by the Cell Index value [https://doi.org/10.1089/adt.2004.2.363]".

Figure 8 and 9 captions have been changed: "(A) Real-time cell electronic sensing (RT-CES) cytotoxicity assay of Hsp90 inhibitor, 17-DMAG, that uses the Cell Index as a measurement of the cell growth rate (see the Material and Methods section)".

– A figure summarising the combined treatment effects for the different patients with a figure to demonstrate how AKT was the top hit in Gleason Groups 1, 2 and 3 and so on, would be helpful.

We have extended Figures S19 (previous S17) and S20 (previous S18) with panels for each GG to show the top hits disaggregated by the Gleason grade. This also allows readers to visually inspect the results from the section "Personalised drug predictions of TCGA Boolean models".

– What is the strange dynamic occurring at about 15 hours where the points shift down and then jump back up?

This was a technical problem with the RT-CES 96-well E-plate reader. At that time, there was a short blackout in our laboratory, and the reader detected a minor voltage fluctuation while the uninterruptible power supply (UPS) was switched on. Thus, these differences are consistent across all samples and replicates: all wild type and drug reads decrease at that time point, except Pictilisib that slightly increases. We could have removed these data points as technical problems but considered that it was better to be transparent, and the overall dynamic was not affected.

We have included a note on the corresponding Methods section to state this explicitly: "Note that around hour 15 our RT-CES reader had a technical problem caused by a short blackout in our laboratory and the reader detected a minor voltage fluctuation while the uninterruptible power supply (UPS) was switched on. This caused differences that are consistent across all samples and replicates: all wild type and drug reads decrease at that time point, except Pictilisib that slightly increases. For the sake of transparency and as the overall dynamic was not affected, we decided to not remove these readings."

– Figure3: Why there were a sharp increase of several elements before the decay for those output with 0 activities in the final states? Why the kinetics of decaying varied across different nodes? Any interpretation?

The stochastic Boolean simulations were done imposing some initial values for input nodes but leaving the activation of internal nodes as random (output nodes are always set to OFF). The different initial conditions cause that at early times the internal nodes that were ON activate some downstream nodes, causing some output nodes to have a non-zero probability. These transient activations are damped upon the first few updates of the model state.

In addition, we know that there is a cycle in this model that causes trajectories to have transient behaviours that are eventually attracted by a stable asymptotic solution.

– Figure 4: The correlation between the Gleason scores and Proliferation score is not clear by the graphics. Any other means to show this?

We were hesitant to use any statistical conclusion on the distribution graphics because it is indeed difficult to differentiate among distributions. We observed some trends, and this is what we reported in the main text: "found that the density of high Proliferation score (close to 1, Figure 4B) tends to increase as the Gleason score increases (from low to intermediate to high)".

We have not found a good way to conclude with confidence that the Gleason grades correlate with the Proliferation or Apoptosis scores as we do not really quantify them. Note that our analyses using the phenotype scores are semi-quantitative: they make sense when compared to a reference value (as in the wild type model against the personalised cell line models) or when comparing groups (as in the TCGA patients and their GG).

Nevertheless, and as per the reviewer's request, we have studied these distributions statistically by using the Kruskal-Wallis rank-sum test to identify if the phenotype distributions across 3- and 5-stage GG could originate from different distributions. Then and only if the Kruskal-Wallis test was significant, we used Dunn's nonparametric pairwise multiple comparisons test to identify which pairs of groups were statistically different. We saw that *Apoptosis* distributions were significantly different (in 3- and 5-stage GG) as well as *Proliferation* and *Invasion* (only in 3-stage GG).

We have included these analyses in the main text and in Appendix 1, Section 4.1.

– Figure 8/9 BCD/FGH seem redundant with Figure 8/9 A/E. You can combine the two types of figures. Also, there seems a discontinuous segment in Figure8/9A/E. Is it an editing error of images? You may consider integrate them as a whole panel.

Figure 8 BCD are zoom-ins of A, and Figure 9 FGH are zoom-ins of E, as we explain in the caption using the brown lines at the 24, 48 and 72 hours marks. The information is the same but shows the results differently to allow readers to compare better the different concentrations of drugs. Nevertheless, we have removed the snippets (BCD and FGH panels) and left the main figures (A and E). The former figures 8 and 9 are now in the Appendix 1, as Figures S38 and S39.

The discontinuous segment is at the 15 hours mark, and, as explained three questions above, we think it is a technical problem that, for the sake of transparency, we would rather not remove. We have added a note in this regard in the methods section.

Reviewer #1 (Recommendations for the authors):Below are some queries/comments I have for the authors.General queries:– Why does the cell viability increase with drug (e.g. 2nM) compared to no drug (i.e. 0 nM) in Figure 7? Do the drugs promote proliferation in small doses?– It would be good to test the drug concentration cell viability experiment for larger doses of drug so we can actually see the maximal efficacy of these drugs. It seems for some of the drugs they don’t even achieve their half-effect in the doses tested. Is there a reason for this choice?Text queries/suggestions:– Figure 2, when the authors say phenotypes, is this also the 6 variable outputs that they mention in the text? If so it might be good to say this in the caption so it’s clear to readers– It’s challenging to really fully grasp the full model, I think the readers would benefit from more reference to appropriate sections of the supplementary material– Can you elaborate on how the combined perturbations for already-developed drugs was performed? How was the model changed for a given drug combination?– What is EGF, FGF, TGF etc? Are these acronyms? Since they are “physiological conditions of interest” it might be good for the readers to know what these represent– When you say “the final model accounts for 133 nodes and 449 edges” can this be linked to it’s biological counterpart. i.e. Is this then 133 proteins and 449 protein-protein pathways/interactions?Figures queries/suggestions:– Figure S1 the text cannot be read; it needs to be larger and the graphic needs to be higher quality.– Figure 2 should also be much bigger, it’s too difficult to make out the blue rectangles and in turn most of the paths are difficult to discern.– Meaning for acronyms in Figure 4 should be given before they are used, i.e. PCA and GG.– Provided more details of what is mean by Cell index (a.u.) in Figure 8 and Figure 9 in the caption.– A figure summarising the combined treatment effects for the different patients with a figure to demonstrate how AKT was the top hit in Gleason Groups 1, 2 and 3 and so on, would be helpful.– What is the strange dynamic occurring at about 15 hours where the points shift down and then jump back up?

These questions were addressed in the "Essential revision" section.

Reviewer #2 (Recommendations for the authors):Indicating the general criteria for the logical rules and giving an example in the Appendix would help a lot.

This question was addressed in the Essential revision section.

Reviewer #3 (Recommendations for the authors):1. The main limitation is that the bioinformatics conclusion was validated using only one cell line experimentally. The authors might consider how to consolidate the general utility of this model.2. Another major limitation is that this model points to the existing drug targets as highlighted in the signal databases. Is it possible to identify new drug targets? When there are multiple reagents hitting the same drug target, can it advise which chemical to use? The authors should comment on this limitation.3. The authors should comment on the contribution of tumor microenvironment, and whether this model could address this issue since it seems mainly built on signaling in tumor cells.4. The author used Monte-Carlo kinetic algorithm to generate the time trajectories for the state transition graph and set a maximum time to ensure an asymptotic solution. It would be better to provide more detail of situations if an asymptotic solution can not be found or such solution can always be found and converge to the same result.5. The author has not mentioned enough detail about the transition probability between states (e.g., does the transition probability based on prior knowledge, will the transition probability, will the transition probability change when constructing a personalized Boolean model, and how does it change if it changes). It will be good if the authors can provide a transition probability matrix as an example, and any influence from prior knowledge.6. In the supplementary information, "Personalized Boolean model of prostate cell lines" section, the plots (S19/S20) used simulation result from random initial condition, does that mean the result is independent of the choice of initial condition? It would be helpful if the author could analyze the influence of the model output with different initial conditions (e.g., whether it will converge to the same results or different ones).

These questions were addressed in the "Essential revision" section.